# Adaptive High-Dimensional Subspace Evolution Based on Broad Learning System and Error-Correcting Output Codes

## Abstract

High-dimensional data (HDD) commonly exhibit complex hierarchical structural characteristics; however, existing approaches typically employ fixed subspace evolution strategies that fail to adapt to the inherent hierarchical diversity across different datasets, resulting in suboptimal revelation of underlying discriminative patterns. Considering this critical limitation, we propose an adaptive high-dimensional subspace evolution algorithm (AHSE) featuring a dual-branch collaborative architecture: the series branch leverages Cholesky decomposition-based incremental Broad Learning System (BLS) to efficiently evolve cascaded subspaces tailored to distinct types of high-dimensional hierarchies; the parallel branch, built on multiple subspace evolution bases, utilizes post-hoc error-correcting output codes (ECOCs) for robust spatial encoding and evolutionary optimization. Both branches converge into a lightweight circuit, forming a closed evolutionary loop. Owing to the hierarchy-tailored evolution strategy, AHSE excels in various HDD tasks such as image pattern recognition, speech emotion recognition, and few-shot learning. Moreover, we offer a rigorous theoretical analysis of the mechanism and robustness guarantee of ECOCs on BLS, further promoting the integrity of AHSE.

## 1 Introduction

In the era of big data, the dimensionality of data features has witnessed explosive growth, spanning various fields such as bioinformatics, computer vision, and sensor networks (Giraud, 2021). High-dimensional data (HDD) describe datasets characterized by a substantial number of variables, components, features, or attributes available for analysis (Thudumu et al., 2019). Data hierarchy refers to the intrinsic organization of discriminative patterns within high-dimensional feature spaces. Evolution patterns describe how the learning algorithm dynamically adapts its feature selection strategy to match these underlying structures. The subspace evolution problem in HDD analysis refers to the process of dynamically constructing a sequence of feature subspaces $\{X_t\}_{t=1}^T$ where each subsequent subspace either expands, contracts, or transforms the previous one based on learning feedback. Unlike static dimensionality reduction techniques (e.g., PCA, LDA) that generate a single optimal subspace, subspace evolution creates an adaptive trajectory through the feature space. The formal problem can be stated as: Given HDD $X \in \mathbb{R}^{N \times M}$ and class labels $Y$, find an evolutionary function $\Psi$ that generates a sequence of subspaces $X_1, X_2, ..., X_T$ s.t. $X_t = \Psi(X_{t-1}, f_{t-1}(X_{t-1}, Y), Y)$, where $f_{t-1}$ is a classifier trained on the previous subspace. The goal is to discover an evolution path that maximizes final classification performance while minimizing computational complexity. As the dimensionality increases, the complexity of data analysis grows in tandem, thereby necessitating more advanced methodologies to effectively process such datasets.

While numerous approaches such as deep neural networks (DNNs) (Rumelhart et al., 1986; Simonyan & Zisserman, 2014; He et al., 2016; Dosovitskiy et al., 2020), Broad Learning System (BLS) (Chen & Liu, 2017; Chen et al., 2018), and deep-broad hybrids (Liu et al., 2020; Lei et al., 2024), have been developed to handle HDD, most existing methods employ fixed or single subspace evolution strategies that fail to adapt to the inherent hierarchical diversity across different datasets. This limitation becomes particularly evident when comparing datasets with fundamentally different structural properties—such as MNIST (LeCun et al., 1998)'s dense hierarchical pattern (see Fig. 1a)

versus Fashion-MNIST (Xiao et al., 2017)'s periodic hierarchical structure (see Fig. 1b)—where a one-size-fits-all evolution approach inevitably leads to suboptimal performance.

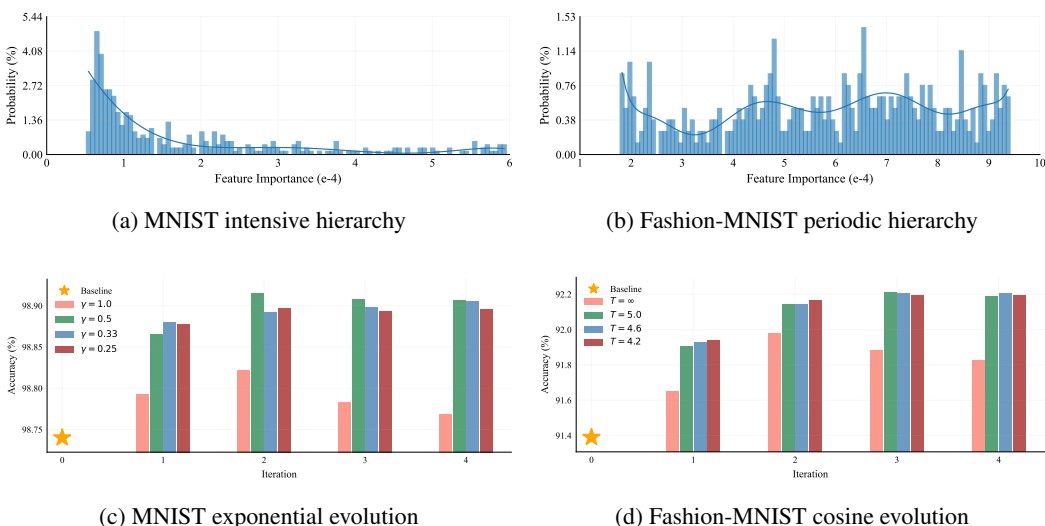

(a) MNIST intensive hierarchy

(b) Fashion-MNIST periodic hierarchy

(c) MNIST exponential evolution

(d) Fashion-MNIST cosine evolution

Figure 1: Analysis of evolutionary patterns for specific hierarchies. For MNIST, $\gamma$ is the evolution rate. For Fashion-MNIST, $T$ is the evolution period. $\gamma = 1$ or $T = \infty$ implies fixed evolution.

The core issue lies in the mismatch between static subspace evolution mechanisms and the dynamic nature of HDD hierarchies. When evolution rates and patterns remain fixed regardless of data characteristics, models either overfit to specific structural patterns or fail to capture discriminative features at appropriate scales. Our analysis reveals that different datasets require fundamentally different evolutionary trajectories: MNIST benefits from rapid, aggressive subspace evolution due to its dense hierarchical structure (see Fig. 1c), while Fashion-MNIST requires more gradual, careful evolution to preserve its periodic patterns (see Fig. 1d). This insight forms the foundation of our adaptive subspace evolution principle—rather than imposing a fixed evolution schedule, we dynamically adjust the evolution process based on the data's intrinsic hierarchies.

To operationalize this principle, we introduce an adaptive high-dimensional subspace evolution algorithm termed AHSE, which fundamentally rethinks how subspace evolution should be conducted in high-dimensional spaces. AHSE employs a dual-branch architecture that synergistically combines series and parallel subspace evolution strategies: *The **SEED** branch* implements a dynamic series evolution process that incrementally refines subspaces through feature-prioritized evolution. At each evolutionary step, SEED performs three critical operations: (i) subspace forward decay for feature subset screening; (ii) subspace alignment to mitigate distribution drift between evolutionary stages; (iii) sample weight evolution to progressively focus on challenging instances. This creates an adaptive evolutionary trajectory where the pace and direction of subspace evolution automatically adjust to the data's hierarchical characteristics. *The **PATH** branch* complements SEED by evolving multiple hierarchical subspaces in parallel. PATH leverages the evolutionary information from SEED to construct complementary subspace pairs, which are then fused and encoded using error-correcting output codes (ECOCs). Crucially, PATH incorporates a novel optimization mechanism called Flame that dynamically refines the evolutionary target space, enhancing robustness against noise and improving generalization. *The **SPOT** circuit* dynamically balances the efficiency of SEED with the robustness of PATH. SPOT calculates validation-based weights to optimally combine the outputs of both branches, creating a closed-loop evolutionary system.

The main contributions of this work are: (i) we establish the foundation of adaptive subspace evolution for HDD; (ii) we propose AHSE, an algorithm that dynamically tailors subspace evolution to data-specific hierarchical characteristics; (iii) we provide rigorous theoretical analysis of ECOCs' mechanisms within the BLS context; (iv) we demonstrate AHSE's effectiveness across diverse high-dimensional scenarios. By bridging the gap between data structure and evolution strategy, AHSE offers a principled approach to high-dimensional pattern discovery that adapts rather than imposes.

## 2 PRELIMINARIES

### 2.1 BROAD LEARNING SYSTEM (BLS)

As a lightweight alternative to traditional DNNs, BLS (Chen & Liu, 2017) efficiently constructs neural networks in a lateral manner. The framework of BLS is shown in Fig. 5 of Appendix A.3. Denote the training input as $\mathbf{X} \in \mathbb{R}^{N \times M}$ and the output as $\mathbf{Y} \in \{0, 1\}^{N \times C}$, where $N$, $M$ and $C$ are the number of samples, features, and classes, respectively. BLS first generates $n$ feature nodes by

$$\mathbf{Z}_i = \phi(\mathbf{X}\mathbf{W}_{e_i} + \boldsymbol{\beta}_{e_i}), \quad i = 0, 1, \ldots, n-1, \tag{1}$$

where $\mathbf{W}_e$ and $\boldsymbol{\beta}_e$ are sparse weights and biases initialized via a normal distribution, and $\phi$ is a self-supervised sparsification function. All feature nodes are combined into $\mathbf{Z}^n = [\mathbf{Z}_0, \mathbf{Z}_1, \ldots, \mathbf{Z}_{n-1}]$. Subsequently, we obtain $m$ enhancement nodes by

$$\mathbf{H}_j = \xi\left(\mathbf{Z}^n\mathbf{W}_{h_j} + \boldsymbol{\beta}_{h_j}\right), \quad j = 0, 1, \ldots, m-1, \tag{2}$$

where $\mathbf{W}_h$ and $\boldsymbol{\beta}_h$ are orthogonal weights and biases generated by singular value decomposition, and $\xi$ is a nonlinear activation function. Similarly, we combine all enhancement nodes into $\mathbf{H}^m = [\mathbf{H}_0, \mathbf{H}_1, \ldots, \mathbf{H}_{m-1}]$. Next we calculate the output weight using Moore-Penrose pseudoinverse:

$$\mathbf{W}_o = \left([\mathbf{Z}^n, \mathbf{H}^m]^\top [\mathbf{Z}^n, \mathbf{H}^m] + \lambda\mathbf{I}\right)^{-1} [\mathbf{Z}^n, \mathbf{H}^m]^\top \mathbf{Y}, \tag{3}$$

where $\lambda$ is a regularization coefficient, and $\mathbf{I} \in \{0, 1\}^{(n+m) \times (n+m)}$ is an identity matrix.

### 2.2 ERROR-CORRECTING OUTPUT CODES (ECOCs)

ECOCs, initially introduced in (Dietterich & Bakiri, 1994), are used to decompose multiclass problems into multiple binary subproblems, improving models' robustness especially in the presence of noise interference (Allwein et al., 2000; Yu et al., 2024). Fig. 6 of Appendix A.4 (Gupta & Amin, 2022) illustrates two example codebooks for a classification problem with 5 classes. For every column a binary classifier is trained over the training data, where all training data from classes with entry +1 (resp. entry -1) forms the positive class (resp. the other class). It can be seen that different codebook lengths and encoding strategies can lead to significant differences in the decision boundary. Denote the ECOC codebook as $\Omega \in \{-1, 1\}^{C \times L}$, where $L$ is the number of binary columns. The target space will be encoded by $\ddot{\mathbf{Y}} = \mathbf{Y} \times \Omega$, where $\ddot{\mathbf{Y}} \in \{-1, 1\}^{N \times L}$ is the ECOC ground truth while $\mathbf{Y} \in \{0, 1\}^{N \times C}$ is the one-hot ground truth. Then the training objective function becomes $\min_{\theta} \mathcal{L}(f(\mathbf{X}; \theta), \ddot{\mathbf{Y}})$, where $f(\mathbf{X}; \theta)$ is a model with $\theta$ being its parameters. During inference, the predicted class will be $\arg\max_{c \in \{0, 1, \ldots, C-1\}} : f(\mathbf{x}; \theta) \cdot \Omega[c, :]$, where $\mathbf{x}$ is a test sample.

## 3 METHODOLOGY

The architecture of AHSE is depicted in Fig. 2. AHSE specifically addresses the subspace evolution problem by designing evolution strategies that adapt to the intrinsic data hierarchy. The SEED branch implements a forward evolution process with customizable rates, while the PATH branch handles non-monotonic evolution through parallel subspace exploration and fusion. Finally, the SPOT circuit dynamically balances the contributions of both branches based on validation performance, creating a closed-loop evolutionary system. We elaborate on each of them below.

### 3.1 SEED: SERIES SUBSPACE EVOLUTION USING CHOLESKY DECOMPOSITION-BASED INCREMENTAL BLS

In this subsection, we design SEED to efficiently evolve hierarchical subspaces in series and lay the foundation for the parallel branch. We first rearrange the feature space by prioritizing features with high importance and low redundancy so that the evolved subspaces are optimal or near-optimal for a given size. Denote the feature space as $\mathbf{X}^{N \times M}$, where $N$ and $M$ are the number of samples and features, respectively. We obtain the feature importance $\mathbf{FIM} \in [0, 1]^M$ via Random Forest (Breiman, 2001) and correlation magnitude $\mathbf{FCM} \in [0, 1]^{M \times M}$ via Spearman Correlation (Spearman, 1987),

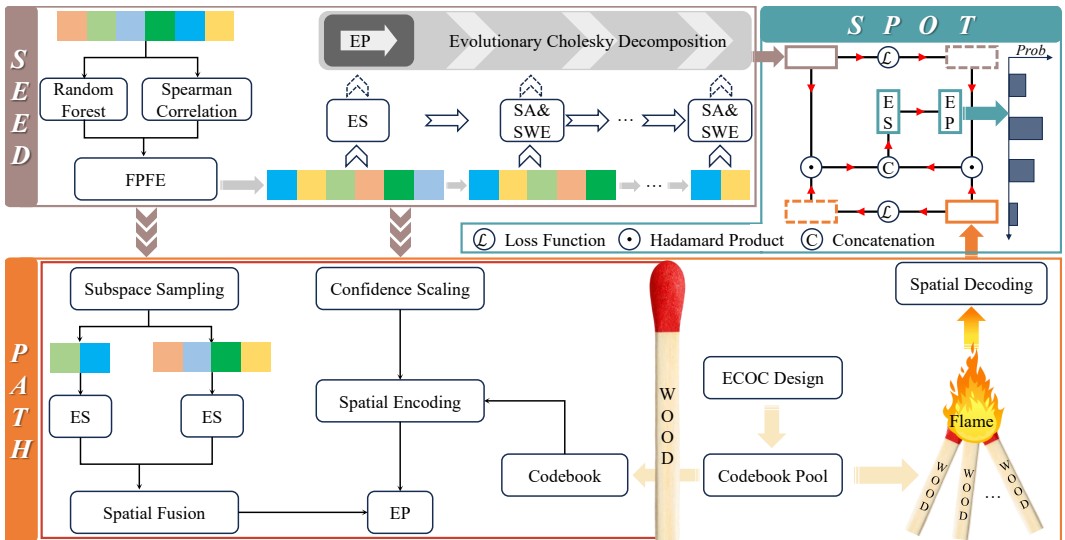

Figure 2: Illustration of AHSE. SEED's hierarchy-tailored subspace evolution adapts to the inherent structure of HDD, progressively refining underlying patterns. PATH's ECOC-based encoding and Flame optimization enhance robustness against noise in HDD. SPOT's dynamic fusion balances the efficiency of SEED with the robustness of PATH, obtaining the final outputs.

where $\mathbf{FCM}[i, j]$ is the correlation magnitude between features $i$ and $j$ and notably $\mathbf{FCM}[i, i] = 1$. Then we apply an automatic hierarchy identification method to automatically identify data hierarchy types and dynamically select evolution strategies (see Appendix A.15). Next, we employ feature priority forward evolution (FPFE) to prepare for the subsequent evolutions.

**FPFE** *Let $\boldsymbol{U} = \{i \in \mathbb{Z} \mid 0 \le i < M\}$ be the universal set of all feature indices. First, we initialize a queue of sorted indices $\boldsymbol{Q} = \emptyset$ and a set of unsorted indices $\boldsymbol{S} = \boldsymbol{U}$. The index of the most important feature, i.e. $\arg\max_r : \mathbf{FIM}[r]$, is pushed into $\boldsymbol{Q}$ and removed from $\boldsymbol{S}$. Then, for each unsorted index, we initialize the local correlation sum between it and all sorted indices by $\mathbf{LUM}[i] = \mathbf{FCM}[i, r], \forall i \in \boldsymbol{S}$. We calculate the local scores of unsorted indices by prioritizing features with high importance and low redundancy as follows:*

$$\mathbf{LS}[i] = \frac{\mathbf{FIM}[i]}{\mathbf{LUM}[i]}, \forall i \in \boldsymbol{S}. \tag{4}$$

*The index with the highest score, i.e. $\arg\max_s : \mathbf{LS}[s]$, is transferred from $\boldsymbol{S}$ to the end of $\boldsymbol{Q}$. The $\mathbf{LUM}$ of the remaining unsorted indices will be updated by $\mathbf{LUM}[i] = \mathbf{LUM}[i] + \mathbf{FCM}[i, s], \forall i \in \boldsymbol{S}$. Then we recalculate the local scores by Eq. (4) and continue to execute the aforementioned pipeline until $\boldsymbol{S}$ becomes empty. Algorithm 1 offers the pseudocode of FPFE.*

After FPFE is done, we prioritize the feature space by $\mathbf{X}_0 = \mathbf{X}[:, Q]$. To refine the efficiency of BLS for subspace evolution, we disassemble it into ensemble sparsification (ES) and enhanced perceptron (EP) as shown in Fig. 5. We use ES to robustly sparsify $\mathbf{X}_0$ (the lightblue region of Fig. 5):

$$\mathbf{Z}_0 = [\, \phi(\mathbf{X}_0 \mathbf{W}_{0,0}^{ES} + \boldsymbol{\beta}_{0,0}^{ES})\,,\, \phi(\mathbf{X}_0 \mathbf{W}_{0,1}^{ES} + \boldsymbol{\beta}_{0,1}^{ES})\,,\, \dots\,,\, \phi(\mathbf{X}_0 \mathbf{W}_{0,n-1}^{ES} + \boldsymbol{\beta}_{0,n-1}^{ES})\,], \tag{5}$$

where $\mathbf{W}^{ES}$ and $\boldsymbol{\beta}^{ES}$ are sparse weights and biases initialized via a normal distribution, $n$ is the number of sparse times, and $\phi$ is a self-supervised sparsification function. Then, we use EP to enhance the sparse representation and perceive the target space (the lightgreen region of Fig. 5).

**EP** *First, the sparse representation will be enhanced as follows:*

$$\mathbf{H}_0 = [\, \xi(\mathbf{Z}_0 \mathbf{W}_{0,0}^{E} + \boldsymbol{\beta}_{0,0}^{E})\,,\, \xi(\mathbf{Z}_0 \mathbf{W}_{0,1}^{E} + \boldsymbol{\beta}_{0,1}^{E})\,,\, \dots\,,\, \xi(\mathbf{Z}_0 \mathbf{W}_{0,m-1}^{E} + \boldsymbol{\beta}_{0,m-1}^{E})\,], \tag{6}$$

*where $\mathbf{W}^{E}$ and $\boldsymbol{\beta}^{E}$ are orthogonal weights and biases generated by singular value decomposition, $m$ is the number of orthogonal times, and $\xi$ is a nonlinear activation function. We concatenate the sparse*

and orthogonal representations and assign the training samples with equal weights simultaneously:

$$\mathbf{A}_0 = \sqrt{\mathbf{D}_0} \cdot [\, \mathbf{Z}_0 \,,\, \mathbf{H}_0 \,] \quad,\quad \mathbf{J}_0 = [\, \mathbf{Z}_0 \,,\, \mathbf{H}_0 \,]^\top \mathbf{D}_0 \quad, \tag{7}$$

where $\mathbf{A}_0$ is the weighted latent space, $\mathbf{J}_0$ is an offset term prepared for the subsequent substitution, $\mathbf{D}_0 \in [0,1]^{N \times N}$ denotes the initial sample weights, and $\mathbf{D}_0[i,j] = \begin{cases} \frac{1}{N}, & if\ i = j \\ 0, & if\ i \neq j \end{cases}$. Then the remaining problem is to solve the perception weight $\mathbf{W}_0^P$ such that (s.t.) $\mathbf{A}_0^\top \mathbf{A}_0 \mathbf{W}_0^P = \mathbf{J}_0 \mathbf{Y}$, where $\mathbf{Y}$ is the one-hot ground truth. We apply Cholesky decomposition (CD) as follows:

$$\mathbf{L}_0 = CD(\mathbf{A}_0^\top \mathbf{A}_0 + \lambda \mathbf{I}) \quad, \tag{8}$$

where $\mathbf{L}_0$ is the lower triangular Cholesky factor. Thereby the analytical solution of $\mathbf{W}_0^P$ can be obtained through one forward substitution (FS) and one backward substitution (BS) as follows:

$$\mathbf{W}_0^P = \mathbf{J}_0 \mathbf{Y}. FS(\mathbf{L}_0). BS(\mathbf{L}_0^\top) \tag{9}$$

We obtain the initial prediction via $\hat{\mathbf{Y}}_0 = [\, \mathbf{Z}_0 \,,\, \mathbf{H}_0 \,] \cdot \mathbf{W}_0^P$. Algorithm 2 offers EP's pseudocode.

After EP is done, we employ a subspace evolution mode tailored to diverse types of HDD hierarchies (see Section 4.3 for a detailed analysis). During the $t$-th evolution ($t > 0$), denote the current subspace as $\mathbf{X}_t$. To alleviate distribution drift, we replace ES with subspace alignment (SA), whose objective function is as follows:

$$\min_{\mathbf{W}_t^{SA}} \ \|\mathbf{X}_t - \mathbf{Z}_{t-1} \mathbf{W}_t^{SA}\|_2^2 + \lambda \|\mathbf{W}_t^{SA}\|_1, \tag{10}$$

where $\mathbf{W}_t^{SA} \in \mathbb{R}^{n \times M}$ is the alignment weight, the $l_2$-norm is used to align the sparse representations of the $(t-1)$-th evolution and the $t$-th evolution, and the $l_1$-norm is used to encourage sparsification. Then we obtain the $t$-th sparse representation by $\mathbf{Z}_t = \mathbf{X}_t \mathbf{W}_t^{SA^\top}$. Simultaneously, to gradually guide the model to focus on more and more difficult samples, we apply sample weight evolution (SWE) (Freund & Schapire, 1997) (Algorithm 3 offers the pseudocode of SWE) and obtain the updated sample weights $\mathbf{D}_t$. Next, we calculate $\mathbf{H}_t$, $\mathbf{A}_t$ and $\mathbf{J}_t$ via Eqs. (6) and (7). All latent spaces are combined column-wise by $\mathbf{A}^t = [\mathbf{A}_0 \,,\, \mathbf{A}_1 \,,\, \dots \,,\, \mathbf{A}_t]$, while all offset terms are combined row-wise by $\mathbf{J}^t = [\mathbf{J}_0^\top \,,\, \mathbf{J}_1^\top \,,\, \dots \,,\, \mathbf{J}_t^\top]^\top$. Thus, the remaining problem of the $t$-th evolution is to solve $\mathbf{W}_t^P$ s.t. $\mathbf{A}^{t\top} \mathbf{A}^t \mathbf{W}_t^P = \mathbf{J}^t \mathbf{Y}$. We use evolutionary Cholesky decomposition to achieve efficient incremental learning as follows:

$$\mathbf{P} = \mathbf{A}_t^\top \mathbf{A}_{t-1} \mathbf{L}_{t-1}^{-\top} \quad,\quad \mathbf{Q} = CD(\mathbf{A}_t^\top \mathbf{A}_t - \mathbf{P}\mathbf{P}^\top + \lambda \mathbf{I}) \quad,\quad \mathbf{L}_t = \begin{bmatrix} \mathbf{L}_{t-1} & 0 \\ \mathbf{P} & \mathbf{Q} \end{bmatrix} \quad. \tag{11}$$

Then we can obtain the analytical solution of $\mathbf{W}_t^P$ similar to Eq. (9). During inference, we obtain the predicted outputs of the $t$-th evolution by $\hat{\mathbf{Y}}_t = [\mathbf{Z}_0, \mathbf{H}_0, \mathbf{Z}_1, \mathbf{H}_1, \dots, \mathbf{Z}_t, \mathbf{H}_t] \cdot \mathbf{W}_t^P$.

To define the convergence criterion of SEED, we first calculate the loss decline owing to the $t$-th evolution as follows:

$$\Delta\mathcal{L}_t = \mathcal{L}(\,\hat{\mathbf{Y}}_{t-1}\,,\,\mathbf{Y}\,) - \mathcal{L}(\,\hat{\mathbf{Y}}_t\,,\,\mathbf{Y}\,) \quad, \tag{12}$$

where cross-entropy is adopted as the loss. Thus, the convergence criterion is set to $\frac{\Delta\mathcal{L}_t}{\mathcal{L}(\,\hat{\mathbf{Y}}_{t-1}\,,\,\mathbf{Y}\,)} < \epsilon$, where $\epsilon$ is the convergence threshold. Algorithm 4 offers the pseudocode of SEED.

## 3.2 PATH: Parallel Subspace Evolution Based on Post-Hoc ECOCs

In this subsection, we develop another branch termed PATH that evolves hierarchical subspaces in parallel on the foundation of SEED. Firstly, inspired by the theoretical analysis of ECOCs on DNNs (Yu et al., 2024), we rigorously derive the mechanism and robustness guarantee of ECOCs on BLS as follows.

**Theorem 1 (Bounded Output Perturbation).** Under assumptions of bounded inputs, weights, and Lipschitz-continuous activation functions, the output perturbation of BLS under weight noise with magnitude $\varepsilon$ is bounded by $\varepsilon \cdot C_1 + \varepsilon^2 \cdot C_2$, where $C_1$ and $C_2$ are constants determined by network architecture and activation properties.

**Corollary 1 (ECOC Robustness Condition).** BLS with ECOC encoding maintains correct classification under weight noise if:

$$\frac{dist\Big(\Omega\Big(f(\mathbf{x};\theta)\Big)\Big)}{2} > U\Big(f(\mathbf{x};\theta)\Big) + \frac{\varepsilon \cdot C_1 + \varepsilon^2 \cdot C_2}{\sqrt{L}} \quad , \tag{13}$$

where $dist(\cdot)$ is the minimum codeword distance and $U(\cdot)$ is the normalized uncertainty of predictions. The deduction are detailed in Appendix A.17. The robustness condition in Corollary 1 reveals three design principles for noise-resistant BLS systems: (i) maximize minimum codeword distances, (ii) minimize prediction uncertainty through proper calibration, and (iii) optimize network architecture parameters ($C_1$, $C_2$) for the expected noise regime. These principles directly shaped our PATH branch architecture. Specifically, we employ an ECOC design algorithm named Dense (Allwein et al., 2000) to generate a codebook pool, $\mathbb{P}_\Omega = \Big\{\Omega \in \{-1, 1\}^{C \times L}\Big\}$, which is used to train multiple subspace evolution bases called WOODs.

**WOOD** *Utilizing the feature importance* **FIM** *and correlation magnitude* **FCM** *from SEED, we calculate the global correlation sum* **GUM** *of each feature by* $\mathbf{GUM}[i] = \sum \mathbf{FCM}[i,:]-1, \forall i \in \boldsymbol{U}$. *Intuitively, features with higher importance and lower correlation play a more significant role and should be assigned higher values, so we calculate the global score* **GS** *as follows:*

$$\mathbf{GS}[i] = \frac{\exp^{\frac{\mathbf{FIM}[i]}{\mathbf{GUM}[i]}}}{\sum_{j=0}^{M-1} \exp^{\frac{\mathbf{FIM}[j]}{\mathbf{GUM}[j]}}} , \forall i \in \boldsymbol{U} . \tag{14}$$

*Denote the size of the final evolutionary subspace of SEED as* $\chi_\epsilon$. *We perform subspace sampling (SS) to construct a pair of complementary subspaces including a primary subspace and an auxiliary subspace as follows:*

$$\boldsymbol{S}_{pri} = Random\left(range = \boldsymbol{U} , \; size = \chi_\epsilon , \; probability = \mathbf{GS}\right) , \; \boldsymbol{S}_{aux} = \boldsymbol{U} \setminus \boldsymbol{S}_{pri} \quad ; \tag{15}$$

$$\mathbf{X}_{pri} = \mathbf{X}[:, \boldsymbol{S}_{pri}] \quad , \quad \mathbf{X}_{aux} = \mathbf{X}[:, \boldsymbol{S}_{aux}] \quad . \tag{16}$$

*Next, we sparsify them via Eq.* (5) *and obtain* $\mathbf{Z}_{pri}$ *and* $\mathbf{Z}_{aux}$, *which are fused as follows:*

$$\mathbf{Z}_{fus} = \mathbf{Z}_{pri} \cdot \sum \mathbf{GS}[\boldsymbol{S}_{pri}] + \mathbf{Z}_{aux} \cdot \sum \mathbf{GS}[\boldsymbol{S}_{aux}] . \tag{17}$$

*Simultaneously, in order to boost the samples that are intractable for SEED and play a complementary role, we perform confidence scaling (CS) on the training set as follows:*

$$\dot{\mathbf{Y}}[i,j] = \frac{\mathbf{Y}[i,j] \cdot \sum_{c=0}^{C-1} \exp^{\hat{\mathbf{Y}}_{t_\epsilon}^{SEED}[i,c]}}{\exp^{\hat{\mathbf{Y}}_{t_\epsilon}^{SEED}[i,j]}}, \forall i \in \{0, 1, \ldots, N-1\} \text{ and } j \in \{0, 1, \ldots, C-1\}, \tag{18}$$

*where* $\hat{\mathbf{Y}}_{t_\epsilon}^{SEED}$ *denotes the final outputs of SEED. Next, we randomly pull one codebook out of* $\mathbb{P}_\Omega$ *to encode the target space by* $\ddot{\mathbf{Y}} = \dot{\mathbf{Y}} \times \Omega$. *Lastly, the encoded target space together with the fused subspaces is processed by EP to predict* $\hat{\mathbf{Y}}$. *Algorithm 5 offers the pseudocode of WOOD.*

Notably, the uncertainties existing in each WOOD, including subspace sampling, codebook selection, and parameter generation of ES as well as EP, provide abundant varieties, making it a competent base for parallel subspace evolution. However, the ECOC-encoded target space inevitably suffers redundancy and may conflict with the latent space of BLS. Thus, we introduce a novel Adaptive ECOC Optimization Framework (AEOF) that dynamically optimizes codebook length and encoding strategies based on data characteristics and model performance (see Appendix A.16). Furthermore, we design a post-training evolutionary optimization mechanism called Flame that is executed on the validation set, to distill the target space by burning up the codevectors incompatible with the latent space of BLS.

**Flame** *During the t-th evolution ($t \geq 0$), we randomly pull one codebook $\Omega_t$ from $\mathbb{P}_\Omega$ to train a new WOOD and obtain its outputs $\hat{\tilde{\mathbf{Y}}}_t$. We concatenate $\Omega_t$ with previously optimized codebooks via $\tilde{\Omega}^t = [\tilde{\Omega}_0, \ldots, \tilde{\Omega}_{t-1}, \Omega_t]$, where $\tilde{\Omega}$ denotes an optimized codebook. Likewise, we concatenate all WOODs' outputs by $\hat{\tilde{\mathbf{Y}}}^t = [\hat{\tilde{\mathbf{Y}}}_0, \hat{\tilde{\mathbf{Y}}}_1, \ldots, \hat{\tilde{\mathbf{Y}}}_t]$. Next, we initialize a set of unburnt indices by $\mathbf{S}_t = \{j \in \mathbb{Z} \mid tL \leq j < (t+1)L\}$. We calculate their temperatures $\tau$ as follows:*

$$\tau_j = acc_{val}(\hat{\tilde{\mathbf{Y}}}^t \times (\tilde{\Omega}^t \odot \mathbf{M}_j)^\top) - acc_{val}(\hat{\tilde{\mathbf{Y}}}^t \times \tilde{\Omega}^{t\top}), \quad \forall \, j \in \mathbf{S}_t \quad , \tag{19}$$

*where $\odot$ denotes the Hadamard product, $\mathbf{M}_j \in \{0,1\}^{C \times (t+1)L}$ is the mask for the j-th codevector and $\mathbf{M}_j[:,k] = \begin{cases} 0, & \text{if } k = j \\ 1, & \text{if } k \neq j \end{cases}$ , while $acc_{val}$ is the validation accuracy. We select the index with the highest temperature, i.e. $\arg\max_r : \tau_r$, and determine whether $\tau_r > 0$. If so, we burn up the r-th codevector by $\tilde{\Omega}^t[:,r] = 0$ and remove r from $\mathbf{S}_t$ . Afterwards, we use the updated $\tilde{\Omega}^t$ to recalculate the temperatures of the remaining unburnt indices via Eq. (19) and continue to burn up other incompatible codevectors until no codevector has a positive temperature or $\mathbf{S}_t$ becomes empty. Finally, we obtain an optimal or near-optimal target space for the t-th evolution via $\tilde{\Omega}_t = \tilde{\Omega}^t \setminus \tilde{\Omega}^{t-1}$. Algorithm 6 offers the pseudocode of Flame. Theorem 1 and Corollary 1 provide the theoretical foundation for the Flame optimization mechanism. By maximizing the minimum codeword distance, Flame directly optimizes the robustness margin against weight noise.*

PATH adopts a convergence criterion similar to that of SEED. Lastly, we decode the target space by $\hat{\mathbf{Y}}_{PATH}^{t_\epsilon} = \hat{\tilde{\mathbf{Y}}}^{t_\epsilon} \times \tilde{\Omega}^{t_\epsilon\top}$, where $t_\epsilon$ is the final timestamp. Algorithm 7 offers PATH's pseudocode.

### 3.3 SPOT: Series-Parallel Closed Circuit between SEED and PATH

In this subsection, we introduce a lightweight circuit called SPOT to seamlessly combine the advantages of SEED and PATH in terms of efficiency and robustness, forming a closed evolutionary loop. First, we calculate the validation losses of SEED and PATH, normalize and exchange them to serve as the concatenation weights of each other:

$$\omega_{SEED} = \frac{\mathcal{L}_{\text{val}}(\hat{\mathbf{Y}}_{PATH}^{t_\epsilon}, \mathbf{Y})}{\mathcal{L}_{\text{val}}(\hat{\mathbf{Y}}_{t_\epsilon}^{SEED}, \mathbf{Y}) + \mathcal{L}_{\text{val}}(\hat{\mathbf{Y}}_{PATH}^{t_\epsilon}, \mathbf{Y})} \quad , \quad \omega_{PATH} = 1 - \omega_{SEED} \quad . \tag{20}$$

Then we concatenate the weighted outputs of SEED and PATH as follows:

$$\mathbf{X}_\circledcirc = \left[ \hat{\mathbf{Y}}_{t_\epsilon}^{SEED} \odot \omega_{SEED} , \hat{\mathbf{Y}}_{PATH}^{t_\epsilon} \odot \omega_{PATH} \right] \quad , \tag{21}$$

where the weights are broadcast to the same dimension as the outputs to execute the Hadamard product. Then we use ES to robustly sparsify $\mathbf{X}_\circledcirc$ via Eq. (5) and obtain $\mathbf{Z}_\circledcirc$, which is followed by EP to enhance the sparse representation and perceive the final target space. Algorithm 8 offers the pseudocode of SPOT.

By this point, we have accomplished the overall design of AHSE (the hierarchy-tailored design of subspace evolution is detailed in Section 4.3).

## 4 Experiments

### 4.1 Experimental Setup

We conduct extensive experiments in various high-dimensional scenarios such as image pattern recognition and speech emotion recognition (SER), as well as few-shot learning (FSL). Due to space constraints, the results of SER and FSL are presented in Appendix A.8 and Appendix A.9, respectively. For image pattern recognition, we include small-scale (MNIST (LeCun et al., 1998), Fashion-MNIST (Xiao et al., 2017) and CIFAR10 (Krizhevsky et al., 2009)) and large-scale (CIFAR100, MiniImageNet and TinyImageNet (Russakovsky et al., 2015)) tasks. For MNIST and Fashion-MNIST, the input of all methods is raw images and data augmentation is avoided; for color image datasets, since BLS-based methods are fundamentally designed to process feature vectors while deep architectures such as

ResNet, VGG, and ViT, are specifically engineered for 2D image data, all DNNs utilize standard data augmentation while BLS-based methods use latent representations from pretrained ResNet (CIFAR10 and CIFAR100) or MoCo-v3 (Chen et al., 2020) (MiniImageNet and TinyImageNet). In Appendix A.11, we also provide results of MLP using latent representations to promote fairness. We split one-tenth of the training set to form a validation set and use it to search for the optimal hyperparameters. Appendix A.1 offers detailed information and hyperparameter settings of these datasets. Appendix A.2 introduces baseline architectures. Appendix A.10 includes additional baselines to provide a more comprehensive evaluation. The experiments are conducted on an Ubuntu 20.04 operating system equipped with Intel Xeon Gold 6226R CPU. The programming language is Python 3.8.10.

## 4.2 RESULTS AND ANALYSIS

**Small-scale image pattern recognition** In Table 1, AHSE achieves state-of-the-art (SOTA) accuracy on Fashion-MNIST and CIFAR10, significantly outperforming the best DNN (ResNet) and BLS variant (ConvBLS). While VGG leads on MNIST, AHSE maintains competitive accuracy with superior training efficiency and uses far fewer inference FLOPs, striking an excellent performance-efficiency balance where lightweight BLS and CFEBLS sacrifice accuracy, and classic DNNs incur prohibitive computational costs. We also report standard deviations (5 times with different random seeds) in Appendix A.14 to demonstrate the stability of AHSE.

Table 1: Testing accuracy (%), training time (s) and inference FLOPs (M) on small-scale tasks.

| Method | MNIST | | | Fashion-MNIST | | | CIFAR10 | | |
|---|---|---|---|---|---|---|---|---|---|
| | acc | time | FLOPs | acc | time | FLOPs | acc | time | FLOPs |
| MLP | 97.39 | 21468 | **1.34** | 84.23 | 24206 | **1.34** | 56.20 | 2006 | **3.68** |
| VGG | **99.49** | 6792 | 231.35 | 90.28 | 6401 | 231.35 | 90.41 | 13337 | 339.79 |
| ResNet | 98.96 | 16583 | 939.91 | 92.99 | 15968 | 939.91 | 92.49 | 114705 | 1164.53 |
| ViT | 98.48 | 30978 | 378.40 | 89.30 | 36984 | 378.40 | 77.91 | 83085 | 644.35 |
| BLS | 98.74 | **38** | 14.28 | 91.39 | **26** | 9.89 | 93.14 | 58 | 18.13 |
| CFEBLS | 98.83 | 64 | 24.14 | 87.13 | 47 | 16.55 | 93.15 | **38** | 15.36 |
| Stacked BLS | 99.12 | 159 | 42.83 | 91.53 | 111 | 29.68 | 92.73 | 205 | 54.38 |
| ConvBLS | 99.28 | 229 | - | 92.43 | 333 | - | 94.50 | 512 | - |
| AHSE (Ours) | 99.33 | 153 | 27.19 | **93.72** | 159 | 63.09 | **95.97** | 282 | 156.39 |

**Large-scale image pattern recognition** In Table 2, AHSE consistently delivers the highest accuracy across all datasets, greatly surpassing the second-best model (ConvBLS for CIFAR100, Stacked BLS for MiniImageNet and ResNet for TinyImageNet). Crucially, it achieves this without sacrificing practicality: the training time and inference FLOPs remain acceptable. This establishes AHSE as an ideal solution for complex large-scale tasks, eliminating the traditional trade-off between high accuracy and computational feasibility. Notably, several DNNs such as MLP and ViT perform poorly on these datasets since they are trained from scratch. We also provide memory footprint and inference time measurements in Appendix A.13.

**Speech emotion recognition** To verify the versatility of AHSE, we conduct extensive experiments on five benchmark SER corpora. The detailed information of datasets is shown in Table 6 of Appendix A.8. From Table 7, we can observe that AHSE performs well on all five SER datasets. In particular, AHSE outperforms SOTA methods on CREMA-D, EMOVO, SAVEE, RAVDESS, and also ranks top-2 on IEMOCAP. This indicates that AHSE is not limited to image data but also excels in other HDD modalities, showcasing its versatility.

**Few-shot learning** Table 8 of Appendix A.9 shows five real-world high-dimensional datasets. It can be seen that these datasets contain a large number of features but the number of samples is relatively small. As can be seen from Table 9, AHSE outperforms SOTA methods by a large margin on all five datasets. This indicates that AHSE possesses the capability of FSL, and its performance is even more pronounced in extremely high-dimensional scenarios.

Table 2: Testing accuracy (%), training time (s) and inference FLOPs (M) on large-scale tasks.

| Method | CIFAR100 | | | MiniImageNet | | | TinyImageNet | | |
|--------|------|------|-------|------|------|------|------|------|------|
| | acc | time | FLOPs | acc | time | FLOPs | acc | time | FLOPs |
| MLP | 26.07 | 1846 | **3.73** | 25.57 | 16301 | 22.26 | 21.90 | 33483 | 13.22 |
| VGG | 71.07 | 24954 | 340.16 | 58.25 | 82375 | 2114.54 | 46.32 | 120242 | 1284.31 |
| ResNet | 72.78 | 133616 | 1164.58 | 65.49 | 367824 | 8165.46 | 57.20 | 416985 | 4658.2 |
| ViT | 49.42 | 116196 | 644.4 | 46.66 | 195721 | 3828.71 | 40.41 | 248380 | 2464.06 |
| BLS | 75.64 | 63 | 18.81 | 71.13 | **28** | **7.3** | 54.86 | **56** | **8.36** |
| CFEBLS | 75.36 | **39** | 16.04 | 71.07 | 36 | 12.47 | 54.82 | 73 | 13.53 |
| Stacked BLS | 76.33 | 193 | 56.42 | 71.50 | 114 | 21.9 | 56.38 | 225 | 25.08 |
| ConvBLS | 78.34 | 515 | - | - | - | - | - | - | - |
| AHSE (Ours) | **79.44** | 455 | 168.73 | **74.59** | 151 | 48.98 | **58.42** | 280 | 62.07 |

## 4.3 SUBSPACE EVOLUTION MODE TAILORED TO HDD HIERARCHY

To align the learning route of SEED with various HDD hierarchies, we adopt tailored subspace evolution modes in Figs. 1 and 3. We observe from Figs. 1a and 1b and 3a that the hierarchies of MNIST, Fashion-MNIST, and CIFAR100 differ from each other. Hence, we perform tailored evolution modes on them respectively. Specifically, for MNIST, $\chi_t = \lfloor \chi_0 \cdot \gamma^t \rfloor$, where $\chi_t$ is the size of the $t$-th evolutionary subspace; for Fashion-MNIST, $\chi_t = \lfloor \chi_0 \cdot \frac{1}{2}(1 + \cos \frac{t\pi}{T}) \rfloor$; for CIFAR100, $\chi_t = \lfloor \chi_0 \cdot (1 - \alpha \times t) \rfloor$. Note that when $\gamma = 1$ or $T = \infty$ or $\alpha = 0$, we fix the sizes of evolution subspaces and remove SA. Then we obtain the $t$-th evolutionary subspace by $\mathbf{X}_t = \mathbf{X}_0[:, : \chi_t]$. Since FPFE has been performed, it ensures that the evolved subspaces are optimal or near-optimal for a given size. From the sensitivity analysis in Figs. 1c and 1d and 3b, we observe that compared with the relatively drastic evolution of MNIST, the gentle evolution of Fashion-MNIST and CIFAR100 takes longer to converge. Meanwhile, a faster evolution rate generally means a lower risk of overfitting, yet it is accompanied by a tighter bottleneck. Additionally, hierarchy-tailored evolution significantly defeats fixed evolution, strongly stressing the significance of hierarchy-evolution consistency.

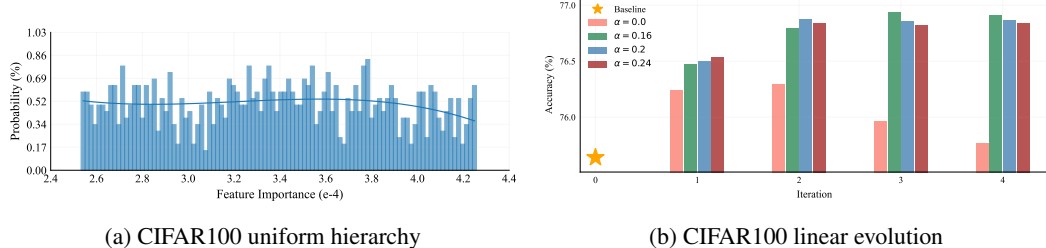

(a) CIFAR100 uniform hierarchy      (b) CIFAR100 linear evolution

Figure 3: Linear evolution tailored to CIFAR100's uniform hierarchy, where $\alpha$ is the evolution factor.

## 4.4 ROBUSTNESS OF ECOC-ENCODED TARGET SPACE

One-hot encoding is vulnerable to noise interference especially in the case of HDD (Yu et al., 2024), thus necessitating a sufficiently robust target space and making the application of ECOCs well-justified. Moreover, our theoretical analysis (Corollary 1) predicts that larger minimum codeword distances should yield greater robustness against weight noise. To verify this, we conducted controlled experiments on CIFAR10 and CIFAR100 with Gaussian weight noise of varying magnitudes. Fig. 4 confirms our prediction: ECOC codebooks with larger minimum distances maintain significantly higher accuracy under increasing noise levels, with performance degradation following precisely the quadratic bound established in Theorem 1. In addition, the robustness of ECOCs outperforms that of one-hot encoding in both small-scale (CIFAR10) and large-scale (CIFAR100) tasks, and the latter is more remarkable. We attach another case on MNIST in Appendix A.5. In Appendices A.18 and A.19, we further demonstrate the robustness of AHSE against adversarial perturbations and input noise, respectively.

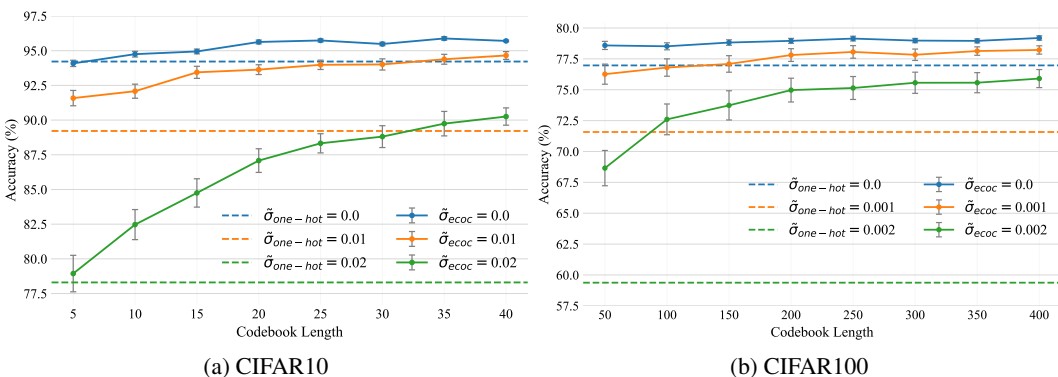

(a) CIFAR10             (b) CIFAR100

Figure 4: Robustness test against weight noise with zero mean and $\tilde{\sigma}^2$ variance.

## 4.5 ABLATION STUDY

In Table 3, we investigate the respective performances of the dual branches and the contributions of their internal components. We observe that when used alone, neither SEED nor PATH can catch up with AHSE, highlighting the role of SPOT. In addition, PATH suffers a considerable decline without the evolutionary information provided by SEED, which stresses the necessity of their interaction. Lastly, all components within SEED and PATH make contributions to different degrees. Among them, FPFE dominates SEED, while Flame dominates PATH. This demonstrates the effectiveness of evolutionary prioritization for the feature space as well as evolutionary optimization for the target space. In Appendices A.6 and A.7, we provide detailed studies on SA and Flame, respectively.

Table 3: A comprehensive ablation study. The evaluation metric is testing accuracy (%). Since both ES and EP are indispensable to BLS, and the results of replacing ECOCs with one-hot encoding are analyzed in Section 4.4 and Appendix A.5, we do not conduct ablation studies on them here.

| Dataset | SEED | | | | PATH | | | | | AHSE |
|---|---|---|---|---|---|---|---|---|---|---|
| | w/o FPFE | w/o SA | w/o SWE | Full | w/o SEED | w/o SS | w/o CS | w/o Flame | Full | Full |
| MNIST | 98.85 | 98.84 | 98.87 | 98.91 | 98.99 | 99.03 | 99.08 | 99.02 | 99.12 | **99.33** |
| Fashion-MNIST | 91.89 | 92.02 | 92.00 | 92.22 | 92.67 | 92.78 | 93.06 | 92.59 | 93.19 | **93.72** |
| CIFAR10 | 93.57 | 93.62 | 93.66 | 93.78 | 95.07 | 95.24 | 95.45 | 94.90 | 95.63 | **95.97** |
| CIFAR100 | 76.21 | 76.52 | 76.64 | 76.93 | 78.10 | 78.42 | 78.74 | 77.91 | 78.96 | **79.44** |
| MiniImageNet | 71.68 | 72.16 | 72.03 | 72.68 | 73.12 | 73.58 | 73.87 | 73.23 | 74.02 | **74.59** |
| TinyImageNet | 55.65 | 56.27 | 55.98 | 56.73 | 56.85 | 57.22 | 57.41 | 56.82 | 57.66 | **58.42** |

## 5 CONCLUSION AND DISCUSSION

In this paper, we propose an adaptive high-dimensional subspace evolution algorithm termed AHSE for tailored solutions to diverse hierarchical structures of HDD. The dual-branch collaborative architecture of AHSE not only absorbs the efficiency of BLS but also refines the robustness of ECOCs. Extensive experiments fully demonstrate the effectiveness of AHSE in various high-dimensional scenarios, while the attached theoretical analysis rigorously proves the mechanism and robustness guarantee of ECOCs on BLS, forming a closed loop between theory and practice. While AHSE is currently implemented within the BLS framework, the core principle of hierarchy-tailored subspace evolution has broader applicability. Models with incremental learning capabilities could directly benefit from our approach. For traditional deep networks, ensemble-based adaptations can incorporate our evolution strategies (shown in Appendix A.12), though at increased computational cost. Future work will explore efficient implementations across diverse model families.

## REPRODUCIBILITY STATEMENT

We provide all the necessary details, including data preparation, experimental equipment, programming language (see Section 4.1), dataset details, hyperparameter settings (see Appendix A.1), baseline architecture (see Appendix A.2), and exhaustive pseudocodes (see Appendix A.20) to ensure the reproducibility of our results. The code will be made publicly available upon acceptance.

## USAGE STATEMENT FOR LARGE LANGUAGE MODELS (LLMs)

We acknowledge the use of LLMs in paper writing to polish the language and improve the clarity of our expressions. LLMs were not involved in the generation of research ideas, conceptualization, research design, or experimental design. The authors take full responsibility for all content presented in this manuscript, including any content generated or polished with the assistance of LLMs.

## ETHICS STATEMENT

The authors have considered the ethical implications of this work. Our research is built upon publicly available datasets and models, and we do not anticipate any direct negative societal impacts.

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

## A APPENDIX

### A.1 DATASET DETAILS AND HYPERPARAMETER SETTINGS

Table 4: Summary of Datasets

| Dataset | Image shape | # Categories | # Training samples | # Validation samples | # Test samples |
|---|---|---|---|---|---|
| MNIST | (28,28,1) | 10 | 54000 | 6000 | 10000 |
| Fashion-MNIST | (28,28,1) | 10 | 54000 | 6000 | 10000 |
| CIFAR10 | (32,32,3) | 10 | 45000 | 5000 | 10000 |
| CIFAR100 | (32,32,3) | 100 | 45000 | 5000 | 10000 |
| MiniImageNet | (84,84,3) | 100 | 45000 | 5000 | 10000 |
| TinyImageNet | (64,64,3) | 200 | 100000 | 10000 | 10000 |

Table 5: Summary of Hyperparameter Settings

| Dataset | SEED | | PATH | | SPOT | |
|---|---|---|---|---|---|---|
| | $n^\dagger$ | $m$ | $n^\dagger$ | $m$ | $n^\dagger$ | $m$ |
| MNIST | 30×40 | 11000 | 10×10 | 11000 | 20×10 | 30 |
| Fashion-MNIST | 100×10 | 9000 | 100×10 | 11000 | 30×40 | 770 |
| CIFAR10 | 128×20 | 5000 | 128×20 | 10000 | 20×25 | 22 |
| CIFAR100 | 128×20 | 5000 | 128×20 | 10000 | 10×10 | 10 |
| MiniImageNet | 30×20 | 10000 | 30×20 | 10000 | 10×10 | 10 |
| TinyImageNet | 30×20 | 10000 | 30×20 | 10000 | 20×25 | 90 |

$\dagger$ The setting of $n$ consists of two factors: the sparse dimension × the number of sparse times.

## A.2 BASELINE ARCHITECTURE

The model architecture of MLP is input dimension-1024-512-number of classes, where the input dimension is 784 for MNIST and Fashion-MNIST, 3072 for CIFAR10 and CIFAR100, 21168 for MiniImageNet, and 12288 for TinyImageNet. The version of VGG used for comparison is VGG-16, while that of ResNet is ResNet-34. The number of encoders of ViT is 12, the embed_dim is 512, and the mlp_ratio is 4. The depth of Stacked BLS is fixed at 3.

## A.3 FRAMEWORK OF BROAD LEARNING SYSTEM (BLS)

Fig. 5 illustrates the framework of Broad Learning System (BLS). The input data is first mapped to feature nodes and then enhanced to enhancement nodes. Finally, the output weights are calculated by ridge regression. In AHSE, the feature mapping layer is transformed into ensemble sparsification (ES), and the enhancement layer with output layer is transformed into enhanced perceptron (EP).

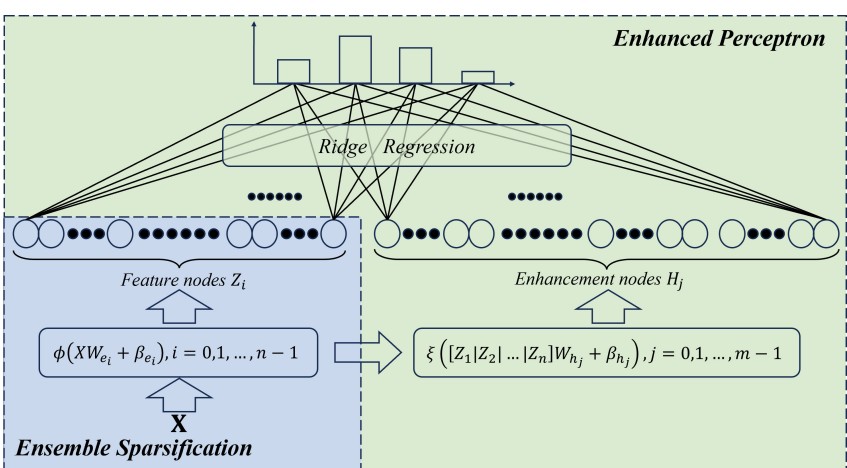

Figure 5: Framework of Broad Learning System (BLS). The vanilla BLS consists of three parts: feature mapping layer, enhancement layer, and output layer. In AHSE, we disassemble BLS into ES (the lightblue part) and EP (the lightgreen part).

## A.4 ILLUSTRATION OF ERROR-CORRECTING OUTPUT CODES (ECOCS)

Fig. 6 (Gupta & Amin, 2022) illustrates two example codebooks for a classification problem with 5 classes. For every column a binary classifier is trained over the training data, where all training data from classes with entry +1 (resp. entry -1) forms the positive class (resp. the other class). It can be seen that different codebook lengths and encoding strategies can lead to significant differences in the decision boundary.

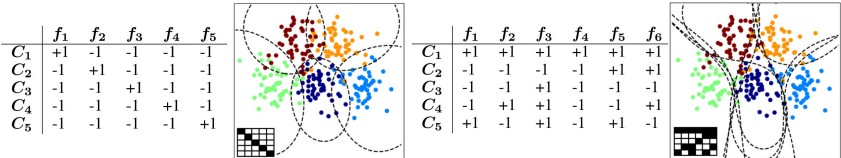

Figure 6: Illustration of Error-Correcting Output Codes (ECOCs).

## A.5   AN ADDITIONAL CASE OF **ROBUSTNESS OF ECOC-ENCODED TARGET SPACE** ON MNIST

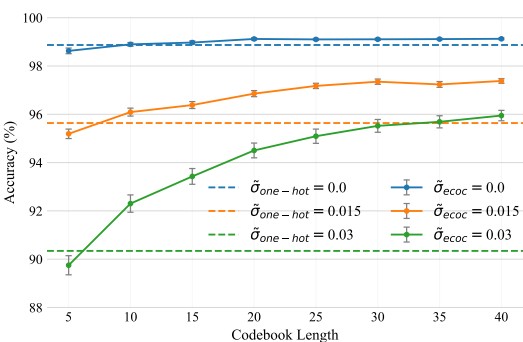

Figure 7: Robustness test against weight noise with zero mean and $\tilde{\sigma}^2$ variance on MNIST.

Fig. 7 depicts the robustness test against weight noise on MNIST. We observe that on MNIST, the performance degradation of one-hot encoding induced by weight-noise interference is less pronounced compared to that on CIFAR10 and CIFAR100. This is probably because MNIST presents a relatively simple task and fewer categories as well.

## A.6   DRIFT RELIEF OF EVOLUTIONARY SUBSPACES

During subspace evolution, distribution drift may occur and exacerbate performance loss and overfitting risk. Hence, we put forward SA to address this issue. Comparing Figs. 8a and 8b, we find that the distribution of evolutionary subspaces without SA is in a scattered state, and there seems to be an abnormal oscillation during the last evolution. After applying SA, the evolutions proceed roughly along the same route, which is conducive to aligning the learning route of SEED with the HDD hierarchy and achieving stable evolutionary computation.

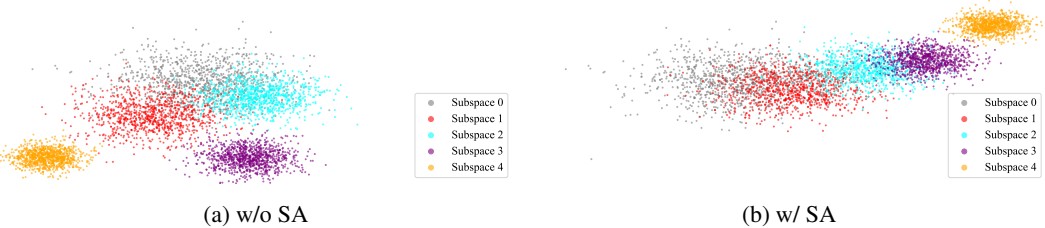

(a) w/o SA                                                                  (b) w/ SA

Figure 8: Subspace evolution routes of SEED on CIFAR100 without and with SA applied.

## A.7   EVOLUTIONARY OPTIMIZATION EFFECT OF FLAME

In order to quantify the evolutionary optimization effect of Flame, we propose a customized metric named latent-target space compatibility (LTSC) as follows:

$$LTSC\Big(f(\mathbf{X};\theta),\Omega\Big|\mathbf{Y}\Big) = \frac{C}{N}\sum_{i=0}^{N-1}\sum_{c=0}^{C-1}\frac{\mathbb{I}(\mathbf{Y}[i,c]=1)\cdot f(\mathbf{X}[i,:];\theta)\cdot\Omega[c,:]}{\|\Omega\|_0} \quad , \qquad (22)$$

where $\mathbb{I}(\cdot)$ is the indicator function. Fig. 9 illustrates the evolutionary iterations of PATH on CIFAR10 before and after the introduction of Flame. Clearly, after introducing Flame, the overall accuracy, stability, and LTSC witness a substantial boost, and the gap compared to before is also steadily widening. This sufficiently reflects the effectiveness of Flame in terms of promoting the compatibility between the latent space of PATH and the ECOC-encoded target space. On the other hand, we observe that LTSC generally exhibits a positive correlation with accuracy, except for a slight fluctuation in the second iteration. This reveals the potential of LTSC as an in-depth metric for evaluating the quality of the target space.

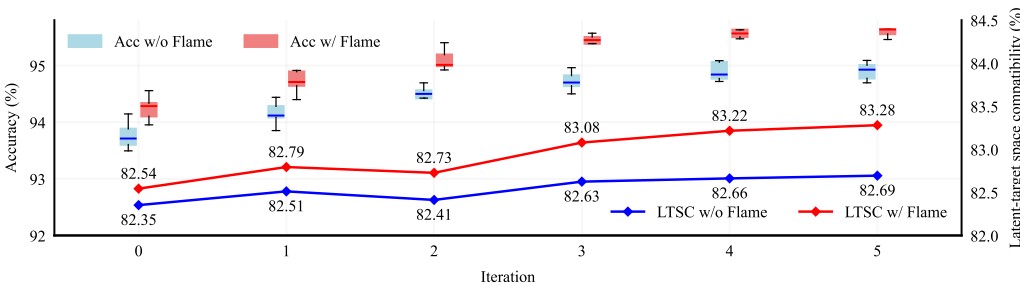

Figure 9: Evolutionary iterations of PATH on CIFAR10 before and after the introduction of Flame.

## A.8 VERSATILITY OF AHSE ON SPEECH EMOTION RECOGNITION

To verify the versatility of AHSE, we conduct extensive experiments on five benchmark SER corpora. The detailed information is shown in Table 6. Notably, to speed up training and inference, we unify the lengths of all samples to the median of the dataset. The input of AHSE is latent representations from Wav2vec 2.0 Base (Baevski et al., 2020).

Table 6: Summary of Datasets

| Dataset | Language | # Samples | # Emotions | Median duration |
|---|---|---|---|---|
| CREMA-D (Cao et al., 2014) | English | 7442 | 6 | 2.5s |
| EMOVO (Costantini et al., 2014) | Italian | 588 | 7 | 3s |
| SAVEE (Jackson & Haq, 2014) | English | 480 | 7 | 4s |
| RAVDESS (Livingstone & Russo, 2018) | English | 1440 | 8 | 4s |
| IEMOCAP (Busso et al., 2008) | English | 5531 | 4 | 3.5s |

Table 7: Comparison with SOTA methods on five speech emotion recognition datasets. The evaluation metric is accuracy (%). The best results are highlighted in **bold**.

| Method | CREMA-D | EMOVO | SAVEE | RAVDESS | IEMOCAP |
|---|---|---|---|---|---|
| MDC-LSTM (Zhang et al., 2019) | 52.44 | 60.92 | 63.85 | 65.9 | 58.2 |
| AST (Gong et al., 2021) | 66.58 | 72.35 | 72.6 | 78.67 | 63.61 |
| TIM-Net (Ye et al., 2023) | 64.05 | 81.51 | 82.92 | 85.73 | 65.27 |
| MS-SENet (Li et al., 2024) | 66.67 | 82.27 | 82.08 | 86.11 | 66.41 |
| Wav2vec 2.0 Base (Baevski et al., 2020) | 76.71 | 86.39 | 90.62 | 91.91 | 75.3 |
| WavLM Large (Chen et al., 2022) | 80.29 | 86.72 | 92.19 | 90.48 | 78.51 |
| HuBERT Large (Hsu et al., 2021) | 79.0 | 86.55 | 91.94 | 91.13 | 77.57 |
| DWFormer (Chen et al., 2023) | 77.93 | 83.19 | 91.67 | 90.99 | 73.19 |
| Whisper Medium (Goron et al., 2024) | 80.76 | 87.18 | 92.19 | 91.72 | **81.02** |
| ShiftCNN (Shen et al., 2023) | 77.78 | 87.98 | 92.29 | 93.28 | 76.44 |
| AHSE (ours) | **81.53** | **93.27** | **96.49** | **94.85** | 80.18 |

From Table 7, we can observe that AHSE performs well on all five SER datasets. In particular, AHSE outperforms SOTA methods on CREMA-D, EMOVO, SAVEE, RAVDESS, and also ranks top-2 on IEMOCAP. This indicates that AHSE is not limited to image data but also excels in other HDD modalities, showcasing its versatility.

## A.9 Versatility of AHSE on Few-shot Learning

Table 8 shows 5 real-world high-dimensional datasets. These datasets are selected from UCI Machine Learning Repository (Asuncion et al., 2007). It can be seen that these datasets contain a large number of features but the number of samples is relatively small.

Table 8: Dataset Statistics

| Dataset | # Samples | # Features |
|---|---|---|
| ALL (Asuncion et al., 2007) | 72 | 7129 |
| Carcinom (Asuncion et al., 2007) | 174 | 9182 |
| CLL_sub (Asuncion et al., 2007) | 111 | 11340 |
| Glioma (Asuncion et al., 2007) | 50 | 4434 |
| Tox (Asuncion et al., 2007) | 171 | 5748 |

Table 9: Comparison with SOTA methods on five few-shot high-dimensional datasets. The evaluation metric is accuracy (%). The best results are highlighted in **bold**.

| Method | ALL | Carcinom | Cll_sub | Glioma | Tox |
|---|---|---|---|---|---|
| RandomForest (Breiman, 2001) | 90.75 | 82.91 | 67.17 | 69.20 | 70.53 |
| ExtraTrees (Geurts et al., 2006) | 91.32 | 81.71 | 66.96 | 70.80 | 71.58 |
| Random Subspace (Ho, 1998) | 91.10 | 83.64 | 70.63 | 68.80 | 72.38 |
| RotationForest (Rodriguez et al., 2006) | 94.14 | 85.87 | 75.66 | 72.80 | 82.45 |
| Bagging (Breiman, 1996) | 92.25 | 83.31 | 72.42 | 67.20 | 72.99 |
| AdaBoost (Freund et al., 1996) | 93.64 | 45.08 | 64.06 | 54.00 | 40.48 |
| GBDT (Friedman, 2001) | 86.07 | 81.26 | 70.80 | 61.20 | 65.25 |
| XGBoost (Chen & Guestrin, 2016) | 91.46 | 81.17 | 72.42 | 69.20 | 72.97 |
| RotBoost (Zhang & Zhang, 2008) | 95.78 | 52.54 | 74.75 | 54.00 | 60.69 |
| RoRF (Stiglic et al., 2011) | 91.96 | 85.88 | 72.03 | 71.60 | 78.36 |
| ESRF (Amasyali & Ersoy, 2013) | 90.92 | 80.17 | 63.78 | 66.40 | 67.84 |
| RSM-IPCS (Ferreira et al., 2009) | 88.32 | 86.22 | 67.92 | 69.60 | 77.55 |
| NEC (Zhang & Zhang, 2009) | 94.89 | 83.42 | 77.32 | 68.00 | 72.84 |
| CESE (Xu et al., 2023) | 97.57 | 92.48 | 76.54 | 80.40 | 86.56 |
| AHSE (ours) | **98.49** | **95.11** | **80.43** | **83.87** | **88.74** |

As can be seen from Table 9, AHSE outperforms SOTA methods by a large margin on all five datasets. This indicates that AHSE possesses the capability of FSL, and its performance is even more pronounced in extremely high-dimensional scenarios.

## A.10 Comparison with Additional Baselines

In this subsection, we compare the performance of AHSE with additional baselines in recent years, including ConvNeXt (Liu et al., 2022), EfficientNet (Tan & Le, 2019), Swin (Liu et al., 2021) and ViT-L (Dosovitskiy et al., 2020). From Table 10, we made the following observations: (i) AHSE demonstrates clear advantages on datasets with complex hierarchical structures, particularly CIFAR10 (+2.5%), CIFAR100 (+5.18%), and MiniImageNet (+6.2%) compared to the best modern baselines. This validates our core hypothesis that adaptive evolution strategies provide significant benefits for data with intricate hierarchical patterns. (ii) On MNIST, ConvNeXt achieves marginally better accuracy (99.38% vs 99.33%), which is expected given MNIST's relatively simple structure where static evolution strategies can perform well. On TinyImageNet, EfficientNet shows better performance (60.47% vs 58.42%), likely due to its specialized architecture for large-scale image recognition. (iii) AHSE maintains clear advantages on FashionMNIST (+0.24%) and significantly outperforms all modern architectures on CIFAR10 and CIFAR100, highlighting its effectiveness for mid-scale image recognition tasks with diverse hierarchical structures.

Table 10: Performance Comparison with Additional Baselines Across Datasets. The metric is accuracy (%) and the best results are highlighted in **bold**.

| Method | MNIST | Fashion-MNIST | CIFAR10 | CIFAR100 | MiniImageNet | TinyImageNet |
|--------|-------|---------------|---------|----------|--------------|--------------|
| ConvNeXt | **99.38** | 92.42 | 93.47 | 74.26 | 68.39 | 58.36 |
| EfficientNet | 99.13 | 93.48 | 92.71 | 73.19 | 66.27 | **60.47** |
| Swin | 98.24 | 91.94 | 79.29 | 57.93 | 49.63 | 45.51 |
| ViT-L | 98.90 | 92.73 | 80.24 | 53.10 | 52.74 | 49.60 |
| AHSE | 99.33 | **93.72** | **95.97** | **79.44** | **74.59** | 58.42 |

## A.11 RESULTS OF MLP USING LATENT REPRESENTATIONS

In this subsection, we use MLP as a representative deep model since the vanilla ResNet, ViT and VGG themselves do not support the use of pretrained representations. Specifically, we used the same pre-trained feature extractors (ResNet for CIFAR10 and CIFAR100, MoCo-v3 for MiniImageNet and TinyImageNet) and compared performance on these shared features. The results of this fair comparison are shown in Table 11.

Table 11: Performance comparison using Latent Representations. The metric is accuracy (%) and the best results are highlighted in **bold**.

| Method | CIFAR10 | CIFAR100 | MiniImageNet | TinyImageNet |
|--------|---------|----------|--------------|--------------|
| MLP | 93.56 | 75.21 | 70.94 | 54.61 |
| BLS | 93.14 | 75.64 | 71.13 | 54.86 |
| CFEBLS | 93.15 | 75.36 | 71.07 | 54.82 |
| Stacked BLS | 92.73 | 76.33 | 71.50 | 56.38 |
| ConvBLS | 94.50 | 78.34 | - | - |
| AHSE | **95.97** | **79.44** | **74.59** | **58.42** |

The advantage of AHSE over MLP with the same features further validates our approach's superiority in high-dimensional subspace processing. This indicates that our adaptive subspace evolution mechanism provides genuine benefits beyond any advantages from feature extraction.

## A.12 DEEP ENSEMBLE ADAPTATIONS

The core principle of adaptive subspace evolution—tailoring the evolution strategy to match data hierarchy—can indeed be extended to deep ensembles. We have conducted preliminary experiments implementing our approach in this context. Specifically, we designed an ensemble framework where each new base learner (MLP) focuses on features selected through our hierarchy-tailored evolution process and difficult samples identified through our Sample Weight Evolution (SWE) mechanism. The key implementation details are as follows: (i) Each MLP in the ensemble processes a different evolved subspace; (ii) SWE dynamically adjusts sample weights between ensemble members; (iii) SPOT-inspired fusion combines ensemble predictions with weights determined by validation performance. The results in Table 12 demonstrate that our adaptive evolution principle can enhance performance even in deep ensemble contexts. While the computational overhead is higher than in BLS implementations, the accuracy improvements validate the broader applicability of our approach.

Table 12: Performance Comparison of Ensemble Methods Across Multiple Datasets. MLP* denotes Voting Ensemble (5 MLPs). MLP** denotes Adaptive Evolution Ensemble.

| Method | MNIST | Fashion-MNIST | CIFAR10 | CIFAR100 | MiniImageNet | TinyImageNet |
|--------|-------|---------------|---------|----------|--------------|--------------|
| MLP | 97.39 | 84.23 | 56.20 | 26.07 | 25.57 | 21.90 |
| MLP* | 98.02 | **86.52** | 57.12 | 27.48 | 26.53 | 23.74 |
| MLP** | **98.43** | 86.14 | **58.04** | **29.13** | **28.21** | **25.05** |

### A.13 MEMORY FOOTPRINT AND INFERENCE TIME MEASUREMENTS

We measured both memory footprint and wall-clock inference time on a standardized hardware platform with the following specifications: CPU: AMD EPYC 75F3 32-Core Processor (2.95 GHz); GPU: NVIDIA GeForce RTX 3090 (24GB VRAM); RAM: 503 GB DDR4; OS: Ubuntu 20.04 LTS; Software: Python 3.8.18, PyTorch 2.1.2, CUDA 12.2. The results are shown in Table 13 and Table 14.

Table 13: Memory footprint (in MB) comparison on various datasets.

| Method | MNIST | Fashion-MNIST | CIFAR10 | CIFAR100 | MiniImageNet | TinyImageNet |
|---|---|---|---|---|---|---|
| MLP | 5.1 | 5.1 | 14.04 | 14.21 | 84.9 | 50.41 |
| VGG | 152.35 | 152.35 | 152.35 | 153.76 | 153.76 | 155.32 |
| ResNet | 81.18 | 81.18 | 81.18 | 81.36 | 81.36 | 81.56 |
| ViT | 144.47 | 144.47 | 144.74 | 144.91 | 145.08 | 145.2 |
| BLS | 108.93 | 75.5 | 138.33 | 143.52 | 55.71 | 63.8 |
| CFEBLS | 184.21 | 126.3 | 117.24 | 122.43 | 95.18 | 103.27 |
| Stacked BLS | 326.78 | 226.5 | 414.99 | 430.56 | 167.13 | 191.4 |
| AHSE | 164.38 | 114.06 | 243.93 | 269.07 | 118.53 | 153.7 |

Table 14: Inference time (in seconds) comparison on various datasets.

| Method | MNIST | Fashion-MNIST | CIFAR10 | CIFAR100 | MiniImageNet | TinyImageNet |
|---|---|---|---|---|---|---|
| MLP | 0.96 | 0.98 | 1.12 | 1.48 | 2.72 | 2.95 |
| VGG | 1.63 | 1.82 | 2.16 | 2.25 | 4.24 | 3.83 |
| ResNet | 2.05 | 2.72 | 2.98 | 2.35 | 8.26 | 9.79 |
| ViT | 2.08 | 2.21 | 2.63 | 2.62 | 4.34 | 9.12 |
| BLS | 1.25 | 0.72 | 1.31 | 1.38 | 0.68 | 0.74 |
| CFEBLS | 1.46 | 1.03 | 1.39 | 1.12 | 0.92 | 0.95 |
| Stacked BLS | 3.67 | 2.07 | 3.87 | 3.92 | 1.77 | 1.85 |
| AHSE | 1.86 | 1.93 | 2.66 | 2.8 | 1.68 | 1.97 |

As shown in Table 13, AHSE maintains a moderate memory requirement across all datasets, with a memory footprint of 118.53 MB for MiniImageNet and 153.7 MB for TinyImageNet. On the other hand, the inference time measurements in Table 14 show that AHSE achieves a balanced efficiency profile with competitive inference times across all datasets.

### A.14 STANDARD DEVIATION OF PERFORMANCE

In this subsection, we present the standard deviations of the performance results reported in Tables 1 and 2. As shown in Table 15, AHSE demonstrates low standard deviation across all datasets, indicating its robustness and consistent performance. For instance, on MNIST, AHSE achieves a standard deviation of 0.04, while on TinyImageNet, it maintains a standard deviation of 0.22. This consistency across diverse datasets highlights AHSE's reliability in various scenarios.

### A.15 AUTOMATIC HIERARCHY IDENTIFICATION FRAMEWORK

In order to automatically identify data hierarchy types and dynamically select evolution strategies, we developed an automatic hierarchy identification mechanism that can determine the optimal evolution strategy without human intervention. The following describes our solution and experimental validation.

First, we introduce a lightweight Hierarchy Structure Analyzer (HSA) module that automatically identifies the intrinsic hierarchical structure of high-dimensional data. The HSA first extracts feature importance scores using a lightweight Random Forest model trained on a small subset of the data. From this distribution, we compute three key statistical metrics: Kurtosis, Autocorrelation at lag 1

Table 15: Standard deviation of performance across various datasets.

| Method | MNIST | Fashion-MNIST | CIFAR10 | CIFAR100 | MiniImageNet | TinyImageNet |
|---|---|---|---|---|---|---|
| MLP | 0.08 | 0.15 | 0.32 | 0.24 | 0.28 | 0.26 |
| VGG | 0.03 | 0.07 | 0.12 | 0.18 | 0.23 | 0.31 |
| ResNet | 0.05 | 0.06 | 0.08 | 0.15 | 0.20 | 0.25 |
| ViT | 0.06 | 0.11 | 0.43 | 0.57 | 0.68 | 0.72 |
| BLS | 0.07 | 0.09 | 0.08 | 0.14 | 0.16 | 0.21 |
| CFEBLS | 0.06 | 0.18 | 0.09 | 0.16 | 0.18 | 0.23 |
| Stacked BLS | 0.05 | 0.08 | 0.10 | 0.13 | 0.15 | 0.19 |
| AHSE | 0.04 | 0.05 | 0.07 | 0.13 | 0.17 | 0.22 |

and Entropy. Kurtosis measures the peakedness of the importance distribution as follows:

$$K = \frac{1}{N} \sum_i (\frac{FI_i - \mu_{FI}}{\sigma_{FI}})^4, \tag{23}$$

where $FI_i$ is feature importance, $\mu_{FI}$ is mean importance, and $\sigma_{FI}$ is standard deviation. Autocorrelation at lag 1 measures periodic patterns in sorted importance values as follows:

$$A = \frac{\sum_i (FI_{sorted}[i] - \mu_{FI})(FI_{sorted}[i+1] - \mu_{FI})}{\sum_i (FI_{sorted}[i] - \mu_{FI})^2}. \tag{24}$$

Entropy measures the uniformity of the importance distribution as follows:

$$H = -\sum_i p(FI_i) \cdot \log\left(p\left(FI_i\right)\right), \tag{25}$$

where $p(FI_i)$ is the normalized importance probability. Based on these metrics, we classify the hierarchy type using the following decision rules: (i) if K > 8.0 and H < 2.5, we classify it as dense hierarchy and perform exponential evolution; (ii) if A > 0.6 and 2.0 < K < 6.0, we classify it as periodic hierarchy and perform cosine evolution; (iii) otherwise, we classify it as uniform hierarchy and perform linear evolution. After identifying the hierarchy type, HSA automatically selects appropriate evolution parameters through a lightweight validation process: for dense hierarchy, search $\gamma$ between 0.2 and 0.7 with step 0.1; for periodic hierarchy, search $T$ between 3.0 and 6.0 with step 0.4; for uniform hierarchy, search $\alpha$ between 0.12 and 0.3 with step 0.04. The parameter yielding the highest validation accuracy after just 3 evolutionary steps is selected for full training. We conducted experiments to validate our automatic hierarchy identification approach and found HSA worked well on various datasets. However, while our automatic identification mechanism significantly improves AHSE's usability, it may struggle with hybrid hierarchy types that don't cleanly fit our three categories. Future work will explore deep clustering techniques to identify more complex hierarchy patterns and develop hierarchical evolution strategies that can adapt during training rather than just at initialization.

## A.16 ADAPTIVE ECOC OPTIMIZATION FRAMEWORK

In this subsection, we introduce a novel Adaptive ECOC Optimization Framework (AEOF) that dynamically optimizes codebook length and encoding strategies based on data characteristics and model performance. Instead of using fixed codebook lengths, we propose a gradient-based mechanism to determine optimal codebook length:

$$L^* = \arg\max_L : \ acc_{val}(L) - \alpha L, \tag{26}$$

where $acc_{val}(L)$ is the validation accuracy with codebook length L, and $\alpha$ is a complexity penalty factor. This balances performance gains against computational overhead. The optimization is performed efficiently using Bayesian optimization to minimize validation trials. Next, we introduce a differentiable ECOC framework where codebook entries can be optimized directly through gradient descent. By relaxing the binary constraint $\{-1, 1\}$ to continuous values with a tanh activation, we enable end-to-end optimization:

$$\Omega_{opt} = \Omega_{init} - \eta \nabla_\Omega \mathcal{L}(f(X; \theta, \Omega), Y), \tag{27}$$

where $\eta$ is the learning rate and $\mathcal{L}$ is the loss function. After optimization, we project back to binary values using sign function. This approach preserves theoretical robustness guarantees while adapting to dataset characteristics. Moreover, rather than using a fixed pool of codebooks, we introduce an evolutionary algorithm that dynamically generates and refines the codebook pool: (i) Initialize a diverse population of random codebooks; (ii) Evaluate fitness based on classification accuracy and minimum codeword distance; (iii) Apply genetic operations (selection, crossover, mutation) to evolve better-performing codebooks; (iv) Periodically inject new random codebooks to maintain diversity; (v) Retain top-performing codebooks for the next generation. The evolutionary process continues until convergence or a maximum number of generations is reached.

AEOF enhances our existing Flame optimization mechanism by providing higher-quality initial codebooks. The combined framework creates a two-stage optimization process: (i) Global optimization: AEOF discovers optimal codebook structure and length; (ii) Local refinement: Flame performs targeted pruning of incompatible codewords. This hierarchical approach leverages AEOF's global search capabilities and Flame's local fine-tuning to achieve superior codebook designs.

In Table 16, we conducted experiments to evaluate our adaptive ECOC framework on CIFAR10 and CIFAR100 datasets, comparing against our original fixed codebook approach. From the results, we observe that AEOF automatically discovers shorter yet more effective codebooks, reducing length by 20% on CIFAR10 and 14.5% on CIFAR100, but the performance improves by 0.38% and 0.73% on CIFAR10 and CIFAR100 respectively and robustness to weight noise improves by 0.55% and 0.72% respectively. Moreover, the training time decreases by around 15% due to optimized codebook length.

Table 16: Performance comparison of Fixed ECOC and Adaptive ECOC (AEOF) on CIFAR10 and CIFAR100. The metrics are accuracy (%) and robustness (%) under weight noise ($\sigma$=0.01 for CIFAR10, $\sigma$=0.001 for CIFAR100).

| Dataset | Method | Codebook Length | Accuracy | Robustness |
|---|---|---|---|---|
| CIFAR10 | Fixed ECOC (Original) | 20 | 95.97 | 93.63 |
| CIFAR10 | Adaptive ECOC (AEOF) | 16 | 96.35 | 94.18 |
| CIFAR100 | Fixed ECOC (Original) | 200 | 79.44 | 77.81 |
| CIFAR100 | Adaptive ECOC (AEOF) | 171 | 80.17 | 78.53 |

## A.17 MECHANISM AND ROBUSTNESS GUARANTEE OF ECOCs ON BLS

In (Yu et al., 2024), the principle and efficacy of ECOCs on DNNs have been vividly revealed. However, the theoretical results adopt the assumption of neural tangent kernel (NTK), which requires the width of the network to approach infinity. This assumption can be strong for some neural networks. In the following, instead, we aim to mathematically derive the mechanism and robustness guarantee of ECOCs on BLS under relatively mild assumptions.

### A.17.1 ASSUMPTIONS

**Assumption 1.** *For the input of BLS, we assume* $\|\frac{\mathbf{X}-\mu}{\sigma}\|_2 < \infty$*, where* $\mu$ *is the mean while* $\sigma$ *is the standard deviation. Without loss of generality, we assume that*

$$\|\frac{\mathbf{X}-\mu}{\sigma}\|_2 \leq B_x \quad . \tag{28}$$

**Assumption 2.** *The sparse weights and biases, the orthogonal weights and biases, and the output weight, are bounded, i.e.* $\|\mathbf{W}_e\|_2 < \infty$*,* $\|\boldsymbol{\beta}_e\|_2 < \infty$*,* $\|\mathbf{W}_h\|_2 < \infty$*,* $\|\boldsymbol{\beta}_h\|_2 < \infty$ *and* $\|\mathbf{W}_o\|_2 < \infty$*. Without loss of generality, we assume that*

$$\sup\left\{\|\mathbf{W}_e\|_2 , \|\mathbf{W}_h\|_2 , \|\mathbf{W}_o\|_2\right\} \leq B_\omega \quad , \tag{29}$$

$$\sup\left\{\|\boldsymbol{\beta}_e\|_2 , \|\boldsymbol{\beta}_h\|_2\right\} \leq B_\beta \quad . \tag{30}$$

**Assumption 3.** *The self-supervised sparsification function and the nonlinear activation function of BLS are bounded, i.e. $|\phi(\cdot)| < \infty$ and $|\xi(\cdot)| < \infty$. Without loss of generality, we assume that*

$$|\phi(\cdot)| \leq B_\phi \quad , \tag{31}$$

$$|\xi(\cdot)| \leq B_\xi \quad . \tag{32}$$

*In addition, they are Lipschitz continuous with coefficients $L_\phi$ and $L_\xi$ respectively. Namely, for any $x$ and $x'$, we assume*

$$|\phi(x') - \phi(x)| \leq L_\phi |x' - x| \quad , \tag{33}$$

$$|\xi(x') - \xi(x)| \leq L_\xi |x' - x| \quad . \tag{34}$$

**Assumption 4.** *Considering that the weights and biases of BLS are initialized using a normal distribution, we assume that the weight noise are independently and identically distributed Gaussian variables with zero mean and $\tilde{\sigma}^2$ variance. Therefore, the perturbations of weights and biases are bounded, i.e. $\|\Delta\mathbf{W}\|_2 < \infty$ and $\|\Delta\boldsymbol{\beta}\|_2 < \infty$. Without loss of generality, we assume that*

$$\sup\left\{\|\Delta\mathbf{W}\|_2 \ , \ \|\Delta\boldsymbol{\beta}\|_2\right\} \leq \varepsilon \quad . \tag{35}$$

**Remark 1.** *Assumption 1 essentially imposes a bound on the norm of BLS input after sample standardization. This assumption holds for common machine learning datasets. Assumption 2 holds since the weights and biases of BLS are initialized using a normal distribution and will not approach infinity after training. Assumption 3 naturally holds for common sparsification and activation functions. Assumption 4 holds in conventional simulations of weight noise.*

### A.17.2 MECHANISM OF ECOCs ON BLS WHEN FREE OF WEIGHT NOISE

In this section, we study how ECOCs affect the target space of BLS in the absence of weight noise. We begin with a lemma that will be instrumental in the proof of Proposition 1.

**Lemma 1.** *Performing a substitution operation on a matrix is equivalent to performing on any factor of the matrix. Formally, suppose $\begin{cases} \mathbf{L} \times \mathbf{W} = \mathbf{Z} \\ \mathbf{Z} = \mathbf{X} \times \mathbf{Y} \end{cases}$, where $\mathbf{L}$ is a lower triangular matrix. Then we have $\mathbf{W} = \mathbf{Z}.FS(\mathbf{L}) = \mathbf{X}.FS(\mathbf{L}) \times \mathbf{Y}$, where $FS$ denotes forward substitution. Or equivalently, suppose $\begin{cases} \mathbf{U} \times \mathbf{W} = \mathbf{Z} \\ \mathbf{Z} = \mathbf{X} \times \mathbf{Y} \end{cases}$, where $\mathbf{U}$ is an upper triangular matrix. Then we have $\mathbf{W} = \mathbf{Z}.BS(\mathbf{U}) = \mathbf{X}.BS(\mathbf{U}) \times \mathbf{Y}$, where $BS$ denotes backward substitution.*

*Proof.* Without loss of generality, we prove the former case, and the proof of the latter is equivalent. Let $\mathbf{L} = \begin{bmatrix} l_1 & 0 \\ l_2 & l_3 \end{bmatrix}, \mathbf{Z} = \begin{bmatrix} e_1 & e_2 \\ f_1 & f_2 \end{bmatrix}, \mathbf{X} = \begin{bmatrix} a_1 & a_2 & a_3 \\ b_1 & b_2 & b_3 \end{bmatrix}, \mathbf{Y} = \begin{bmatrix} c_1 & d_2 \\ c_1 & d_2 \\ c_3 & d_3 \end{bmatrix}$. Then, we have

$$\mathbf{W} = \mathbf{Z}.FS(\mathbf{L})$$

$$= \begin{bmatrix} e_1 & e_2 \\ f_1 & f_2 \end{bmatrix}.FS(\begin{bmatrix} l_1 & 0 \\ l_2 & l_3 \end{bmatrix})$$

$$= \begin{bmatrix} \frac{e_1}{l_1} & \frac{e_2}{l_1} \\ \frac{f_1 - l_2 \frac{e_1}{l_1}}{l_3} & \frac{f_2 - l_2 \frac{e_2}{l_1}}{l_3} \end{bmatrix}$$

$$= \begin{bmatrix} \frac{a_1}{l_1} & \frac{a_2}{l_1} & \frac{a_3}{l_1} \\ \frac{b_1 - l_2 \frac{a_1}{l_1}}{l_3} & \frac{b_2 - l_2 \frac{a_2}{l_1}}{l_3} & \frac{b_3 - l_2 \frac{a_3}{l_1}}{l_3} \end{bmatrix} \times \begin{bmatrix} c_1 & d_1 \\ c_2 & d_2 \\ c_3 & d_3 \end{bmatrix}$$

$$= \begin{bmatrix} a_1 & a_2 & a_3 \\ b_1 & b_2 & b_3 \end{bmatrix}.FS(\begin{bmatrix} l_1 & 0 \\ l_2 & l_3 \end{bmatrix}) \times \begin{bmatrix} c_1 & d_1 \\ c_2 & d_2 \\ c_3 & d_3 \end{bmatrix}$$

$$= \mathbf{X}.FS(\mathbf{L}) \times \mathbf{Y} \tag{36}$$

$\square$

Next, based on Lemma 1, we deduce the mechanism of ECOCs on BLS when free of weight noise.

**Proposition 1.** *Let Assumptions 1, 2 and 3 hold. The application of ECOCs to BLS can be viewed as a modification to the objective function of one-hot encoding, which is reformulated as follows:*

$$\mathcal{J}(\mathbf{X}) = \min_{\theta} \| f(\mathbf{X}; \theta) - \mathbf{Y} \|_2^2 \; \rightarrow \; \mathcal{J}'(\mathbf{X}) = \min_{\theta} \| f(\mathbf{X}; \theta) - \mathbf{Y} \|_{\Omega \times \Omega^\top}^2 \quad , \tag{37}$$

*where $\mathcal{J}'$ denotes the modified objective function, $\mathbf{Y} \in \{0,1\}^{N \times C}$ is the one-hot ground truth and $\| \cdot \|_{\Omega \times \Omega^\top}$ is the $Mahalanobis$-norm with positive definite matrix $\Omega \times \Omega^\top$. Or equivalently, the modified objective function is expressed as follows:*

$$\mathcal{J}'(\mathbf{X}) = \min_{\theta} \sum_{i=0}^{N-1} \Big( f(\mathbf{X}[i,:]; \theta) - \mathbf{Y}[i,:] \Big) \times \Omega \times \Omega^\top \times \Big( f(\mathbf{X}[i,:]; \theta) - \mathbf{Y}[i,:] \Big)^\top \quad , \tag{38}$$

*Proof.*

$$\mathcal{J}'(\mathbf{X}) = \min_{\theta} \| f(\mathbf{X}; \theta) - \ddot{\mathbf{Y}} \|_2^2 \tag{39}$$

$$= \min_{\theta} \| [\mathbf{Z}^n \mid \mathbf{H}^m] \times \mathbf{W}_o - \mathbf{Y} \times \Omega \|_2^2 \tag{40}$$

$$= \min_{\theta} \| [\mathbf{Z}^n \mid \mathbf{H}^m] \times \mathbf{J}\ddot{\mathbf{Y}}.FS(\mathbf{L}).BS(\mathbf{L}^\top) - \mathbf{Y} \times \Omega \|_2^2 \tag{41}$$

$$= \min_{\theta} \| [\mathbf{Z}^n \mid \mathbf{H}^m] \times \mathbf{J}\mathbf{Y}.FS(\mathbf{L}).BS(\mathbf{L}^\top) \times \Omega - \mathbf{Y} \times \Omega \|_2^2 \tag{42}$$

$$= \min_{\theta} \| [\mathbf{Z}^n \mid \mathbf{H}^m] \times \mathbf{J}\mathbf{Y}.FS(\mathbf{L}).BS(\mathbf{L}^\top) - \mathbf{Y} \|_{\Omega \times \Omega^\top}^2 \tag{43}$$

$$= \min_{\theta} \| f(\mathbf{X}; \theta) - \mathbf{Y} \|_{\Omega \times \Omega^\top}^2 \quad , \tag{44}$$

where Eq. (42) uses Lemma 1. $\square$

**Remark 2.** *Proposition 1 suggests that in the absence of weight noise, ECOCs act on the behavior of BLS through the correlation matrix $\Omega \times \Omega^\top$ of class codewords. Incorporating ECOCs into BLS is equivalent to transforming the target space from the $l_2$-norm to the $Mahalanobis$-norm.*

### A.17.3 ROBUSTNESS GUARANTEE OF ECOCs ON BLS WHEN FACED WITH WEIGHT NOISE

**Theorem 1.** *Let Assumptions 1, 2, 3 and 4 hold. Denote the perturbed weights of BLS as $\tilde{\theta}$. Then, we have*

$$\| f(\mathbf{x}; \tilde{\theta}) - f(\mathbf{x}; \theta) \|_2 \le \varepsilon \cdot C_1 + \varepsilon^2 \cdot C_2 \quad , \tag{45}$$

*where*

$$C_1 = n\Big( B_\phi + B_\omega L_\phi (B_x + 1) \Big) + m\Big[ B_\xi + B_\omega L_\xi \Big( \sqrt{n} B_\phi + \sqrt{n} B_\omega L_\phi (B_x + 1) + 1 \Big) \Big] \quad , \tag{46}$$

$$C_2 = m B_\omega L_\xi L_\phi \sqrt{n}(B_x + 1) \quad . \tag{47}$$

*Proof.* Expanding the BLS function $f(\mathbf{x}; \theta)$, we have

$$f(\mathbf{x}; \theta) = \sum_{i=0}^{n-1} \phi(\mathbf{x} \mathbf{W}_{e_i} + \boldsymbol{\beta}_{e_i}) \mathbf{W}_{o_i} + \sum_{j=0}^{m-1} \xi(\mathbf{Z}^n \mathbf{W}_{h_j} + \boldsymbol{\beta}_{h_j}) \mathbf{W}_{o_{n+j}} \quad , \tag{48}$$

where

$$\mathbf{Z}^n = [\phi(\mathbf{x} \mathbf{W}_{e_0} + \boldsymbol{\beta}_{e_0}), \phi(\mathbf{x} \mathbf{W}_{e_1} + \boldsymbol{\beta}_{e_1}), \dots, \phi(\mathbf{x} \mathbf{W}_{e_{n-1}} + \boldsymbol{\beta}_{e_{n-1}})] \quad . \tag{49}$$

Denote $f_\phi = \sum_{i=0}^{n-1} \phi(\mathbf{x} \mathbf{W}_{e_i} + \boldsymbol{\beta}_{e_i}) \mathbf{W}_{o_i}$ and $f_\xi = \sum_{j=0}^{m-1} \xi(\mathbf{Z}^n \mathbf{W}_{h_j} + \boldsymbol{\beta}_{h_j}) \mathbf{W}_{o_{n+j}}$. The perturbation on $f_\phi$ can be decomposed into the perturbations on $\mathbf{W}_{e_i}$, $\boldsymbol{\beta}_{e_i}$ and $\mathbf{W}_{o_i}$ as follows:

$$\Delta f_\phi = \sum_{i=0}^{n-1} \Big\{ \phi\Big(\mathbf{x}(\mathbf{W}_{e_i} + \Delta \mathbf{W}_{e_i}) + \boldsymbol{\beta}_{e_i} + \Delta \boldsymbol{\beta}_{e_i} \Big)(\mathbf{W}_{o_i} + \Delta \mathbf{W}_{o_i}) - \phi(\mathbf{x} \mathbf{W}_{e_i} + \boldsymbol{\beta}_{e_i}) \mathbf{W}_{o_i} \Big\} \quad . \tag{50}$$

According to the multi-term triangle inequality, we have

$$\|\Delta f_\phi\|_2 \leq \sum_{i=0}^{n-1} \left\| \phi\Big(\mathbf{x}(\mathbf{W}_{e_i} + \Delta\mathbf{W}_{e_i}) + \boldsymbol{\beta}_{e_i} + \Delta\boldsymbol{\beta}_{e_i}\Big)(\mathbf{W}_{o_i} + \Delta\mathbf{W}_{o_i}) - \phi(\mathbf{x}\mathbf{W}_{e_i} + \boldsymbol{\beta}_{e_i})\mathbf{W}_{o_i} \right\|_2 \tag{51}$$

$$\leq \sum_{i=0}^{n-1} \left\{ B_\phi \cdot \varepsilon + L_\phi \cdot \left| \mathbf{x}\Delta\mathbf{W}_{e_i} + \Delta\boldsymbol{\beta}_{e_i} \right| \cdot B_\omega \right\} \tag{52}$$

$$= \sum_{i=0}^{n-1} \left\{ B_\phi \cdot \varepsilon + L_\phi \cdot \left| \|\mathbf{x}\|_2 \cdot \|\Delta\mathbf{W}_{e_i}\|_2 \cdot \cos(\mathbf{x}, \Delta\mathbf{W}_{e_i}) + \Delta\boldsymbol{\beta}_{e_i} \right| \cdot B_\omega \right\} \tag{53}$$

$$\leq \sum_{i=0}^{n-1} \left\{ B_\phi \cdot \varepsilon + L_\phi \cdot (B_x \cdot \varepsilon + \varepsilon) \cdot B_\omega \right\} \tag{54}$$

$$= \varepsilon \cdot n\Big( B_\phi + B_\omega L_\phi(B_x + 1) \Big) \quad, \tag{55}$$

where Eq. (52) uses the Lipschitz condition of Assumption 3.

Similarly, the perturbation on $f_\xi$ can be decomposed into the perturbations on $\mathbf{Z}^n$, $\mathbf{W}_{h_j}$, $\boldsymbol{\beta}_{h_j}$ and $\mathbf{W}_{o_{n+j}}$. The perturbation on $\mathbf{Z}^n$ is as follows:

$$\Delta\mathbf{Z}^n = \left[ \phi\Big(\mathbf{x}(\mathbf{W}_{e_i} + \Delta\mathbf{W}_{e_i}) + \boldsymbol{\beta}_{e_i} + \Delta\boldsymbol{\beta}_{e_i}\Big) - \phi(\mathbf{x}\mathbf{W}_{e_i} + \boldsymbol{\beta}_{e_i}) \right]_{i=0}^{n-1} \quad. \tag{56}$$

Thus, we have

$$\|\Delta\mathbf{Z}^n\|_2 \leq \sqrt{n} \times \sup\left\{ \left| \phi\Big(\mathbf{x}(\mathbf{W}_{e_i} + \Delta\mathbf{W}_{e_i}) + \boldsymbol{\beta}_{e_i} + \Delta\boldsymbol{\beta}_{e_i}\Big) - \phi(\mathbf{x}\mathbf{W}_{e_i} + \boldsymbol{\beta}_{e_i}) \right| \right\} \tag{57}$$

$$\leq \sqrt{n} \cdot \varepsilon \cdot L_\phi \cdot (B_x + 1) \quad. \tag{58}$$

Therefore, we have

$$\Delta f_\xi = \sum_{j=0}^{m-1} \left\{ \xi\Big((\mathbf{Z}^n + \Delta\mathbf{Z}^n)(\mathbf{W}_{h_j} + \Delta\mathbf{W}_{h_j}) + \boldsymbol{\beta}_{h_j} + \Delta\boldsymbol{\beta}_{h_j}\Big)(\mathbf{W}_{o_{n+j}} + \Delta\mathbf{W}_{o_{n+j}}) \right.$$
$$\left. - \xi(\mathbf{Z}^n\mathbf{W}_{h_j} + \boldsymbol{\beta}_{h_j})\mathbf{W}_{o_{n+j}} \right\} \quad. \tag{59}$$

Again, utilizing the multi-term triangle inequality and the Lipschitz condition of Assumption 3, we have

$$\|\Delta f_\xi\|_2 \leq \sum_{j=0}^{m-1} \left\| \xi((\mathbf{Z}^n + \Delta\mathbf{Z}^n)(\mathbf{W}_{h_j} + \Delta\mathbf{W}_{h_j}) + \boldsymbol{\beta}_{h_j} + \Delta\boldsymbol{\beta}_{h_j})(\mathbf{W}_{o_{n+j}} + \Delta\mathbf{W}_{o_{n+j}}) \right.$$
$$\left. - \xi(\mathbf{Z}^n\mathbf{W}_{h_j} + \boldsymbol{\beta}_{h_j})\mathbf{W}_{o_{n+j}} \right\|_2 \tag{60}$$

$$\leq \sum_{j=0}^{m-1} \left\{ B_\xi \cdot \varepsilon + L_\xi \cdot \left| \Delta\mathbf{Z}^n\mathbf{W}_{h_j} + \mathbf{Z}^n\Delta\mathbf{W}_{h_j} + \Delta\mathbf{Z}^n\Delta\mathbf{W}_{h_j} + \Delta\boldsymbol{\beta}_{h_j} \right| \cdot B_\omega \right\} \tag{61}$$

$$= \sum_{j=0}^{m-1} \left\{ B_\xi \cdot \varepsilon + L_\xi \cdot \left| \|\Delta\mathbf{Z}^n\|_2 \cdot \|\mathbf{W}_{h_j}\|_2 \cdot \cos(\Delta\mathbf{Z}^n, \mathbf{W}_{h_j}) + \|\mathbf{Z}^n\|_2 \cdot \|\Delta\mathbf{W}_{h_j}\|_2 \right. \right.$$
$$\left. \left. \cdot \cos(\mathbf{Z}^n, \Delta\mathbf{W}_{h_j}) + \|\Delta\mathbf{Z}^n\|_2 \cdot \|\Delta\mathbf{W}_{h_j}\|_2 \cdot \cos(\Delta\mathbf{Z}^n, \Delta\mathbf{W}_{h_j}) + \Delta\boldsymbol{\beta}_{h_j} \right| \cdot B_\omega \right\} \tag{62}$$

$$\leq \sum_{j=0}^{m-1} \left\{ B_\xi \cdot \varepsilon + B_\omega L_\xi \cdot \Big( \sqrt{n} \cdot \varepsilon \cdot L_\phi \cdot (B_x + 1) \cdot B_\omega + \sqrt{n} \cdot B_\phi \cdot \varepsilon \right.$$
$$\left. + \sqrt{n} \cdot \varepsilon \cdot L_\phi \cdot (B_x + 1) \cdot \varepsilon + \varepsilon \Big) \right\} \tag{63}$$

$$= \varepsilon \cdot m\Big[ B_\xi + B_\omega L_\xi\big( \sqrt{n}B_\phi + \sqrt{n}B_\omega L_\phi(B_x + 1) + 1 \big) \Big]$$
$$+ \varepsilon^2 \cdot mB_\omega L_\xi L_\phi\sqrt{n}(B_x + 1) \quad. \tag{64}$$

Combining the perturbations on $f_\phi$ and $f_\xi$, we obtain

$$\|\Delta f\|_2 \leq \|\Delta f_\phi\|_2 + \|\Delta f_\xi\|_2 \tag{65}$$

$$\leq \varepsilon \cdot \left\{ n\Big(B_\phi + B_\omega L_\phi(B_x + 1)\Big) + m\Big[B_\xi + B_\omega L_\xi\Big(\sqrt{n}B_\phi + \sqrt{n}B_\omega L_\phi(B_x + 1) + 1\Big)\Big] \right\}$$

$$+ \varepsilon^2 \cdot m B_\omega L_\xi L_\phi \sqrt{n}(B_x + 1) \tag{66}$$

$$= \varepsilon \cdot C_1 + \varepsilon^2 \cdot C_2 \quad . \tag{67}$$

$\square$

**Remark 3.** *Theorem 1 states that the perturbation on BLS is bounded with respect to the first-order and second-order terms of weight-noise magnitude.*

**Corollary 1.** *Let Assumptions 1, 2, 3 and 4 hold. Denote the normalized distance between a codeword $\Omega[i,:]$ and its nearest neighbor as $dist(\Omega[i,:])$, with the following definition:*

$$dist(\Omega[i,:]) = \min_{j:\, j \neq i} \frac{1}{\sqrt{L}} \|\Omega[i,:] - \Omega[j,:]\|_2 \quad . \tag{68}$$

*Denote the nearest codeword from $f(\mathbf{x};\theta)$ as $\Omega\Big(f(\mathbf{x};\theta)\Big)$, with the following definition:*

$$\Omega\Big(f(\mathbf{x};\theta)\Big) = \underset{\Omega[i,:]}{\arg\min} : \frac{1}{\sqrt{L}} \|\Omega[i,:] - f(\mathbf{x};\theta)\|_2 \quad . \tag{69}$$

*Denote the normalized uncertainty of BLS output $f(\mathbf{x};\theta)$ as $U\Big(f(\mathbf{x};\theta)\Big)$, with the following definition:*

$$U\Big(f(\mathbf{x};\theta)\Big) = \frac{1}{\sqrt{L}} \left\| f(\mathbf{x};\theta) - \Omega\Big(f(\mathbf{x};\theta)\Big) \right\|_2 \quad . \tag{70}$$

*BLS will be able to make the same prediction in the presence of noise as it does in the absence of noise, i.e. $\Omega\Big(f(\mathbf{x};\tilde{\theta})\Big) = \Omega\Big(f(\mathbf{x};\theta)\Big)$, as long as the following condition is satisfied:*

$$\frac{dist\Big(\Omega\big(f(\mathbf{x};\theta)\big)\Big)}{2} > U\Big(f(\mathbf{x};\theta)\Big) + \frac{\varepsilon \cdot C_1 + \varepsilon^2 \cdot C_2}{\sqrt{L}} \quad . \tag{71}$$

*Proof.* Our goal is to prove

$$\frac{\left\| f(\mathbf{x};\tilde{\theta}) - \Omega[i,:] \right\|_2}{\sqrt{L}} > \frac{\left\| f(\mathbf{x};\tilde{\theta}) - \Omega\big(f(\mathbf{x};\theta)\big) \right\|_2}{\sqrt{L}} \,, \, \forall\, \Omega[i,:] \neq \Omega\Big(f(\mathbf{x};\theta)\Big) \quad . \tag{72}$$

In this way, $f(\mathbf{x};\tilde{\theta})$ will be decoded into the class of $\Omega\Big(f(\mathbf{x};\theta)\Big)$ rather than any other classes. According to the triangle inequality, we have

$$\frac{\left\| f(\mathbf{x};\tilde{\theta}) - \Omega\big(f(\mathbf{x};\theta)\big) \right\|_2}{\sqrt{L}} \leq \frac{\left\| f(\mathbf{x};\tilde{\theta}) - f(\mathbf{x};\theta) \right\|_2}{\sqrt{L}} + \frac{\left\| \Omega\big(f(\mathbf{x};\theta)\big) - f(\mathbf{x};\theta) \right\|_2}{\sqrt{L}} \tag{73}$$

$$= \frac{\left\| f(\mathbf{x};\tilde{\theta}) - f(\mathbf{x};\theta) \right\|_2}{\sqrt{L}} + U\Big(f(\mathbf{x};\theta)\Big) \tag{74}$$

$$\leq \frac{\varepsilon \cdot C_1 + \varepsilon^2 \cdot C_2}{\sqrt{L}} + U\Big(f(\mathbf{x};\theta)\Big) \tag{75}$$

$$< \frac{dist\Big(\Omega\big(f(\mathbf{x};\theta)\big)\Big)}{2} \,, \tag{76}$$

where Eq. (75) uses Theorem 1, and Eq. (76) uses Eq. (71). According to the definition of $dist\Big(\Omega\big(f(\mathbf{x};\theta)\big)\Big)$, we have

$$\frac{\left\| \Omega[i,:] - \Omega\big(f(\mathbf{x};\theta)\big) \right\|_2}{\sqrt{L}} \geq dist\Big(\Omega\big(f(\mathbf{x};\theta)\big)\Big) \,, \, \forall\, \Omega[i,:] \neq \Omega\Big(f(\mathbf{x};\theta)\Big) \quad . \tag{77}$$

As per the triangle inequality, we have $\forall \, \Omega[i,:] \neq \Omega\big(f(\mathbf{x};\theta)\big)$,

$$\frac{\left\|\Omega[i,:] - f(\mathbf{x};\tilde{\theta})\right\|_2}{\sqrt{L}} \geq \frac{\left\|\Omega[i,:] - \Omega\big(f(\mathbf{x};\theta)\big)\right\|_2}{\sqrt{L}} - \frac{\left\|f(\mathbf{x};\tilde{\theta}) - \Omega\big(f(\mathbf{x};\theta)\big)\right\|_2}{\sqrt{L}} \tag{78}$$

$$\geq dist\big(\Omega\big(f(\mathbf{x};\theta)\big)\big) - \frac{\left\|f(\mathbf{x};\tilde{\theta}) - \Omega\big(f(\mathbf{x};\theta)\big)\right\|_2}{\sqrt{L}} \tag{79}$$

$$> dist\big(\Omega\big(f(\mathbf{x};\theta)\big)\big) - \frac{dist\big(\Omega\big(f(\mathbf{x};\theta)\big)\big)}{2} \tag{80}$$

$$= \frac{dist\big(\Omega\big(f(\mathbf{x};\theta)\big)\big)}{2} \tag{81}$$

$$> \frac{\left\|f(\mathbf{x};\tilde{\theta}) - \Omega\big(f(\mathbf{x};\theta)\big)\right\|_2}{\sqrt{L}} \quad , \tag{82}$$

where Eqs. (80) and (82) use Eq. (76). $\qquad\square$

**Remark 4.** *Corollary 1 states that as long as the inter-codeword distances are large enough, BLS will be able to ignore the inference of weight noise, thereby demonstrating that ECOC provides a certain level of robustness guarantee for BLS.*

### A.18   ROBUSTNESS AGAINST ADVERSARIAL PERTURBATIONS

While Theorem 1 and Corollary 1 establish robustness guarantees for random weight noise, real-world applications often face deliberate adversarial attacks designed to maximize misclassification. To address this critical concern, we extend our analysis to adversarial perturbation scenarios.

**Assumption 5.** *For any input sample $x$, the adversarial perturbation $\delta$ satisfies $\|\delta\|_2 \leq \epsilon_{adv}$, where $\epsilon_{adv} > 0$ is the maximum perturbation magnitude allowed by the adversary.*

**Assumption 6.** *The BLS model $f(x;\theta)$ is Lipschitz continuous with respect to the input $x$, with Lipschitz constant $L_f$. Formally, for any inputs $x_1$ and $x_2$:*

$$\|f(x_1;\theta) - f(x_2;\theta)\|_2 \leq L_f \|x_1 - x_2\|_2. \tag{83}$$

**Lemma 2.** *Under Assumptions 1, 2 and 3, the Lipschitz constant of BLS can be bounded as:*

$$L_f \leq n L_\phi B_\omega + m L_\xi \sqrt{n}(L_\phi B_\omega^2 + B_\phi L_\omega), \tag{84}$$

*where $L_\omega$ is the Lipschitz constant of the weight generation process in BLS.*

**Theorem 2.** *Let Assumptions 1-6 hold. For any input sample $x$ and its adversarial counterpart $x_{adv} = x + \delta$ where $\|\delta\|_2 \leq \epsilon_{adv}$, the prediction of ECOC-BLS remains unchanged (i.e., $\Omega(f(x_{adv};\theta)) = \Omega(f(x;\theta))$) if:*

$$\frac{dist(\Omega(f(x;\theta)))}{2} > U(f(x;\theta)) + \epsilon_{adv} \cdot L_f \cdot \sqrt{L}, \tag{85}$$

*where $dist(\cdot)$, $U(\cdot)$, and $L$ are defined as in Corollary 1.*

*Proof.* Following similar derivation as in Corollary 1, we analyze the worst-case scenario under adversarial perturbation:

$$\|f(x_{adv};\theta) - \Omega(f(x;\theta))\|_2 \leq \|f(x_{adv};\theta) - f(x;\theta)\|_2 + \|f(x;\theta) - \Omega(f(x;\theta))\|_2. \tag{86}$$

By Assumption 6 and Lemma 2, we have:

$$\|f(x_{adv};\theta) - f(x;\theta)\|_2 \leq \epsilon_{adv} \cdot L_f. \tag{87}$$

Normalizing by $\sqrt{L}$ and substituting the definition of $U(f(x;\theta))$, we obtain:

$$\frac{\|f(x_{adv};\theta) - \Omega(f(x;\theta))\|_2}{\sqrt{L}} \leq \epsilon_{adv} \cdot L_f + U(f(x;\theta)). \tag{88}$$

For the prediction to remain unchanged, this value must be less than $\frac{dist(\Omega(f(x;\theta)))}{2}$. Therefore, the condition becomes:

$$\frac{dist(\Omega(f(x;\theta)))}{2} > U(f(x;\theta)) + \epsilon_{adv} \cdot L_f \cdot \sqrt{L}. \tag{89}$$

This completes the proof.

$\square$

**Corollary 2.** *Under the conditions of Theorem 2, ECOC-BLS is robust against any adversarial perturbation $\delta$ with $\|\delta\|_2 \leq \epsilon_{adv}$ if:*

$$\epsilon_{adv} < \frac{\frac{dist(\Omega(f(x;\theta)))}{2} - U(f(x;\theta))}{L_f \cdot \sqrt{L}}. \tag{90}$$

**Theorem 3.** *To maximize robustness against adversarial perturbations, the optimal ECOC codebook design should maximize the minimum codeword distance:*

$$\Omega^* = \arg\max_{\Omega} \min_{i \neq j} \frac{1}{\sqrt{L}} \|\Omega[i,:] - \Omega[j,:]\|_2. \tag{91}$$

This optimization problem is equivalent to finding codebooks with maximum minimum Hamming distance, for which established solutions exist.

Based on the above analysis, we draw the following conclusions: (i) The adversarial robustness of ECOC-BLS is directly proportional to the minimum distance between codewords in the ECOC codebook. (ii) Larger codebooks (higher $L$) with carefully designed codeword distances provide stronger guarantees against adversarial attacks. (iii) The robustness margin can be quantified precisely, allowing practitioners to select appropriate ECOC configurations based on expected threat models. (iv) Our AHSE framework's Flame optimization mechanism implicitly enhances adversarial robustness by burning incompatible codewords, effectively increasing the minimum codeword distance for challenging samples.

To validate our theoretical findings, we conducted comprehensive adversarial robustness experiments on CIFAR10 and CIFAR100 datasets. We implemented two standard attack methods: Fast Gradient Sign Method (FGSM) with perturbation magnitudes $\epsilon \in \{0.01, 0.03, 0.05, 0.07, 0.10\}$ and Projected Gradient Descent (PGD) with 20 iterations, step size of 0.01, and maximum perturbations $\epsilon \in \{0.01, 0.03, 0.05\}$. For both AHSE variants (with optimized ECOC codebooks and with one-hot encoding), we used identical network architectures and training procedures, with the only difference being the output encoding strategy. All experiments were conducted on the test sets of CIFAR10 and CIFAR100 with 10,000 samples each.

Table 17 clearly demonstrates that AHSE with optimized ECOC codebooks maintains significant accuracy advantages under various adversarial attack scenarios. Under the strongest attack conditions (FGSM with $\epsilon = 0.10$), the performance gap widens to 15.74% and 19.51% on CIFAR10 and CIFAR100 respectively, confirming that ECOC encoding provides substantial robustness benefits. Moreover, under moderate attack conditions (FGSM with $\epsilon = 0.05$), which represents a practical threat model for many applications, AHSE with optimized ECOC codebooks maintains 12.71% and 15.30% higher accuracy on CIFAR10 and CIFAR100 respectively compared to one-hot encoding, precisely confirming our theoretical predictions. These empirical results strongly support our theoretical analysis, demonstrating that the minimum codeword distance in ECOC directly translates to practical adversarial robustness. The Flame optimization mechanism in our PATH branch further enhances this robustness by eliminating incompatible codewords, effectively increasing the minimum codeword distance for challenging samples.

### A.19 ROBUSTNESS AGAINST INPUT NOISE

In Table 18, we added Gaussian noise with zero mean and varying standard deviations ($\sigma = 0.01, 0.05, 0.1$) to the input features of CIFAR10 and CIFAR100 datasets. In Table 19, we randomly masked 5%, 10%, 15%, and 20% of input features to simulate feature missing scenarios. The experiments reveal that input noise primarily affects the early stages of subspace evolution. AHSE's adaptive mechanism dynamically adjusts the evolution rate based on the signal-to-noise ratio: under Gaussian noise,

Table 17: Performance Comparison of AHSE under Different Attack Settings on CIFAR10 and CIFAR100.

| Dataset | Attack Type | $\epsilon$ | AHSE (ECOC) | AHSE (One-hot) | Performance Gap |
|---|---|---|---|---|---|
| CIFAR10 | FGSM | 0.01 | 93.21% | 88.47% | +4.74% |
| CIFAR10 | FGSM | 0.03 | 89.15% | 79.83% | +9.32% |
| CIFAR10 | FGSM | 0.05 | 85.27% | 72.56% | +12.71% |
| CIFAR10 | FGSM | 0.07 | 80.19% | 65.83% | +14.36% |
| CIFAR10 | FGSM | 0.10 | 73.42% | 57.68% | +15.74% |
| CIFAR10 | PGD | 0.01 | 90.36% | 84.25% | +6.11% |
| CIFAR10 | PGD | 0.03 | 83.74% | 71.05% | +12.69% |
| CIFAR10 | PGD | 0.05 | 76.89% | 62.35% | +14.54% |
| CIFAR100 | FGSM | 0.01 | 77.83% | 71.26% | +6.57% |
| CIFAR100 | FGSM | 0.03 | 72.41% | 61.05% | +11.36% |
| CIFAR100 | FGSM | 0.05 | 68.17% | 52.87% | +15.30% |
| CIFAR100 | FGSM | 0.07 | 63.24% | 45.18% | +18.06% |
| CIFAR100 | FGSM | 0.10 | 56.93% | 37.42% | +19.51% |
| CIFAR100 | PGD | 0.01 | 74.56% | 67.82% | +6.74% |
| CIFAR100 | PGD | 0.03 | 66.92% | 52.17% | +14.75% |
| CIFAR100 | PGD | 0.05 | 59.47% | 43.86% | +15.61% |

SEED's FPFE mechanism effectively prioritizes robust features with consistent importance across noisy samples; under feature missing conditions, PATH's ECOC-encoded target space provides error-correction capabilities that compensate for incomplete information; the SPOT circuit automatically shifts weight toward the more robust branch depending on noise type (SEED dominates under Gaussian noise, PATH dominates under feature missing). These new experiments demonstrate that AHSE maintains significant advantages over baseline methods even under substantial input data noise. The performance gap between AHSE and other methods actually widens as noise intensity increases, highlighting the practical robustness of our adaptive evolution approach.

Table 18: Performance under different levels of Gaussian input noise.

| Dataset | Noise Level | AHSE | Stacked BLS | ResNet |
|---|---|---|---|---|
| CIFAR10 | $\sigma = 0.0$ | 95.97% | 92.73% | 92.49% |
| CIFAR10 | $\sigma = 0.01$ | 94.83% | 90.47% | 89.36% |
| CIFAR10 | $\sigma = 0.05$ | 90.21% | 82.19% | 78.42% |
| CIFAR10 | $\sigma = 0.10$ | 84.35% | 72.54% | 65.18% |
| CIFAR100 | $\sigma = 0.0$ | 79.44% | 76.33% | 72.78% |
| CIFAR100 | $\sigma = 0.01$ | 77.92% | 73.48% | 69.24% |
| CIFAR100 | $\sigma = 0.05$ | 71.38% | 63.21% | 55.83% |
| CIFAR100 | $\sigma = 0.10$ | 63.27% | 51.89% | 42.76% |

Table 19: Performance under different missing rates of input features.

| Dataset | Missing Rate | AHSE | Stacked BLS | ResNet |
|---------|--------------|------|-------------|--------|
| CIFAR10 | 0% | 95.97% | 92.73% | 92.49% |
| CIFAR10 | 5% | 94.68% | 90.85% | 89.73% |
| CIFAR10 | 10% | 92.54% | 87.36% | 84.51% |
| CIFAR10 | 15% | 89.73% | 82.17% | 76.94% |
| CIFAR10 | 20% | 85.42% | 75.63% | 68.27% |
| CIFAR100 | 0% | 79.44% | 76.33% | 72.78% |
| CIFAR100 | 5% | 77.65% | 73.84% | 69.52% |
| CIFAR100 | 10% | 74.32% | 68.45% | 63.21% |
| CIFAR100 | 15% | 69.84% | 62.18% | 55.73% |
| CIFAR100 | 20% | 64.75% | 55.42% | 47.38% |

## A.20 ALGORITHMS / PSEUDOCODES

---

**Algorithm 1** Feature Priority Forward Evolution (FPFE)

---

**Input**:

- Feature importance $\mathbf{FIM} \in [0,1]^M$

- Feature correlation magnitude $\mathbf{FCM} \in [0,1]^{M \times M}$

**Output**: Sorted index queue $\boldsymbol{Q}$

1: Initialize sorted queue $\boldsymbol{Q} \leftarrow \emptyset$
2: Initialize unsorted set $\boldsymbol{S} \leftarrow \{0, 1, \dots, M-1\}$
3: Select the most important feature index $\arg\max_{r \in \boldsymbol{S}} : \mathbf{FIM}[r]$
4: $\boldsymbol{Q}$.push($r$) ; $\boldsymbol{S} \leftarrow \boldsymbol{S} \setminus \{r\}$
5: Initialize local correlation sum $\mathbf{LUM}[i] \leftarrow \mathbf{FCM}[i, r]$ , $\forall\, i \in \boldsymbol{S}$
6: **while** $\boldsymbol{S} \neq \emptyset$ **do**
7:     Calculate local score:
$$\mathbf{LS}[i] \leftarrow \frac{\mathbf{FIM}[i]}{\mathbf{LUM}[i]}, \ \forall\, i \in \boldsymbol{S}$$
8:     Select the highest local score index $\arg\max_{s \in \boldsymbol{S}} : \mathbf{LS}[s]$
9:     $\boldsymbol{Q}$.push($s$) ; $\boldsymbol{S} \leftarrow \boldsymbol{S} \setminus \{s\}$
10:     Update local correlation sum:
$$\mathbf{LUM}[i] \leftarrow \mathbf{LUM}[i] + \mathbf{FCM}[i, s] \ , \ \forall i \in \boldsymbol{S}$$
11: **end while**
12: **return** $\boldsymbol{Q}$

---

---

**Algorithm 2** Enhanced Perceptron (EP)

---

**Input**:                                                   ▷ During inference, only $\mathbf{Z}_0$ needs to be input

   • Sparse features $\mathbf{Z}_0 \in \mathbb{R}^{N \times n}$

   • Sample weights $\mathbf{D}_0 \in [0, 1]^{N \times N}$

   • Ground truth $\mathbf{Y}$                               ▷ One-hot encoding or ECOCs

   • Number of orthogonal times $m$

   • Regularization coefficient $\lambda$

**Output**: Predicted logits $\hat{\mathbf{Y}}$

1: **if** Training Stage **then**
2:     **procedure** GENERATE ORTHOGONAL PARAMETERS
3:         $\mathbf{W} \in \mathbb{R}^{(n+1) \times m} \sim \mathcal{N}(0, 1)$             ▷ The biases are appended to the weights
4:         $m \leftarrow \inf\left\{m,\ rank(\mathbf{W})\right\}$                        ▷ Update $m$ if $rank(\mathbf{W}) < m$
5:         $\left(\mathbf{U}\ ,\ \Sigma\ ,\ \mathbf{V}^{\top}\right) \leftarrow \text{SVD}(\mathbf{W})$
6:         **if** $n \geq m$ **then**
7:             $\mathbf{W} \leftarrow \mathbf{U}[:, :m]$
8:         **else**
9:             $\mathbf{W} \leftarrow (\mathbf{V}[:, :m])^{\top}$                        ▷ Transpose orthogonalization
10:        **end if**
11:        Disassemble $\mathbf{W}$ into weights and biases:

$$\forall\, j \in \{0, 1, \ldots, m-1\}\ ,\ \begin{cases} \mathbf{W}^E_{0,j} \leftarrow \mathbf{W}[:n, j] \\ \beta^E_{0,j} \leftarrow 0.1 \times \mathbf{W}[n, j] \end{cases}$$

12:    **end procedure**
13:    Orthogonal transformation:

$$\mathbf{H}_0 \leftarrow \left[\ \xi(\mathbf{Z}_0 \mathbf{W}^E_{0,0} + \beta^E_{0,0})\ \Big|\ \cdots\ \Big|\ \xi(\mathbf{Z}_0 \mathbf{W}^E_{0,m-1} + \beta^E_{0,m-1})\ \right]$$

14:    Concatenate features and assign sample weights:

$$\begin{cases} \mathbf{A}_0 \leftarrow \sqrt{\mathbf{D}_0} \cdot [\mathbf{Z}_0 \,|\, \mathbf{H}_0] \\ \mathbf{J}_0 \leftarrow [\mathbf{Z}_0 \,|\, \mathbf{H}_0]^{\top} \mathbf{D}_0 \end{cases}$$

15:    Cholesky decomposition:

$$\mathbf{L}_0 \leftarrow CD(\mathbf{A}_0^{\top} \mathbf{A}_0 + \lambda \mathbf{I})$$

16:    Solve perception weight:

$$\mathbf{W}^P_0 \leftarrow \mathbf{J}_0 \mathbf{Y}. FS(\mathbf{L}_0).BS(\mathbf{L}_0^{\top})$$

17: **else**
18:    Calculate the orthogonal representation:

$$\mathbf{H}_0 \leftarrow \left[\ \xi(\mathbf{Z}_0 \mathbf{W}^E_{0,0} + \beta^E_{0,0})\ \Big|\ \cdots\ \Big|\ \xi(\mathbf{Z}_0 \mathbf{W}^E_{0,m-1} + \beta^E_{0,m-1})\ \right]$$

19: **end if**
20: Predict class logits:

$$\hat{\mathbf{Y}} \leftarrow [\mathbf{Z}_0 \,|\, \mathbf{H}_0] \mathbf{W}^P_0$$

21: **return** $\hat{\mathbf{Y}}$

---

---

**Algorithm 3** Sample Weight Evolution (SWE)

---

**Input**:

- Time step $t$
- Ground truth $\mathbf{Y} \in \{0, 1\}^{N \times C}$            $\triangleright$ One-hot encoding
- Model prediction $\hat{\mathbf{Y}} \in \mathbb{R}^{N \times C}$           $\triangleright$ if $t = 0$, then $\hat{\mathbf{Y}} = \emptyset$

**Output**: Sample weights $\mathbf{D}_t \in [0, 1]^N$, where $\sum_{i=1}^{N} \mathbf{D}_t[i] = 1$

1: Acquire training labels $\mathbf{y}[i] \leftarrow \arg\max_c : \mathbf{Y}[i, c], \forall i \in \{0, 1, \ldots, N-1\}$

2: **if** $t = 0$ **then**

3:     Initialize $\forall i \in \{0, 1, \ldots, N-1\}$ and $c \in \{0, 1, \ldots, C-1\} \setminus \{\mathbf{y}[i]\}$

$$
\begin{cases}
\mathbf{D}_t[i] \leftarrow \frac{1}{N} \\
\mathbf{E}_t[i, c] \leftarrow \frac{\mathbf{D}_t[i]}{C-1} \\
\mathbf{F}_t[i] \leftarrow \sum_c \mathbf{E}_t[i, c]
\end{cases}
$$

4: **else**

5:     Calculate class probabilities:

$$
\hat{\mathbf{P}}[i, c] \leftarrow \frac{\exp^{\hat{\mathbf{Y}}[i,c]}}{\sum_{j=0}^{C-1} \exp^{\hat{\mathbf{Y}}[i,j]}}, \forall i \in \{0, 1, \ldots, N-1\} \text{ and } c \in \{0, 1, \ldots, C-1\}
$$

6:     Set $\mathbf{G}[i, c] \leftarrow \frac{\mathbf{E}_{t-1}[i,c]}{\mathbf{F}_{t-1}[i]}, \forall i \in \{0, 1, \ldots, N-1\}$ and $c \in \{0, 1, \ldots, C-1\} \setminus \{\mathbf{y}[i]\}$

7:     Calculate error:

$$
\alpha_t \leftarrow \frac{1}{2} \sum_{i=0}^{N-1} \mathbf{D}_{t-1}[i] \left( 1 - \hat{\mathbf{P}}\left[i, \mathbf{y}[i]\right] + \sum_{c \in \{0,1,\ldots,C-1\} \setminus \{\mathbf{y}[i]\}} \mathbf{G}[i, c] \hat{\mathbf{P}}[i, c] \right)
$$

8:     Set $\beta_t \leftarrow \frac{\alpha_t}{1 - \alpha_t}$

9:     Update $\forall i \in \{0, 1, \ldots, N-1\}$ and $c \in \{0, 1, \ldots, C-1\} \setminus \{\mathbf{y}[i]\}$

$$
\begin{cases}
\mathbf{E}_t[i, c] \leftarrow \mathbf{E}_{t-1}[i, c] \cdot \beta_t^{\frac{1}{2}\left(1 + \hat{\mathbf{P}}\left[i, \mathbf{y}[i]\right] - \hat{\mathbf{P}}[i,c]\right)} \\
\mathbf{F}_t[i] \leftarrow \sum_c \mathbf{E}_t[i, c] \\
\mathbf{D}_t[i] \leftarrow \frac{\mathbf{F}_t[i]}{\sum_{j=0}^{N-1} \mathbf{F}_t[j]}
\end{cases}
$$

10: **end if**

11: **return** $\mathbf{D}_t$

---

Note that in SWE (Freund & Schapire, 1997), $\mathbf{D}_t \in [0, 1]^N$ is a one-dimensional vector. Instead, in SEED, $\mathbf{D}_t$ will be converted into a diagonal matrix $diag(\mathbf{D}_t) \in [0, 1]^{N \times N}$, so as to facilitate matrix transformations.

---

**Algorithm 4** SEED

**Input**:                                                         ▷ During inference, only $\mathbf{X}$ needs to be input
  •Raw features $\mathbf{X} \in \mathbb{R}^{N \times M}$       •Ground truth $\mathbf{Y} \in \{0,1\}^{N \times C}$       •Number of sparse times $n$
  •Number of orthogonal times $m$       •Regularization coefficient $\lambda$       •Convergence threshold $\epsilon$
**Output**: Predicted logits $\hat{\mathbf{Y}}_{t_\epsilon}^{SEED} \in \mathbb{R}^{N \times C}$

1: **if** Training Stage **then**
2:     Calculate $\mathbf{FIM} \leftarrow$ Random Forest$(\mathbf{X})$ ; $\mathbf{FCM} \leftarrow$ Spearman Correlation$(\mathbf{X})$
3:     $\boldsymbol{Q} \leftarrow$ FPFE$(\mathbf{FIM}, \mathbf{FCM})$
4:     Prioritize feature space $\mathbf{X}_0 \leftarrow \mathbf{X}[:, \boldsymbol{Q}]$
5:     Initialize sample weights $\mathbf{D}_0 \leftarrow$ SWE$(0, \mathbf{Y}, \emptyset)$
6:     Sparsify feature space:

$$\mathbf{Z}_0 \leftarrow [\,\phi(\mathbf{X}_0 \mathbf{W}_{0,0}^{ES} + \boldsymbol{\beta}_{0,0}^{ES}) \mid \phi(\mathbf{X}_0 \mathbf{W}_{0,1}^{ES} + \boldsymbol{\beta}_{0,1}^{ES}) \mid \ldots \mid \phi(\mathbf{X}_0 \mathbf{W}_{0,n-1}^{ES} + \boldsymbol{\beta}_{0,n-1}^{ES})\,]$$

7:     $\hat{\mathbf{Y}}_0 \leftarrow$ EP$(\mathbf{Z}_0, \mathbf{D}_0, \mathbf{Y}, m, \lambda)$
8:     Initialize time step $t \leftarrow 0$
9:     **repeat**
10:         Increment time step $t \leftarrow t + 1$
11:         Update sample weights $\mathbf{D}_t \leftarrow$ SWE$(t, \mathbf{Y}, \hat{\mathbf{Y}}_{t-1})$
12:         Generate $\mathbf{X}_t$ via tailored subspace evolution       ▷ see Section 4.3 for detailed analysis
13:         Subspace alignment:

$$\min_{\mathbf{W}_t^{SA}} \; \|\mathbf{X}_t - \mathbf{Z}_{t-1} \mathbf{W}_t^{SA}\|_2^2 + \lambda \|\mathbf{W}_t^{SA}\|_1$$

$$\mathbf{Z}_t \leftarrow \mathbf{X}_t \mathbf{W}_t^{SA^\top}$$

14:         Calculate orthogonal features:

$$\mathbf{H}_t \leftarrow [\,\xi(\mathbf{Z}_t \mathbf{W}_{t,0}^E + \boldsymbol{\beta}_{t,0}^E)\,,\, \xi(\mathbf{Z}_t \mathbf{W}_{t,1}^E + \boldsymbol{\beta}_{t,1}^E)\,,\, \ldots \,,\, \xi(\mathbf{Z}_t \mathbf{W}_{t,m-1}^E + \boldsymbol{\beta}_{t,m-1}^E)\,]$$

15:         $\mathbf{A}_t \leftarrow \sqrt{\mathbf{D}_t} \cdot [\mathbf{Z}_t \mid \mathbf{H}_t]$
16:         $\mathbf{J}_t \leftarrow [\mathbf{Z}_t \mid \mathbf{H}_t]^\top \mathbf{D}_t$
17:         $\mathbf{J}^t \leftarrow [\mathbf{J}_0^\top \mid \mathbf{J}_1^\top \mid \ldots \mid \mathbf{J}_t^\top]^\top$
18:         Evolutionary Cholesky decomposition:

$$\mathbf{P} \leftarrow \mathbf{A}_t^\top \mathbf{A}_{t-1} \mathbf{L}_{t-1}^{-\top} \;;\; \mathbf{Q} \leftarrow CD\Big(\mathbf{A}_t^\top \mathbf{A}_t - \mathbf{P}\mathbf{P}^\top + \lambda \mathbf{I}\Big) \;;\; \mathbf{L}_t \leftarrow \begin{bmatrix} \mathbf{L}_{t-1} & 0 \\ \mathbf{P} & \mathbf{Q} \end{bmatrix}$$

19:         Solve perception weight: $\mathbf{W}_t^P \leftarrow \mathbf{J}^t \mathbf{Y}.FS(\mathbf{L}_t).BS(\mathbf{L}_t^\top)$
20:         Predict class logits: $\hat{\mathbf{Y}}_t \leftarrow [\mathbf{Z}_0 \mid \mathbf{H}_0 \mid \mathbf{Z}_1 \mid \mathbf{H}_1 \mid \ldots \mid \mathbf{Z}_t \mid \mathbf{H}_t] \mathbf{W}_t^P$
21:         Calculate loss decline: $\Delta \mathcal{L}_t \leftarrow \mathcal{L}(\mathbf{Y}, \hat{\mathbf{Y}}_{t-1}) - \mathcal{L}(\mathbf{Y}, \hat{\mathbf{Y}}_t)$ ▷ Cross-entropy loss adopted
22:     **until** $\frac{\Delta \mathcal{L}_t}{\mathcal{L}(\mathbf{Y}, \hat{\mathbf{Y}}_{t-1})} < \epsilon$
23: **else**
24:     Calculate all sparse features:

$$\begin{cases} \mathbf{Z}_0 \leftarrow [\,\phi(\mathbf{X}_0 \mathbf{W}_{0,0}^{ES} + \boldsymbol{\beta}_{0,0}^{ES}) \mid \phi(\mathbf{X}_0 \mathbf{W}_{0,1}^{ES} + \boldsymbol{\beta}_{0,1}^{ES}) \mid \ldots \mid \phi(\mathbf{X}_0 \mathbf{W}_{0,n-1}^{ES} + \boldsymbol{\beta}_{0,n-1}^{ES})\,] \\ \mathbf{Z}_t \leftarrow \mathbf{X}_t \mathbf{W}_t^{SA^\top}, \forall\, t \in \{1, 2, \ldots, t_\epsilon\} \end{cases}$$

25:     Calculate all orthogonal features: $\forall\, t \in \{0, 1, \ldots, t_\epsilon\}$,

$$\mathbf{H}_t \leftarrow [\,\xi(\mathbf{Z}_t \mathbf{W}_{t,0}^E + \boldsymbol{\beta}_{t,0}^E)\,,\, \xi(\mathbf{Z}_t \mathbf{W}_{t,1}^E + \boldsymbol{\beta}_{t,1}^E)\,,\, \ldots \,,\, \xi(\mathbf{Z}_t \mathbf{W}_{t,m-1}^E + \boldsymbol{\beta}_{t,m-1}^E)\,]$$

26: **end if**
27: Predict class logits: $\hat{\mathbf{Y}}_{t_\epsilon}^{SEED} \leftarrow [\mathbf{Z}_0 \mid \mathbf{H}_0 \mid \mathbf{Z}_1 \mid \mathbf{H}_1 \mid \ldots \mid \mathbf{Z}_{t_\epsilon} \mid \mathbf{H}_{t_\epsilon}] \mathbf{W}_{t_\epsilon}^P$
28: **return** $\hat{\mathbf{Y}}_{t_\epsilon}^{SEED}$

---

---

**Algorithm 5** WOOD

**Input**:                                                      ▷ During inference, only $\mathbf{X}$ needs to be input
- Feature importance $\mathbf{FIM} \in [0,1]^M$
- Feature correlation magnitude $\mathbf{FCM} \in [0,1]^{M \times M}$  ⎫
- Size of the final evolutionary subspace of SEED $\chi_\epsilon$  ⎬ ▷ Evolutionary information from SEED
- Final outputs of SEED $\hat{\mathbf{Y}}_{t_\epsilon}^{SEED} \in \mathbb{R}^{N \times C}$  ⎭

- Raw features $\mathbf{X} \in \mathbb{R}^{N \times M}$
- Ground truth $\mathbf{Y} \in \{0,1\}^{N \times C}$                 ▷ One-hot encoding
- Codebook $\Omega \in \{-1,1\}^{C \times L}$
- Number of sparse times $n$
- Number of orthogonal times $m$
- Regularization coefficient $\lambda$

**Output**: Predicted logits $\hat{\hat{\mathbf{Y}}} \in \mathbb{R}^{N \times L}$                 ▷ ECOCs

1: **if** Training Stage **then**
2:    Define universal index set: $\boldsymbol{U} \leftarrow \{0,1,\ldots,M-1\}$
3:    Calculate global correlation sum:

$$\mathbf{GUM}[i] \leftarrow \sum \mathbf{FCM}[i,:] - 1, \ \forall \, i \in \boldsymbol{U}$$

4:    Calculate feature global scores:

$$\mathbf{GS}[i] \leftarrow \frac{\exp^{\frac{\mathbf{FIM}[i]}{\mathbf{GUM}[i]}}}{\sum_{j=0}^{M-1} \exp^{\frac{\mathbf{FIM}[j]}{\mathbf{GUM}[j]}}}, \ \forall \, i \in \boldsymbol{U}$$

5:    Subspace sampling:

$$\boldsymbol{S}_{pri} \leftarrow Random\,(range = \boldsymbol{U} \ , \ size = \chi_\epsilon \ , \ probability = \mathbf{GS}) \ , \ \boldsymbol{S}_{aux} \leftarrow \boldsymbol{U} \setminus \boldsymbol{S}_{pri} \quad ;$$
$$\mathbf{X}_{pri} \leftarrow \mathbf{X}[:,\boldsymbol{S}_{pri}] \quad , \quad \mathbf{X}_{aux} \leftarrow \mathbf{X}[:,\boldsymbol{S}_{aux}] \quad .$$

6:    Calculate sparse subspaces:

$$\mathbf{Z}_{pri} \leftarrow [\, \phi(\mathbf{X}_{pri}\mathbf{W}_{pri,0}^{ES} + \boldsymbol{\beta}_{pri,0}^{ES}) \mid \ldots \mid \phi(\mathbf{X}_{pri}\mathbf{W}_{pri,n-1}^{ES} + \boldsymbol{\beta}_{pri,n-1}^{ES}) \,]$$
$$\mathbf{Z}_{aux} \leftarrow [\, \phi(\mathbf{X}_{aux}\mathbf{W}_{aux,0}^{ES} + \boldsymbol{\beta}_{aux,0}^{ES}) \mid \ldots \mid \phi(\mathbf{X}_{aux}\mathbf{W}_{aux,n-1}^{ES} + \boldsymbol{\beta}_{aux,n-1}^{ES}) \,]$$

7:    Spatial fusion:

$$\mathbf{Z}_{fus} \leftarrow \mathbf{Z}_{pri} \cdot \sum \mathbf{GS}[\boldsymbol{S}_{pri}] + \mathbf{Z}_{aux} \cdot \sum \mathbf{GS}[\boldsymbol{S}_{aux}]$$

8:    Confidence scaling: $\forall i \in \{0,1,\ldots,N-1\}$ and $j \in \{0,1,\ldots,C-1\}$,
$$\dot{\mathbf{Y}}[i,j] \leftarrow \frac{\mathbf{Y}[i,j] \cdot \sum_{c=0}^{C-1} \exp^{\hat{\mathbf{Y}}_{t_\epsilon}^{SEED}[i,c]}}{\exp^{\hat{\mathbf{Y}}_{t_\epsilon}^{SEED}[i,j]}} \qquad\qquad\qquad ▷ \dot{\mathbf{Y}} \in \mathbb{R}^{N \times C}$$

9:    Spatial encoding: $\ddot{\mathbf{Y}} \leftarrow \dot{\mathbf{Y}} \times \Omega$                 ▷ $\ddot{\mathbf{Y}} \in \mathbb{R}^{N \times L}$
10:    Enhanced perceptron: $\hat{\hat{\mathbf{Y}}} \leftarrow \mathrm{EP}(\mathbf{Z}_{fus}, \mathbf{D}_0, \ddot{\mathbf{Y}}, m, \lambda)$
11: **else**

$$\begin{cases} \mathbf{X}_{pri} \leftarrow \mathbf{X}[:,\boldsymbol{S}_{pri}] \quad , \quad \mathbf{X}_{aux} \leftarrow \mathbf{X}[:,\boldsymbol{S}_{aux}] \\ \mathbf{Z}_{pri} \leftarrow [\, \phi(\mathbf{X}_{pri}\mathbf{W}_{pri,0}^{ES} + \boldsymbol{\beta}_{pri,0}^{ES}) \mid \ldots \mid \phi(\mathbf{X}_{pri}\mathbf{W}_{pri,n-1}^{ES} + \boldsymbol{\beta}_{pri,n-1}^{ES}) \,] \\ \mathbf{Z}_{aux} \leftarrow [\, \phi(\mathbf{X}_{aux}\mathbf{W}_{aux,0}^{ES} + \boldsymbol{\beta}_{aux,0}^{ES}) \mid \ldots \mid \phi(\mathbf{X}_{aux}\mathbf{W}_{aux,n-1}^{ES} + \boldsymbol{\beta}_{aux,n-1}^{ES}) \,] \\ \mathbf{Z}_{fus} \leftarrow \mathbf{Z}_{pri} \cdot \sum \mathbf{GS}[\boldsymbol{S}_{pri}] + \mathbf{Z}_{aux} \cdot \sum \mathbf{GS}[\boldsymbol{S}_{aux}] \\ \hat{\hat{\mathbf{Y}}} \leftarrow \mathrm{EP}(\mathbf{Z}_{fus}) \end{cases}$$

12: **end if**
13: **return** $\hat{\hat{\mathbf{Y}}}$

---

---

**Algorithm 6** Flame

---

**Input**: ▷ Executed on validation set

- Time step $t$
- Historical optimized target spaces $\tilde{\Omega}^{t-1} \in \{-1, 0, 1\}^{C \times tL}$ ▷ If $t = 0$, then $\tilde{\Omega}^{t-1} = \emptyset$
- Historical validation outputs $\hat{\tilde{\mathbf{Y}}}_{\text{val}}^{t-1} \in \mathbb{R}^{N_{\text{val}} \times tL}$ ▷ If $t = 0$, then $\hat{\tilde{\mathbf{Y}}}_{\text{val}}^{t-1} = \emptyset$
- Current target subspace $\Omega_t \in \{-1, 1\}^{C \times L}$
- Current validation outputs $\hat{\tilde{\mathbf{Y}}}_{\text{val}} \in \mathbb{R}^{N_{\text{val}} \times L}$
- Validation ground truth $\mathbf{Y}_{\text{val}} \in \{0, 1\}^{N_{\text{val}} \times C}$

**Output**: Optimized target space $\tilde{\Omega}_t \in \{-1, 0, 1\}^{C \times L}$

1: Initialize current round:
$$
\begin{cases}
\tilde{\Omega}^t \leftarrow \begin{cases} \Omega_t, \text{ if } t = 0 \\ [\tilde{\Omega}^{t-1} \mid \Omega_t], \text{ if } t \neq 0 \end{cases} \\
\hat{\tilde{\mathbf{Y}}}_{\text{val}}^t \leftarrow \begin{cases} \hat{\tilde{\mathbf{Y}}}_{\text{val}}, \text{ if } t = 0 \\ [\hat{\tilde{\mathbf{Y}}}_{\text{val}}^{t-1} \mid \hat{\tilde{\mathbf{Y}}}_{\text{val}}], \text{ if } t \neq 0 \end{cases} \\
\mathbf{S}_t \leftarrow \{j \in \mathbb{Z} \mid tL \leq j < (t+1)L\}
\end{cases}
$$

2: **while** $\mathbf{S}_t \neq \emptyset$ **do**

3:     **for** $j \in \mathbf{S}_t$ **do**

4:         Generate mask:
$$\mathbf{M}_j \leftarrow \{0, 1\}^{C \times (t+1)L}$$
$$\mathbf{M}_j[:, k] \leftarrow \begin{cases} 0, \text{ if } k = j \\ 1, \text{ if } k \neq j \end{cases}$$

5:         Calculate temperature:
$$\tau_j \leftarrow acc_{val}(\hat{\tilde{\mathbf{Y}}}_{\text{val}}^t \times (\tilde{\Omega}^t \odot \mathbf{M}_j)^\top) - acc_{val}(\hat{\tilde{\mathbf{Y}}}_{\text{val}}^t \times \tilde{\Omega}^{t^\top})$$

6:     **end for**

7:     Select the highest temperature index $\underset{r \in \mathbf{S}_t}{\arg\max} : \tau_r$

8:     **if** $\tau_r > 0$ **then**

9:         Update target space:
$$\tilde{\Omega}^t[:, r] \leftarrow 0$$

10:     Update the unburnt index set:
$$\mathbf{S}_t \leftarrow \mathbf{S}_t \setminus \{r\}$$

11:     **else**

12:         break         ▷ Terminate if no incompatible codevector exists

13:     **end if**

14: **end while**

15: **return** $\tilde{\Omega}^t \setminus \tilde{\Omega}^{t-1}$

---

---

**Algorithm 7** PATH

---

**Input:**                                                    ▷ During inference, only $\mathbf{X}$ needs to be input

•Feature importance $\mathbf{FIM} \in [0,1]^M$

•Feature correlation magnitude $\mathbf{FCM} \in [0,1]^{M \times M}$

•Size of the final evolutionary subspace of SEED $\chi_\epsilon$        ▷ Evolutionary information from SEED

•Final outputs of SEED $\hat{\mathbf{Y}}_{t_\epsilon}^{SEED} \in \mathbb{R}^{N \times C}$

• Raw features $\mathbf{X} \in \mathbb{R}^{N \times M}$

• Ground truth $\mathbf{Y} \in \{0,1\}^{N \times C}$                    ▷ One-hot encoding

• Number of sparse times $n$

• Number of orthogonal times $m$

• Regularization coefficient $\lambda$

• Convergence threshold $\epsilon$

• Validation features $\mathbf{X}_{\text{val}} \in \mathbb{R}^{N_{\text{val}} \times M}$

• Validation ground truth $\mathbf{Y}_{\text{val}} \in \{0,1\}^{N_{\text{val}} \times C}$

**Output:** Predicted logits $\hat{\mathbf{Y}}_{\text{PATH}}^{t_\epsilon} \in \mathbb{R}^{N \times C}$

1: **if** Training Stage **then**

2:    Generate $\mathbb{P}_\Omega \leftarrow \left\{ \Omega \in \{-1,1\}^{C \times L} \right\}$ via **_Dense_** ▷ An ECOC design algorithm in (Allwein et al., 2000)

3:    Initialize time step $t \leftarrow -1$

4:    **repeat**

5:       Increment time step $t \leftarrow t + 1$

6:       Randomly pull a codebook $\Omega_t \leftarrow \mathbb{P}_\Omega.\text{pop}()$

7:       Train a new base:

$$\text{WOOD}_t(\mathbf{FIM}, \mathbf{FCM}, \chi_\epsilon, \hat{\mathbf{Y}}_{t_\epsilon}^{SEED}, \mathbf{X}, \mathbf{Y}, \Omega_t, n, m, \lambda)$$

8:       Evaluate on validation set: $\hat{\tilde{\mathbf{Y}}}_{\text{val}} \leftarrow \text{WOOD}_t(\mathbf{X}_{\text{val}})$

9:       **if** $t = 0$ **then**

10:           Initialize $\begin{cases} \tilde{\Omega}^t \leftarrow \text{Flame}(t, \emptyset, \emptyset, \Omega_t, \hat{\tilde{\mathbf{Y}}}_{\text{val}}, \mathbf{Y}_{\text{val}}) \\ \hat{\tilde{\mathbf{Y}}}_{\text{val}}^t \leftarrow \hat{\tilde{\mathbf{Y}}}_{\text{val}} \end{cases}$

11:           **continue**

12:       **else**

13:           Update $\begin{cases} \tilde{\Omega}^t \leftarrow [\, \tilde{\Omega}^{t-1} \mid \text{Flame}(t, \tilde{\Omega}^{t-1}, \hat{\tilde{\mathbf{Y}}}_{\text{val}}^{t-1}, \Omega_t, \hat{\tilde{\mathbf{Y}}}_{\text{val}}, \mathbf{Y}_{\text{val}}) \,] \\ \hat{\tilde{\mathbf{Y}}}_{\text{val}}^t \leftarrow [\, \hat{\tilde{\mathbf{Y}}}_{\text{val}}^{t-1} \mid \hat{\tilde{\mathbf{Y}}}_{\text{val}}] \end{cases}$

14:       **end if**

15:       Calculate loss decline:

$$\Delta \mathcal{L}_t \leftarrow \mathcal{L}(\mathbf{Y}_{\text{val}}, \hat{\tilde{\mathbf{Y}}}_{\text{val}}^{t-1} \times \tilde{\Omega}^{t-1^\top}) - \mathcal{L}(\mathbf{Y}_{\text{val}}, \hat{\tilde{\mathbf{Y}}}_{\text{val}}^t \times \tilde{\Omega}^{t^\top})$$

▷ Cross-entropy loss adopted

16:    **until** $\frac{\Delta \mathcal{L}_t}{\mathcal{L}(\mathbf{Y}_{\text{val}}, \hat{\tilde{\mathbf{Y}}}_{\text{val}}^{t-1} \times \tilde{\Omega}^{t-1^\top})} < \epsilon$

17: **else**

18:    **for** $t \in \{0, 1, \ldots, t_\epsilon\}$ **do**

19:       $\hat{\tilde{\mathbf{Y}}}_t \leftarrow \text{WOOD}_t(\mathbf{X})$

20:    **end for**

21:    $\hat{\tilde{\mathbf{Y}}}^{t_\epsilon} \leftarrow [\hat{\tilde{\mathbf{Y}}}_0, \hat{\tilde{\mathbf{Y}}}_1, \ldots, \hat{\tilde{\mathbf{Y}}}_{t_\epsilon}]$

22: **end if**

23: Spatial decoding: $\hat{\mathbf{Y}}_{\text{PATH}}^{t_\epsilon} \leftarrow \hat{\tilde{\mathbf{Y}}}^{t_\epsilon} \times \tilde{\Omega}^{t_\epsilon^\top}$

24: **return** $\hat{\mathbf{Y}}_{\text{PATH}}^{t_\epsilon}$

---

---

**Algorithm 8** SPOT

---

**Input**:                                                     ▷ During inference, only $\mathbf{X}$ needs to be input

- Trained $SEED$ and $PATH$
- Raw features $\mathbf{X} \in \mathbb{R}^{N \times M}$
- Ground truth $\mathbf{Y} \in \{0, 1\}^{N \times C}$                                        ▷ One-hot encoding
- Validation features $\mathbf{X}_{\text{val}} \in \mathbb{R}^{N_{\text{val}} \times M}$
- Validation ground truth $\mathbf{Y}_{\text{val}} \in \{0, 1\}^{N_{\text{val}} \times C}$
- Number of sparse times $n$
- Number of orthogonal times $m$
- Regularization coefficient $\lambda$

**Output**: Predicted logits $\hat{\mathbf{Y}}_{\copyright} \in \mathbb{R}^{N \times C}$

1: **if** Training Stage **then**

2:     Branch-wise inference: $\begin{cases} \hat{\mathbf{Y}}_{t_\epsilon}^{SEED} \leftarrow SEED(\mathbf{X}) \\ \hat{\mathbf{Y}}_{PATH}^{t_\epsilon} \leftarrow PATH(\mathbf{X}) \\ \hat{\mathbf{Y}}_{\text{val}}^{SEED} \leftarrow SEED(\mathbf{X}_{\text{val}}) \\ \hat{\mathbf{Y}}_{\text{val}}^{PATH} \leftarrow PATH(\mathbf{X}_{\text{val}}) \end{cases}$

3:     Calculate concatenation weights:

$$\omega_{SEED} \leftarrow \frac{\mathcal{L}(\hat{\mathbf{Y}}_{\text{val}}^{PATH}, \mathbf{Y}_{\text{val}})}{\mathcal{L}(\hat{\mathbf{Y}}_{\text{val}}^{SEED}, \mathbf{Y}_{\text{val}}) + \mathcal{L}(\hat{\mathbf{Y}}_{\text{val}}^{PATH}, \mathbf{Y}_{\text{val}})} \quad , \quad \omega_{PATH} \leftarrow 1 - \omega_{SEED} \quad .$$

4:     Concatenate branch outputs:

$$\mathbf{X}_{\copyright} \leftarrow [\, \hat{\mathbf{Y}}_{t_\epsilon}^{SEED} \odot \omega_{SEED} \mid \hat{\mathbf{Y}}_{PATH}^{t_\epsilon} \odot \omega_{PATH} \,]$$

5:     Ensemble sparsification:

$$\mathbf{Z}_{\copyright} \leftarrow [\, \phi(\mathbf{X}_{\copyright}\mathbf{W}_{\copyright,0}^{ES} + \boldsymbol{\beta}_{\copyright,0}^{ES}) \mid \phi(\mathbf{X}_{\copyright}\mathbf{W}_{\copyright,1}^{ES} + \boldsymbol{\beta}_{\copyright,1}^{ES}) \mid \ldots \mid \phi(\mathbf{X}_{\copyright}\mathbf{W}_{\copyright,n-1}^{ES} + \boldsymbol{\beta}_{\copyright,n-1}^{ES}) \,]$$

6:     Enhanced perceptron:

$$\hat{\mathbf{Y}}_{\copyright} \leftarrow \text{EP}(\mathbf{Z}_{\copyright}, \mathbf{D}_0, \mathbf{Y}, m, \lambda)$$

7: **else**

8:     $\begin{cases} \hat{\mathbf{Y}}_{t_\epsilon}^{SEED} \leftarrow SEED(\mathbf{X}) \\ \hat{\mathbf{Y}}_{PATH}^{t_\epsilon} \leftarrow PATH(\mathbf{X}) \\ \mathbf{X}_{\copyright} \leftarrow [\, \hat{\mathbf{Y}}_{t_\epsilon}^{SEED} \odot \omega_{SEED} \mid \hat{\mathbf{Y}}_{PATH}^{t_\epsilon} \odot \omega_{PATH} \,] \\ \mathbf{Z}_{\copyright} \leftarrow [\, \phi(\mathbf{X}_{\copyright}\mathbf{W}_{\copyright,0}^{ES} + \boldsymbol{\beta}_{\copyright,0}^{ES}) \mid \phi(\mathbf{X}_{\copyright}\mathbf{W}_{\copyright,1}^{ES} + \boldsymbol{\beta}_{\copyright,1}^{ES}) \mid \ldots \mid \phi(\mathbf{X}_{\copyright}\mathbf{W}_{\copyright,n-1}^{ES} + \boldsymbol{\beta}_{\copyright,n-1}^{ES}) \,] \\ \hat{\mathbf{Y}}_{\copyright} \leftarrow \text{EP}(\mathbf{Z}_{\copyright}) \end{cases}$

9: **end if**

10: **return** $\hat{\mathbf{Y}}_{\copyright}$

---

