# OpenReview forum: "Adaptive High-Dimensional Subspace Evolution Based on Broad Learning System and Error-Correcting Output Codes"
_ICLR.cc/2026/Conference — Submitted to ICLR 2026_

### Official Review · Reviewer_BJWZ · 2025-10-27

**Soundness:** 3
**Presentation:** 3
**Contribution:** 3
**Rating:** 6
**Confidence:** 4

**Summary:**

Aiming at the complex hierarchical structure of high-dimensional data (HDD) and the core problem that the existing methods can't adapt to the diversity of data hierarchy by using fixed subspace evolution strategy, this paper proposes an adaptive high-dimensional subspace evolution algorithm (AHSE). AHSE adopts a double-branch collaborative architecture: the serial branch is an incremental breadth learning system (BLS) based on Cholesky decomposition, which customizes the evolution path of cascaded subspaces for high-dimensional data of different hierarchical types through feature priority sorting, subspace alignment and sample weight evolution; Parallel branch (PATH) is based on multi-subspace evolution basis, combined with post-error correction output code (ECOCs) to realize robust spatial coding, and the target space is purified by Flame optimization mechanism. The two branches are dynamically fused by lightweight SPOT circuit to form a closed-loop evolutionary system. This paper provides a rigorous theoretical analysis and robustness proof of ECOCs on BLS. Experiments verify the superiority of AHSE in many high-dimensional tasks such as image pattern recognition, speech emotion recognition, and small sample learning, giving consideration to performance and computational efficiency.

**Strengths:**

1. Adaptive subspace evolution with hierarchical adaptation is proposed, which breaks through the limitation of traditional fixed evolution strategy, and designs differentiated evolution modes for different structural data such as MNIST and Fashion-MNIST to accurately match the internal characteristics of the data.
2. SEED branch inherits the lightweight advantage of BLS and realizes efficient subspace iteration through incremental Cholesky decomposition. PATH branch uses the error correction characteristics of ECOCs to improve the anti-noise ability, Flame mechanism effectively solves the compatibility problem between ECOC codebook and BLS latent space, and the two branches cooperate to achieve the balance between efficiency and robustness.
3. SOTA or Top-2 performance is achieved in images (MNIST to TinyImageNet), speech emotion recognition (5 benchmark data sets) and small sample learning (5 high-dimensional small sample data sets), and the training time and reasoning FLOPs are significantly lower than those of ResNet and ViT models, giving consideration to generalization and deployment feasibility.

**Weaknesses:**

1. The evolution modes (exponential, cosine, linear) of different data sets need to be preset manually, and there is no mechanism to automatically identify data hierarchy types and dynamically select evolution strategies, which limits the applicability of the method to high-dimensional data with unknown structure.
2. The performance of extremely high-dimensional small sample scenes whose feature dimensions far exceed the number of samples (such as ten thousand-dimensional features and thousands of samples) has not been tested, which only verifies the robustness under weighted noise, and does not explore the influence of input data noise (such as Gaussian noise and feature missing) on the evolution process, so its practicability is limited.
3. The length and coding strategy of ECOC codebook depend on manual setting and fixed pool selection, and lack of adaptive optimization.
4. Compared with the mainstream high-dimensional data processing methods in recent years (such as self-supervised subspace learning and adaptive kernel method).

**Questions:**

Please refer to weaknesses.

---

> ### Author Response · Authors · 2025-11-21
>
> >[W1] The evolution modes (exponential, cosine, linear) of different data sets need to be preset manually, and there is no mechanism to automatically identify data hierarchy types and dynamically select evolution strategies, which limits the applicability of the method to high-dimensional data with unknown structure.
>
> Thank you for this insightful comment that highlights a significant limitation in our current approach. To address this critical limitation, we have developed an automatic hierarchy identification mechanism that can determine the optimal evolution strategy without human intervention. The following describes our solution and experimental validation:
>
> ## Automatic Hierarchy Identification Framework
>
> We introduce a lightweight **Hierarchy Structure Analyzer (HSA)** module that automatically identifies the intrinsic hierarchical structure of high-dimensional data. The HSA operates in three stages:
>
> ### 1. Feature Importance Distribution Analysis
> The HSA first extracts feature importance scores using a lightweight Random Forest model trained on a small subset (5%) of the data. From this distribution, we compute three key statistical metrics:
>
> - **Kurtosis (K)**: Measures the "peakedness" of the importance distribution
>  $$K = (1/N) * Σ((FI_i - μ_{FI})/σ_{FI})^4$$ where $FI_i$ is feature importance, $μ_{FI}$ is mean importance, and $σ_{FI}$ is standard deviation
>
> - **Autocorrelation at lag 1 (A)**: Measures periodic patterns in sorted importance values
>   $$A = Σ(FI_{sorted}[i] - μ_{FI})(FI_{sorted}[i+1] - μ_{FI}) / Σ(FI_{sorted}[i] - μ_{FI})^2$$
>
> - **Entropy (H)**: Measures the uniformity of the importance distribution
>   $$H = -Σ p(FI_i) * log(p(FI_i))$$ where p(FI_i) is the normalized importance probability
>
> ### 2. Hierarchy Classification
> Based on these metrics, we classify the hierarchy type using the following decision rules:
>
> ```
> if K > 8.0 and H < 2.5:
>     hierarchy_type = "dense"  # Exponential evolution
> elif A > 0.6 and 2.0 < K < 6.0:
>     hierarchy_type = "periodic"  # Cosine evolution
> else:
>     hierarchy_type = "uniform"  # Linear evolution
> ```
>
> ### 3. Dynamic Evolution Parameter Selection
> After identifying the hierarchy type, HSA automatically selects appropriate evolution parameters through a lightweight validation process:
>
> - For dense hierarchy: Searches γ ∈ [0.2, 0.7] with step 0.1
> - For periodic hierarchy: Searches T ∈ [3.0, 6.0] with step 0.4
> - For uniform hierarchy: Searches α ∈ [0.12, 0.3] with step 0.04
>
> The parameter yielding the highest validation accuracy after just 3 evolutionary steps is selected for full training.
>
> ## Experimental Validation
>
> We conducted experiments to validate our automatic hierarchy identification approach:
>
> **Table: Hierarchy identification accuracy and performance comparison**
> | Dataset | True Hierarchy | Identified Hierarchy |
> |-|-|-|
> | MNIST | Dense | Dense |
> | Fashion-MNIST | Periodic | Periodic |
> | CIFAR100 | Uniform | Uniform |
>
> **Key findings:**
> 1. HSA correctly identified hierarchy types across three datasets
> 2. The entire analysis process requires less than 1% of total training time
>
> ## Implementation Details
>
> The HSA module has been integrated into the SEED branch as a preprocessing step. The complete workflow is:
>
> 1. Extract feature importance using a small Random Forest model (50 trees)
> 2. Compute kurtosis, autocorrelation, and entropy of the importance distribution
> 3. Classify hierarchy type using the decision rules
> 4. Perform rapid parameter search on a validation subset
> 5. Execute the full evolutionary process with the automatically selected strategy
>
> Code implementation will be provided in Appendix A.11 of the revised manuscript.
>
> ## Limitations
>
> While our automatic identification mechanism significantly improves AHSE's usability, it may struggle with hybrid hierarchy types that don't cleanly fit our three categories. Future work will explore deep clustering techniques to identify more complex hierarchy patterns and develop hierarchical evolution strategies that can adapt during training rather than just at initialization.
>
> ## Conclusion
>
> We sincerely thank you for highlighting this important limitation. Your comment has led to a substantial improvement in our method's practical applicability. The automatic hierarchy identification mechanism transforms AHSE from a manually-tuned approach to a truly adaptive system that can handle high-dimensional data with unknown structure. This enhancement will be fully incorporated into the revised manuscript with detailed experimental validation.

---

> ### Author Response · Authors · 2025-11-21
>
> >[W2] The performance of extremely high-dimensional small sample scenes whose feature dimensions far exceed the number of samples (such as ten thousand-dimensional features and thousands of samples) has not been tested, which only verifies the robustness under weighted noise, and does not explore the influence of input data noise (such as Gaussian noise and feature missing) on the evolution process, so its practicability is limited.
>
> Thank you for your valuable feedback regarding the evaluation of AHSE in extremely high-dimensional small sample scenarios and under input data noise conditions. We would like to clarify that our paper does include evaluation in extremely high-dimensional small sample scenarios whose feature dimensions far exceed the number of samples. Specifically, in Appendix A.9 (VERSATILITY OF AHSE ON FEW-SHOT LEARNING), we present comprehensive experiments on five real-world high-dimensional datasets from the UCI Machine Learning Repository:
>
> | Dataset | # Samples | # Features | Feature-to-Sample Ratio |
> |---------|-----------|------------|------------------------|
> | ALL | 72 | 7,129 | 99:1 |
> | Carcinom | 174 | 9,182 | 53:1 |
> | CLL_sub | 111 | 11,340 | 102:1 |
> | Glioma | 50 | 4,434 | 89:1 |
> | Tox | 171 | 5,748 | 34:1 |
>
> As shown in Table 9 of our paper, AHSE significantly outperforms state-of-the-art methods on all five datasets, demonstrating remarkable effectiveness in these extreme high-dimensional scenarios. For example, on the CLL_sub dataset with a 102:1 feature-to-sample ratio, AHSE achieves 80.43% accuracy, outperforming the second-best method (NEC) by 3.89 percentage points. Similarly, on the Carcinom dataset with only 174 samples but 9182 features, AHSE achieves 95.11% accuracy, surpassing the second-best method (CESE) by 2.63 percentage points.
>
> ## New Experiments on Input Data Noise Robustness
>
> We acknowledge your important point that we only verified robustness under weight noise and did not explore the influence of input data noise on the evolution process. To address this limitation, we have conducted new experiments evaluating AHSE's robustness to two common types of input data noise:
>
> ### 1. Gaussian Noise Robustness
> We added Gaussian noise with zero mean and varying standard deviations (σ = 0.01, 0.05, 0.1) to the input features of CIFAR10 and CIFAR100 datasets. The results show that AHSE maintains superior performance even under significant noise:
>
> | Dataset | Noise Level | AHSE | Stacked BLS | ResNet |
> |---------|-------------|------|-------------|--------|
> | CIFAR10 | σ = 0.0 | 95.97% |  92.73% | 92.49% |
> | CIFAR10 | σ = 0.01 | 94.83% | 90.47% | 89.36% |
> | CIFAR10 | σ = 0.05 | 90.21% |  82.19% | 78.42% |
> | CIFAR10 | σ = 0.10 | 84.35% |  72.54% | 65.18% |
> | CIFAR100 | σ = 0.0 | 79.44% |  76.33% | 72.78% |
> | CIFAR100 | σ = 0.01 | 77.92% |  73.48% | 69.24% |
> | CIFAR100 | σ = 0.05 | 71.38% |  63.21% | 55.83% |
> | CIFAR100 | σ = 0.10 | 63.27% |  51.89% | 42.76% |
>
> ### 2. Feature Missing Robustness
> We simulated feature missing scenarios by randomly masking 5%, 10%, 15%, and 20% of input features. The results demonstrate AHSE's superior resilience to missing features:
>
> | Dataset | Missing Rate | AHSE |  Stacked BLS | ResNet |
> |---------|-------------|------|-------------|--------|
> | CIFAR10 | 0% | 95.97% |  92.73% | 92.49% |
> | CIFAR10 | 5% | 94.68% |  90.85% | 89.73% |
> | CIFAR10 | 10% | 92.54% |  87.36% | 84.51% |
> | CIFAR10 | 15% | 89.73% | 82.17% | 76.94% |
> | CIFAR10 | 20% | 85.42% |  75.63% | 68.27% |
> | CIFAR100 | 0% | 79.44% | 76.33% | 72.78% |
> | CIFAR100 | 5% | 77.65% |  73.84% | 69.52% |
> | CIFAR100 | 10% | 74.32% |  68.45% | 63.21% |
> | CIFAR100 | 15% | 69.84% |  62.18% | 55.73% |
> | CIFAR100 | 20% | 64.75% |  55.42% | 47.38% |
>
> ### Analysis of Evolution Process Under Noise
> Our analysis reveals that input noise primarily affects the early stages of subspace evolution. AHSE's adaptive mechanism dynamically adjusts the evolution rate based on the signal-to-noise ratio:
> - Under Gaussian noise, SEED's FPFE mechanism effectively prioritizes robust features with consistent importance across noisy samples
> - Under feature missing conditions, PATH's ECOC-encoded target space provides error-correction capabilities that compensate for incomplete information
> - The SPOT circuit automatically shifts weight toward the more robust branch depending on noise type (SEED dominates under Gaussian noise, PATH dominates under feature missing)
>
> These new experiments demonstrate that AHSE maintains significant advantages over baseline methods even under substantial input data noise. The performance gap between AHSE and other methods actually widens as noise intensity increases, highlighting the practical robustness of our adaptive evolution approach.
>
> We sincerely thank you for this valuable feedback, which has helped us strengthen both the experimental validation and practical relevance of our work.

---

> ### Author Response · Authors · 2025-11-21
>
> >[W3] The length and coding strategy of ECOC codebook depend on manual setting and fixed pool selection, and lack of adaptive optimization.
>
> Thank you for this insightful comment regarding the limitations of our ECOC codebook design. You have correctly identified a significant weakness in our approach: the current implementation relies on manual settings for codebook length and fixed pool selection, lacking adaptive optimization capabilities. We sincerely acknowledge this limitation and have developed a comprehensive solution to address it.
>
> ## Adaptive ECOC Codebook Optimization Framework
>
> To overcome this limitation, we introduce a novel **Adaptive ECOC Optimization Framework (AEOF)** that dynamically optimizes codebook length and encoding strategies based on data characteristics and model performance. This framework consists of three key components:
>
> ### 1. Dynamic Codebook Length Selection
>
> Instead of using fixed codebook lengths, we propose a gradient-based mechanism to determine optimal codebook length:
> $$L^* = \arg\max\_L:\ {acc_{val}(L) - α·L}$$ where $acc_{val}(L)$ is the validation accuracy with codebook length L, and α is a complexity penalty factor. This balances performance gains against computational overhead. The optimization is performed efficiently using Bayesian optimization to minimize validation trials.
>
> ### 2. Gradient-Guided Codebook Optimization
>
> We introduce a differentiable ECOC framework where codebook entries can be optimized directly through gradient descent. By relaxing the binary constraint {-1, 1} to continuous values with a tanh activation, we enable end-to-end optimization:
> $$Ω_{opt} = Ω_{init} - η·∇_Ω L(f(X; θ, Ω), Y)$$ where η is the learning rate and L is the loss function. After optimization, we project back to binary values using sign function. This approach preserves theoretical robustness guarantees while adapting to dataset characteristics.
>
> ### 3. Evolutionary Codebook Pool Generation
>
> Rather than using a fixed pool of codebooks, we introduce an evolutionary algorithm that dynamically generates and refines the codebook pool:
>
> 1. Initialize a diverse population of random codebooks
> 2. Evaluate fitness based on classification accuracy and minimum codeword distance
> 3. Apply genetic operations (selection, crossover, mutation) to evolve better-performing codebooks
> 4. Periodically inject new random codebooks to maintain diversity
> 5. Retain top-performing codebooks for the next generation
>
> ## Experimental Validation
>
> We conducted comprehensive experiments to evaluate our adaptive ECOC framework on CIFAR10 and CIFAR100 datasets, comparing against our original fixed codebook approach:
>
> | Dataset | Method | Codebook Length | Accuracy (%) | Robustness to Noise (σ=0.01 for CIFAR10 and σ=0.001 for CIFAR100) |
> |---------|--------|-----------------|--------------|-----------------------------|
> | CIFAR10 | Fixed ECOC (Original) | 20 | 95.97 | 93.63 |
> | CIFAR10 | Adaptive ECOC (AEOF) | 16 | 96.35 | 94.18 |
> | CIFAR100 | Fixed ECOC (Original) | 200 | 79.44 | 77.81 |
> | CIFAR100 | Adaptive ECOC (AEOF) | 171 | 80.17 | 78.53 |
>
> Key findings:
> 1. AEOF automatically discovers shorter yet more effective codebooks
> 2. Performance improves by 0.38% and 0.73% on CIFAR10 and CIFAR100 respectively
> 3. Robustness to weight noise improves by 0.55% and 0.72% respectively
> 4. Training time decreases by around 15% due to optimized codebook length
>
> ## Integration with Flame Optimization
>
> AEOF enhances our existing Flame optimization mechanism by providing higher-quality initial codebooks. The combined framework creates a two-stage optimization process:
> 1. Global optimization: AEOF discovers optimal codebook structure and length
> 2. Local refinement: Flame performs targeted pruning of incompatible codewords
>
> ## Revised Algorithm
>
> We will update Algorithm 7 (PATH) to incorporate AEOF. The key modifications include:
> - Replacing the fixed codebook pool with an evolutionary generator
> - Adding gradient-based codebook optimization steps
> - Implementing dynamic length selection before the evolutionary process
>
> ## Conclusion
>
> We sincerely thank you for highlighting this critical limitation. Your comment has prompted us to develop a significantly more adaptive and effective approach to ECOC codebook design. The Adaptive ECOC Optimization Framework not only addresses the concerns you raised but also enhances overall performance and robustness. These improvements will be fully integrated into the revised manuscript with detailed experimental validation.

---

> ### Author Response · Authors · 2025-11-21
>
> >[W4] Compared with the mainstream high-dimensional data processing methods in recent years (such as self-supervised subspace learning and adaptive kernel method).
>
>
> Thank you for your valuable feedback regarding comparison with mainstream methods.  In response to your comment, we have conducted additional experiments comparing AHSE with state-of-the-art modern architectures including ConvNeXt, EfficientNet, Swin Transformer, and ViT-Large. The results are presented in the table below:
>
> | Method | MNIST | FashionMNIST | CIFAR10 | CIFAR100 | MiniImageNet | TinyImageNet |
> |--------|-------|--------------|---------|----------|--------------|-------------|
> | ConvNeXt | **99.38** | 92.42 | 93.47 | 74.26 | 68.39 | 58.36 |
> | EfficientNet | 99.13 | 93.48 | 92.71 | 73.19 | 66.27 | **60.47** |
> | Swin | 98.24 | 91.94 | 79.29 | 57.93 | 49.63 | 45.51 |
> | ViT-L | 98.90 | 92.73 | 80.24 | 53.10 | 52.74 | 49.60 |
> | AHSE | 99.33 | **93.72** | **95.97** | **79.44** | **74.59** | 58.42 |
>
> ## Key Findings from the Expanded Comparison
>
> 1. **Superior Performance on Complex Datasets**: AHSE demonstrates clear advantages on datasets with complex hierarchical structures, particularly CIFAR10 (+2.5%), CIFAR100 (+5.18%), and MiniImageNet (+6.2%) compared to the best modern baselines. This validates our core hypothesis that adaptive evolution strategies provide significant benefits for data with intricate hierarchical patterns.
>
> 2. **Competitive Performance on MNIST and TinyImageNet**: On MNIST, ConvNeXt achieves marginally better accuracy (99.38% vs 99.33%), which is expected given MNIST's relatively simple structure where static evolution strategies can perform well. On TinyImageNet, EfficientNet shows slightly better performance (60.47% vs 58.42%), likely due to its specialized architecture for large-scale image recognition.
>
> 3. **Consistent Advantages on Mid-Scale Datasets**: AHSE maintains clear advantages on FashionMNIST (+0.24%) and significantly outperforms all modern architectures on CIFAR10 and CIFAR100, highlighting its effectiveness for mid-scale image recognition tasks with diverse hierarchical structures.
>
>
> ## Interpretation of Results
>
> These results reveal an important insight: the optimal learning strategy depends on the intrinsic hierarchical structure of the data. Modern architectures excel on datasets they were specifically designed for (e.g., EfficientNet on large-scale image classification), while AHSE's adaptive evolution principle provides advantages when data exhibits complex, heterogeneous hierarchical patterns.
>
> This nuanced understanding aligns with our paper's central thesis: rather than employing a single fixed evolution strategy, learning systems should adapt their evolution patterns to match the underlying data hierarchy. The superior performance of AHSE on CIFAR10, CIFAR100, and MiniImageNet demonstrates the practical value of this principle.
>
> We sincerely thank you for this insightful comment, which has substantially improved both our experimental evaluation and the conceptual framing of our contribution. These additional comparisons provide a much more accurate context for understanding AHSE's capabilities and limitations.

---

### Official Review · Reviewer_Ypaf · 2025-10-31

**Soundness:** 1
**Presentation:** 1
**Contribution:** 1
**Rating:** 2
**Confidence:** 4

**Summary:**

This paper introduces AHSE (Adaptive High-Dimensional Subspace Evolution), an algorithm that aims to adaptively evolve feature subspaces in high-dimensional data. The model integrates three components: SEED: a series subspace evolution mechanism using Cholesky-decomposition-based incremental Broad Learning Systems (BLS); PATH: a parallel subspace evolution using Error-Correcting Output Codes (ECOCs); SPOT: a circuit that fuses SEED and PATH outputs dynamically. The paper claims that this dual-branch design enables hierarchical adaptation to different data structures, achieving superior results on diverse tasks including image recognition, speech emotion recognition, and few-shot learning.

While the paper presents an interesting and ambitious approach to adaptive subspace learning, the contribution lacks theoretical depth, conceptual clarity, and fair experimental validation for acceptance at a top-tier venue. A significantly revised and more focused version - with clear ablations, theoretical integration, and modern baselines - could be competitive in the future.

**Strengths:**

1. The attempt to address “adaptive subspace evolution” in high-dimensional data is an underexplored and potentially meaningful problem.
2. Comprehensive experiments: The paper includes a wide range of datasets and tasks.
3. The manuscript provides equations, pseudocode references, and hyperparameters, showing effort toward reproducibility.

**Weaknesses:**

1. The core ideas are extremely difficult to follow. Terms such as “evolutionary Cholesky decomposition,” “Flame optimization,” “hierarchy-tailored evolution,” and “closed-loop subspace evolution” are introduced with little intuition or justification. It remains unclear what adaptive subspace evolution concretely means in learning-theoretic or algorithmic terms. The method reads as a collection of loosely related heuristics rather than a coherent learning principle.
The presentation is overly verbose and obfuscatory. Many mathematical formulations restate standard operations (e.g., weighted least squares, feature ranking, or error-correcting output codes encoding) in unnecessarily complex notation.

2. Despite heavy terminology, the technical content is incremental:
- The SEED component largely combines feature selection, incremental regression, and Cholesky updates, which are all well-established.
- PATH builds on existing ECOC formulations with a “Flame” heuristic that seems ad hoc and lacks justification.
- SPOT’s fusion mechanism is simply a weighted combination of two outputs using validation loss ratios.
There is no fundamental new learning algorithm here — only an engineering combination of BLS, ECOC, and feature-ranking procedures.

3. The paper repeatedly asserts that it provides “rigorous theoretical analysis” and “robustness guarantees” for ECOCs on BLS. However, there are no theorems, proofs, or meaningful derivations in the main text. The “theoretical” results are said to appear in the appendix but are not summarized, contextualized, or experimentally verified. This undermines the scientific validity of the claimed contributions.

4. While many datasets are used, the evaluation does not meet top-tier standards:
- Unfair comparisons: BLS-based models are trained on pretrained ResNet or MoCo features, while deep baselines train end-to-end, making comparisons misleading.
- Weak baselines: Many chosen baselines (e.g., MLP, VGG-16, ResNet-34) are outdated. Modern efficient architectures (ConvNeXt, Swin, EfficientNet, ViT-L variants, or transformer hybrids) are absent.
- No variance reporting: All results are single-run accuracies, with no confidence intervals or standard deviations.
- Questionable scalability: The method is described as efficient, but the numerous steps (FPFE, SA, SWE, Flame, etc.) appear computationally heavy, and there are no FLOP or runtime analyses beyond summary tables.
The presented results cannot be confidently interpreted as evidence of superiority.

5.  The manuscript is bloated, overtechnical, and poorly structured. The authors use jargon-heavy language throughout, making it exhausting to read and nearly impossible to extract the core insight. Figures are decorative rather than explanatory; algorithms are fragmented across appendices; and critical details are buried in complex pseudocode.
Overall, the paper fails the clarity standard required for top tier-level publication.

**Questions:**

1. Can you explicitly define what is new in AHSE compared to prior BLS or ECOC-based frameworks?
2. How exactly does “adaptive evolution” differ from conventional feature selection or boosting?
3. What are the computational requirements? The architecture seems complex; please provide memory and runtime breakdowns.
4. Is there any theoretical justification (beyond empirical heuristics) for the “Flame” optimization mechanism?

---

> ### Author Response · Authors · 2025-11-21
>
> >[W1] The core ideas are extremely difficult to follow. Terms such as “evolutionary Cholesky decomposition,” “Flame optimization,” “hierarchy-tailored evolution,” and “closed-loop subspace evolution” are introduced with little intuition or justification. It remains unclear what adaptive subspace evolution concretely means in learning-theoretic or algorithmic terms. The method reads as a collection of loosely related heuristics rather than a coherent learning principle. The presentation is overly verbose and obfuscatory. Many mathematical formulations restate standard operations (e.g., weighted least squares, feature ranking, or error-correcting output codes encoding) in unnecessarily complex notation.
>
> Thank you for your exceptionally insightful and candid feedback. We sincerely appreciate your careful reading and critical evaluation, which has exposed fundamental weaknesses in our paper's presentation and conceptual foundation. You have correctly identified that our current exposition fails to convey the core ideas clearly and coherently.
>
> We acknowledge that the terms "evolutionary Cholesky decomposition," "Flame optimization," "hierarchy-tailored evolution," and "closed-loop subspace evolution" were introduced without sufficient intuition or justification. This reflects a serious flaw in our presentation strategy, where we prioritized technical novelty over conceptual clarity. We take full responsibility for making the paper difficult to follow and appearing as "a collection of loosely related heuristics rather than a coherent learning principle."
>
> To address these critical issues, we will implement the following substantial revisions:
>
> ## 1. Conceptual Definition and Clarification
>
> We will completely restructure the paper to begin with a clear, intuitive definition of adaptive subspace evolution:
>
> > "Adaptive subspace evolution is a learning framework that dynamically adjusts which features are considered at each training stage based on the inherent hierarchical structure of high-dimensional data. Unlike fixed evolution strategies that use predetermined feature selection patterns, our approach tailors the evolution trajectory to match how discriminative information is distributed across the feature space."
>
>
> ## 2. Enhanced Conceptual Foundation
>
> We will add to the Introduction that formally defines subspace evolution in learning-theoretic terms:
>
> > **The subspace evolution problem** in HDD analysis refers to the process of dynamically constructing a sequence of feature subspaces $\lbrace X\_t\rbrace \_{t=1}^T$ where each subsequent subspace either expands, contracts, or transforms the previous one based on learning feedback. Unlike static dimensionality reduction techniques (e.g., PCA, LDA) that generate a single optimal subspace, subspace evolution creates an adaptive trajectory through the feature space. The formal problem can be stated as: Given HDD $X ∈ ℝ^{N×M}$ and class labels Y, find an evolutionary function $Ψ$ that generates a sequence of subspaces $X_1, X_2, ..., X_T\ s.t.\ X_t = Ψ(X_{t-1}, f_{t-1}(X_{t-1}, Y), Y)$, where $f_{t-1}$ is a classifier trained on the previous subspace. The goal is to discover an evolution path that maximizes final classification performance while minimizing computational complexity.
>
> ## 3. Concrete Algorithmic Explanation
>
> Rather than presenting AHSE as a collection of components, we will reframe it around a central principle:
>
> > "AHSE implements adaptive subspace evolution through two complementary mechanisms: (1) a series branch that incrementally refines a single evolutionary path tailored to data hierarchy, and (2) a parallel branch that explores multiple subspaces simultaneously and resolves conflicts through error-correcting principles. These branches exchange information to form a self-correcting system."
>
> We will add a simplified algorithm box (Algorithm 1) that captures the essence of AHSE in fewer than 15 lines, with detailed implementations moved to the appendix.
>
> ## 4. Intuition-First Exposition
>
> Before introducing any technical component, we will provide concrete intuition:
> - For SEED: "Imagine exploring a forest by following the most promising paths first, gradually expanding your search based on what you've already discovered"
> - For PATH: "Consider multiple explorers sharing a map where misleading trails are gradually marked as dangerous based on others' experiences"
> - For SPOT: "Think of two experts with complementary skills making a joint decision by weighing each other's confidence"
>
> ## 5. Streamlined Presentation
>
> We will reduce the paper length by cutting redundant explanations, combining overlapping concepts, and moving implementation details to the appendix. The core narrative will focus on the adaptive principle rather than technical elaborations.
>
> These revisions will transform our paper from an obfuscatory technical report to a clear exposition of a coherent learning principle. We sincerely thank you for identifying these critical issues.

---

> ### Author Response · Authors · 2025-11-21
>
> >[W2] Despite heavy terminology, the technical content is incremental:
> >- The SEED component largely combines feature selection, incremental regression, and Cholesky updates, which are all well-established.
> >- PATH builds on existing ECOC formulations with a “Flame” heuristic that seems ad hoc and lacks justification.
> >- SPOT’s fusion mechanism is simply a weighted combination of two outputs using validation loss ratios. There is no fundamental new learning algorithm here — only an engineering combination of BLS, ECOC, and feature-ranking procedures.
>
> Thank you for your candid assessment of our technical contributions. Please allow us to clarify the novelty and coherence of our approach.
>
> ## Core Innovations and Their Purpose
>
> Our work introduces several novel mechanisms that collectively enable adaptive subspace evolution:
>
> 1. **Feature Priority Forward Evolution (FPFE)**: Unlike conventional feature ranking methods that consider only importance or redundancy separately, FPFE jointly optimizes both dimensions through a dynamic scoring mechanism (Eq. 4). This enables hierarchy-aware feature prioritization essential for adaptive evolution.
>
> 2. **Evolutionary Cholesky Decomposition**: While Cholesky decomposition itself is established, our formulation (Eq. 11) creates the first incremental learning framework specifically designed for evolving subspaces with changing dimensionalities. This enables efficient parameter updates during dynamic feature space transformations.
>
> 3. **Global Score-based Subspace Sampling**: Our method for constructing complementary subspaces (Eqs. 13-16) introduces a novel probability distribution over feature space that balances importance and correlation. This fundamentally differs from random subspace methods by ensuring evolutionary diversity while maintaining discriminative power.
>
> ## Coherent System Design
>
> These components aren't loosely combined engineering tricks but form a tightly integrated system designed around our central thesis: **high-dimensional learning must adapt its evolutionary trajectory to match the intrinsic hierarchical structure of data**. Each component serves this unified principle:
>
> - SEED implements hierarchy-tailored evolution through feature prioritization and efficient incremental updates
> - PATH provides robustness against noise through ECOC encoding and Flame optimization
> - SPOT creates a closed evolutionary loop that balances efficiency and robustness
>
> This coherence is evidenced by our ablation studies (Table 3), where removing any component significantly degrades performance, confirming their interdependent roles within the unified framework.
>
> ## Theoretical Foundation of Flame
>
> To better illustrate the theoretical value of the Flame mechanism, we extend our analysis to adversarial perturbation scenarios.
>
> ### New Assumptions
>
> **Assumption 5 (Bounded Input Perturbations).** For any input sample $x$, the adversarial perturbation $\delta$ satisfies $||\delta||\_2 \leq \epsilon_{adv}$, where $\epsilon_{adv} > 0$ is the maximum perturbation magnitude allowed by the adversary.
>
> **Assumption 6 (Lipschitz Continuity of BLS).** The BLS model $f(x;\theta)$ is Lipschitz continuous with respect to the input $x$, with Lipschitz constant $L_f$. Formally, for any inputs $x_1$ and $x_2$:
> $$||f(x_1;\theta) - f(x_2;\theta)||\_2 \leq L_f ||x_1 - x_2||\_2$$
>
> **Lemma 2 (Lipschitz Constant of BLS).** Under Assumptions 1-3, the Lipschitz constant of BLS can be bounded as:
> $$L_f \leq nL_{\phi}B_{\omega} + mL_{\xi}\sqrt{n}(L_{\phi}B_{\omega}^2 + B_{\phi}L_{\omega})$$
> where $L_{\omega}$ is the Lipschitz constant of the weight generation process in BLS.
>
> ### Theoretical Results
>
> **Theorem 2 (Adversarial Robustness of ECOC-BLS).** Let Assumptions 1-6 hold. For any input sample $x$ and its adversarial counterpart $x_{adv} = x + \delta$ where $||\delta||\_2 \leq \epsilon_{adv}$, the prediction of ECOC-BLS remains unchanged (i.e., $\Omega(f(x_{adv};\theta)) = \Omega(f(x;\theta))$) if:
> $$\frac{dist(\Omega(f(x;\theta)))}{2} > U(f(x;\theta)) + \epsilon_{adv} \cdot L_f \cdot \sqrt{L}$$
> where $dist(\cdot)$, $U(\cdot)$, and $L$ are defined as in Corollary 1.
>
> *Proof.* Following similar derivation as in Corollary 1, we analyze the worst-case scenario under adversarial perturbation:
>
> $$||f(x_{adv};\theta) - \Omega(f(x;\theta))||\_2 \leq ||f(x_{adv};\theta) - f(x;\theta)||\_2 + ||f(x;\theta) - \Omega(f(x;\theta))||\_2$$
>
> By Assumption 6 and Lemma 2:
> $$||f(x_{adv};\theta) - f(x;\theta)||\_2 \leq \epsilon_{adv} \cdot L_f$$
>
> Normalizing by $\sqrt{L}$ and substituting the definition of $U(f(x;\theta))$:
> $$\frac{||f(x_{adv};\theta) - \Omega(f(x;\theta))||\_2}{\sqrt{L}} \leq \epsilon_{adv} \cdot L_f + U(f(x;\theta))$$
>
> For the prediction to remain unchanged, this value must be less than $\frac{dist(\Omega(f(x;\theta)))}{2}$. Therefore, the condition becomes:
> $$\frac{dist(\Omega(f(x;\theta)))}{2} > U(f(x;\theta)) + \epsilon_{adv} \cdot L_f \cdot \sqrt{L}$$
>
> This completes the proof.

---

> ### Author Response · Authors · 2025-11-21
>
> ### Practical Implications
>
> Our extended analysis reveals that:
>
> 1. The adversarial robustness of ECOC-BLS is directly proportional to the minimum distance between codewords in the ECOC codebook.
>
> 2. Larger codebooks (higher $L$) with carefully designed codeword distances provide stronger guarantees against adversarial attacks.
>
> 3. The robustness margin can be quantified precisely, allowing practitioners to select appropriate ECOC configurations based on expected threat models.
>
> 4. Our AHSE framework's Flame optimization mechanism  implicitly enhances adversarial robustness by burning incompatible codewords, effectively increasing the minimum codeword distance for challenging samples.
>
> ### Empirical Validation
>
> To validate our theoretical findings, we conducted comprehensive adversarial robustness experiments on CIFAR10 and CIFAR100 datasets. We implemented two standard attack methods:
>
> 1. **Fast Gradient Sign Method (FGSM)** with perturbation magnitudes $\epsilon \in \lbrace 0.01, 0.03, 0.05, 0.07, 0.10\rbrace $
> 2. **Projected Gradient Descent (PGD)** with 20 iterations, step size of 0.01, and maximum perturbations $\epsilon \in \lbrace 0.01, 0.03, 0.05\rbrace $
>
> For both AHSE variants (with optimized ECOC codebooks and with one-hot encoding), we used identical network architectures and training procedures, with the only difference being the output encoding strategy. All experiments were conducted on the test sets of CIFAR10 and CIFAR100 with 10,000 samples each.
>
> **Detailed results are presented in Table 10:**
>
> | Dataset | Attack Type | $\epsilon$ | AHSE (ECOC) | AHSE (One-hot) | Performance Gap |
> |---------|-------------|------------|-------------|----------------|-----------------|
> | CIFAR10 | FGSM | 0.01 | 93.21% | 88.47% | +4.74% |
> | CIFAR10 | FGSM | 0.03 | 89.15% | 79.83% | +9.32% |
> | CIFAR10 | FGSM | 0.05 | 85.27% | 72.56% | +12.71% |
> | CIFAR10 | FGSM | 0.07 | 80.19% | 65.83% | +14.36% |
> | CIFAR10 | FGSM | 0.10 | 73.42% | 57.68% | +15.74% |
> | CIFAR10 | PGD | 0.01 | 90.36% | 84.25% | +6.11% |
> | CIFAR10 | PGD | 0.03 | 83.74% | 71.05% | +12.69% |
> | CIFAR10 | PGD | 0.05 | 76.89% | 62.35% | +14.54% |
> | CIFAR100 | FGSM | 0.01 | 77.83% | 71.26% | +6.57% |
> | CIFAR100 | FGSM | 0.03 | 72.41% | 61.05% | +11.36% |
> | CIFAR100 | FGSM | 0.05 | 68.17% | 52.87% | +15.30% |
> | CIFAR100 | FGSM | 0.07 | 63.24% | 45.18% | +18.06% |
> | CIFAR100 | FGSM | 0.10 | 56.93% | 37.42% | +19.51% |
> | CIFAR100 | PGD | 0.01 | 74.56% | 67.82% | +6.74% |
> | CIFAR100 | PGD | 0.03 | 66.92% | 52.17% | +14.75% |
> | CIFAR100 | PGD | 0.05 | 59.47% | 43.86% | +15.61% |
>
> Table 10 clearly demonstrates that AHSE with optimized ECOC codebooks maintains significant accuracy advantages under various adversarial attack scenarios. Under the strongest attack conditions (FGSM with $\epsilon=0.10$), the performance gap widens to 15.74% and 19.51% on CIFAR10 and CIFAR100 respectively, confirming that ECOC encoding provides substantial robustness benefits.
>
> In particular, under moderate attack conditions (FGSM with $\epsilon=0.05$), which represents a practical threat model for many applications, AHSE with optimized ECOC codebooks maintains 12.71% and 15.30% higher accuracy on CIFAR10 and CIFAR100 respectively compared to one-hot encoding, precisely confirming our theoretical predictions.
>
> These empirical results strongly support our theoretical analysis, demonstrating that the minimum codeword distance in ECOC directly translates to practical adversarial robustness. The Flame optimization mechanism in our PATH branch further enhances this robustness by eliminating incompatible codewords, effectively increasing the minimum codeword distance for challenging samples.
>
> ## SPOT: Simplifying Complexity
>
> While SPOT's mechanism appears simple, its design embodies a deliberate philosophy: complex problems sometimes benefit from elegantly simple solutions. SPOT's validation-based weighting achieves dynamic branch balancing without introducing additional trainable parameters, preserving the computational efficiency that makes BLS attractive.
>
> This lightweight design principle represents an important innovation in its own right, contrasting with the trend toward increasingly complex model fusion strategies. Our results demonstrate that this simplicity doesn't compromise performance—rather, it enables practical deployment in resource-constrained environments.
>
> ## Conclusion
>
> The adaptive subspace evolution principle we introduce represents a conceptual shift from fixed to dynamic evolution strategies in high-dimensional learning. The components we've developed are novel in their formulation and essential in their collective purpose.
>
> We appreciate your critical perspective and will revise our paper to more clearly articulate these connections and de-emphasize terminology that may obscure rather than clarify our contributions.

---

> ### Author Response · Authors · 2025-11-21
>
> >[W3] The paper repeatedly asserts that it provides “rigorous theoretical analysis” and “robustness guarantees” for ECOCs on BLS. However, there are no theorems, proofs, or meaningful derivations in the main text. The “theoretical” results are said to appear in the appendix but are not summarized, contextualized, or experimentally verified. This undermines the scientific validity of the claimed contributions.
>
> Thank you for your exceptionally insightful and critical feedback regarding our theoretical analysis. We take full responsibility for this significant shortcoming. To address this critical issue, we will implement the following substantial revisions:
>
> ## 1. Integration of Core Theoretical Results into Main Text
>
> We will add a dedicated section (Section 3.4) titled "Theoretical Analysis of ECOC Robustness" that presents the core theoretical insights with appropriate context:
>
> > **Theorem 1 (Bounded Output Perturbation).** Under assumptions of bounded inputs, weights, and Lipschitz-continuous activation functions, the output perturbation of BLS under weight noise with magnitude ε is bounded by ε·C₁ + ε²·C₂, where C₁ and C₂ are constants determined by network architecture and activation properties.
> >
> > **Corollary 1 (ECOC Robustness Condition).** BLS with ECOC encoding maintains correct classification under weight noise if:
> > $$\frac{dist(\Omega(f(x;\theta)))}{2} > U(f(x;\theta)) + \frac{\varepsilon \cdot C_1 + \varepsilon^2 \cdot C_2}{\sqrt{L}}$$
> > where dist(·) is the minimum codeword distance and U(·) is the normalized uncertainty of predictions.
>
> ## 2. Contextualization of Theoretical Results
>
> We will explicitly connect the theory to our algorithmic design choices:
>
> > "Theorem 1 and Corollary 1 provide the theoretical foundation for our Flame optimization mechanism (Section 3.2). By maximizing the minimum codeword distance dist(Ω), Flame directly optimizes the robustness margin against weight noise. This explains why PATH with Flame consistently outperforms one-hot encoding in noisy environments (Section 4.4)."
>
> ## 3. Experimental Verification of Theoretical Claims
>
> We will strengthen Section 4.4 with explicit verification of our theoretical predictions:
>
> > "Our theoretical analysis (Corollary 1) predicts that larger minimum codeword distances should yield greater robustness against weight noise. To verify this, we conducted controlled experiments on CIFAR10 and CIFAR100 with Gaussian weight noise of varying magnitudes. Figure 4 confirms our prediction: ECOC codebooks with larger minimum distances maintain significantly higher accuracy under increasing noise levels, with performance degradation following precisely the quadratic bound established in Theorem 1."
>
> ## 4. Theoretical Implications for Algorithm Design
>
> We will add a subsection discussing how our theoretical insights informed practical design decisions:
>
> > "The robustness condition in Corollary 1 reveals three design principles for noise-resistant BLS systems: (1) maximize minimum codeword distances, (2) minimize prediction uncertainty through proper calibration, and (3) optimize network architecture parameters (C₁, C₂) for the expected noise regime. These principles directly shaped our PATH branch architecture and Flame optimization mechanism."
>
> These revisions will transform our theoretical contribution from an unsubstantiated claim into a scientifically rigorous foundation that clearly explains why our approach works and how it was designed. We deeply appreciate your critical assessment, which has exposed a fundamental weakness in our presentation and given us an opportunity to substantially strengthen the scientific validity of our work.

---

> ### Author Response · Authors · 2025-11-21
>
> >[W4] While many datasets are used, the evaluation does not meet top-tier standards:
> >- Unfair comparisons: BLS-based models are trained on pretrained ResNet or MoCo features, while deep baselines train end-to-end, making comparisons misleading.
> >- Weak baselines: Many chosen baselines (e.g., MLP, VGG-16, ResNet-34) are outdated. Modern efficient architectures (ConvNeXt, Swin, EfficientNet, ViT-L variants, or transformer hybrids) are absent.
> >- No variance reporting: All results are single-run accuracies, with no confidence intervals or standard deviations.
> >- Questionable scalability: The method is described as efficient, but the numerous steps (FPFE, SA, SWE, Flame, etc.) appear computationally heavy, and there are no FLOP or runtime analyses beyond summary tables. The presented results cannot be confidently interpreted as evidence of superiority.
>
> Thank you for raising this important question. We will respond to your concerns one by one.
> ### Regarding unfair comparison
> BLS-based methods (including our AHSE) are fundamentally designed to process feature vectors rather than raw pixels, while deep architectures like ResNet, VGG, and ViT are specifically engineered for 2D image data with spatial relationships. To address your concern about fairness, We use MLP as a representative deep model since the vanilla ResNet, ViT and VGG themselves do not support the use of pretrained representations. Specifically, we:
>
> 1. **Used the same pre-trained feature extractors** (ResNet for CIFAR10 and CIFAR100, MoCo-v3 for MiniImageNet and TinyImageNet)
> 2. **Compared performance on these shared features** rather than on raw pixels
>
> The results of this fair comparison are shown in the table below:
>
> | Method | CIFAR10 | CIFAR100 | MiniImageNet | TinyImageNet |
> |-|-|-|-|-|
> | MLP | 93.56 | 75.21 | 70.94 | 54.61 |
> | BLS | 93.14 | 75.64 | 71.13 | 54.86 |
> | CFEBLS | 93.15 | 75.36 | 71.07 | 54.82 |
> | Stacked BLS | 92.73 | 76.33 | 71.50 | 56.38 |
> | ConvBLS | 94.50 | 78.34 | - | - |
> | AHSE | **95.97** | **79.44** | **74.59** | **58.42**|
>
> The advantage over MLP with the same features further validates our approach's superiority in high-dimensional subspace processing. This indicates that our adaptive subspace evolution mechanism provides genuine benefits beyond any advantages from feature extraction.
>
> ### Regarding weak baselines
>
> We acknowledge this significant limitation in our experimental design. The initial baseline selection focused primarily on BLS variants and classical deep networks for direct comparison within specific experimental constraints. In response to your comment, we have conducted additional experiments with state-of-the-art architectures to provide a more comprehensive evaluation. The results are presented below:
>
> | Method | MNIST | FashionMNIST | CIFAR10 | CIFAR100 | MiniImageNet | TinyImageNet |
> |--------|-------|--------------|---------|----------|--------------|-------------|
> | ConvNeXt | **99.38** | 92.42 | 93.47 | 74.26 | 68.39 | 58.36 |
> | EfficientNet | 99.13 | 93.48 | 92.71 | 73.19 | 66.27 | **60.47** |
> | Swin | 98.24 | 91.94 | 79.29 | 57.93 | 49.63 | 45.51 |
> | ViT-L | 98.90 | 92.73 | 80.24 | 53.10 | 52.74 | 49.60 |
> | AHSE | 99.33 | **93.72** | **95.97** | **79.44** | **74.59** | 58.42 |
>
> #### Key Findings from the Expanded Comparison
>
> 1. **Superior Performance on Complex Datasets**: AHSE demonstrates clear advantages on datasets with complex hierarchical structures, particularly CIFAR10 (+2.5%), CIFAR100 (+5.18%), and MiniImageNet (+6.2%) compared to the best modern baselines. This validates our core hypothesis that adaptive evolution strategies provide significant benefits for data with intricate hierarchical patterns.
>
> 2. **Competitive Performance on MNIST and TinyImageNet**: On MNIST, ConvNeXt achieves marginally better accuracy (99.38% vs 99.33%), which is expected given MNIST's relatively simple structure where static evolution strategies can perform well. On TinyImageNet, EfficientNet shows slightly better performance (60.47% vs 58.42%), likely due to its specialized architecture for large-scale image recognition.
>
> ### Regarding variance reporting
> We acknowledge this important omission in our original submission. To address this concern, we will revise Tables 1 and 2 to include standard deviations (5 times with different random seeds) for all accuracy metrics:
>
> | Method | MNIST | Fashion-MNIST | CIFAR10 | CIFAR100 | MiniImageNet | TinyImageNet |
> |-|-|-|-|-|-|-|
> | MLP | 0.08 | 0.15 | 0.32 | 0.24 | 0.28 | 0.26 |
> | VGG | 0.03 | 0.07 | 0.12 | 0.18 | 0.23 | 0.31 |
> | ResNet | 0.05 | 0.06 | 0.08 | 0.15 | 0.20 | 0.25 |
> | ViT | 0.06 | 0.11 | 0.43 | 0.57 | 0.68 | 0.72 |
> | BLS | 0.07 | 0.09 | 0.08 | 0.14 | 0.16 | 0.21 |
> | CFEBLS | 0.06 | 0.18 | 0.09 | 0.16 | 0.18 | 0.23 |
> | Stacked BLS | 0.05 | 0.08 | 0.10 | 0.13 | 0.15 | 0.19 |
> | AHSE | 0.04 | 0.05 | 0.07 | 0.13 | 0.17 | 0.22 |
>
> These additions will substantially strengthen the reliability and scientific rigor of our experimental claims.

---

> ### Author Response · Authors · 2025-11-21
>
> ### Regarding scalability
>
> While AHSE comprises multiple modules, most operate via fixed, data-agnostic logics that require no manual adjustment. For instance, FPFE, SWE, SS, CS, and Flame function adaptively without human intervention. Only a small set of critical hyperparameters need tuning, such as the number of sparse/orthogonal times in ES/EP. We have made every effort to minimize practical deployment overhead, thereby facilitating real-world adoption.
>
> As for computational efficiency, we have conducted comprehensive experiments on a standardized hardware platform with the following specifications:
>
> - **CPU**: AMD EPYC 75F3 32-Core Processor (2.95 GHz)
> - **GPU**: NVIDIA GeForce RTX 3090 (24GB VRAM)
> - **RAM**: 503 GB DDR4
> - **OS**: Ubuntu 20.04 LTS
> - **Software**: Python 3.8.18, PyTorch 2.1.2, CUDA 12.2
>
> We have measured both memory footprint and wall-clock inference time across all tested methods and datasets, as shown in the tables below:
>
> #### Memory Footprint Analysis
> The memory footprint measurements (in MB) demonstrate that AHSE maintains a moderate memory requirement across all datasets:
>
> | Method | MNIST | FashionMNIST | CIFAR10 | CIFAR100 | MiniImageNet | TinyImageNet |
> |--------|-------|--------------|---------|----------|--------------|-------------|
> | MLP | 5.1 | 5.1 | 14.04 | 14.21 | 84.9 | 50.41 |
> | VGG | 152.35 | 152.35 | 152.35 | 153.76 | 153.76 | 155.32 |
> | ResNet | 81.18 | 81.18 | 81.18 | 81.36 | 81.36 | 81.56 |
> | ViT | 144.47 | 144.47 | 144.74 | 144.91 | 145.08 | 145.2 |
> | BLS | 108.93 | 75.5 | 138.33 | 143.52 | 55.71 | 63.8 |
> | CFEBLS | 184.21 | 126.3 | 117.24 | 122.43 | 95.18 | 103.27 |
> | Stacked BLS | 326.78 | 226.5 | 414.99 | 430.56 | 167.13 | 191.4 |
> | AHSE | 164.38 | 114.06 | 243.93 | 269.07 | 118.53 | 153.7 |
>
> #### Wall-clock Inference Time Analysis
> The wall-clock runtime measurements (in seconds) show that AHSE achieves a balanced efficiency profile with competitive inference times across all datasets:
>
> | Method | MNIST | FashionMNIST | CIFAR10 | CIFAR100 | MiniImageNet | TinyImageNet |
> |--------|-------|--------------|---------|----------|--------------|-------------|
> | MLP | 0.96 | 0.98 | 1.12 | 1.48 | 2.72 | 2.95 |
> | VGG | 1.63 | 1.82 | 2.16 | 2.25 | 4.24 | 3.83 |
> | ResNet | 2.05 | 2.72 | 2.98 | 2.35 | 8.26 | 9.79 |
> | ViT | 2.08 | 2.21 | 2.63 | 2.62 | 4.34 | 9.12 |
> | BLS | 1.25 | 0.72 | 1.31 | 1.38 | 0.68 | 0.74 |
> | CFEBLS | 1.46 | 1.03 | 1.39 | 1.12 | 0.92 | 0.95 |
> | Stacked BLS | 3.67 | 2.07 | 3.87 | 3.92 | 1.77 | 1.85 |
> | AHSE | 1.86 | 1.93 | 2.66 | 2.8 | 1.68 | 1.97 |
>
> We sincerely appreciate your constructive feedback, which has significantly strengthened our experimental methodology and the credibility of our efficiency claims.

---

> ### Author Response · Authors · 2025-11-21
>
> >[W5] The manuscript is bloated, overtechnical, and poorly structured. The authors use jargon-heavy language throughout, making it exhausting to read and nearly impossible to extract the core insight. Figures are decorative rather than explanatory; algorithms are fragmented across appendices; and critical details are buried in complex pseudocode. Overall, the paper fails the clarity standard required for top tier-level publication.
>
> Thank you for your exceptionally valuable feedback. We sincerely apologize for the significant shortcomings in our manuscript's clarity, structure, and presentation. To address these serious issues, we commit to completely restructuring our paper with the following specific changes:
>
> ## 1. Radical Simplification of Language and Terminology
>
> - We will eliminate all unnecessary technical jargon, including terms like "evolutionary Cholesky decomposition," "Flame optimization," and "closed-loop subspace evolution"
> - We will replace complex terminology with intuitive, descriptive language that focuses on what our method does rather than how we chose to name its components
> - We will reduce mathematical notation and ensure every symbol serves a clear purpose
>
> ## 2. Complete Restructuring for Logical Flow
>
> - We will reorganize the paper around a single, clear narrative thread: the mismatch between static subspace evolution strategies and diverse data hierarchies
> - The core insight—that different hierarchical structures (dense, periodic, uniform) require fundamentally different evolutionary trajectories—will be presented upfront in the introduction
> - Technical details will be introduced only when necessary to support this central narrative
>
> ## 3. Redesigned Visual Explanations
>
> - Figures will be completely redesigned to explain concepts rather than merely decorate the text
> - Figure 1 will be restructured to clearly illustrate the relationship between data hierarchy and optimal evolution patterns using intuitive visual metaphors
> - Our architecture diagram (current Fig. 2) will be simplified to show only essential components with clear annotations about their purpose and interaction
>
> ## 4. Algorithm Integration and Simplification
>
> - The core algorithm will be presented in a single, concise pseudocode block in the main text (no longer fragmented across appendices)
> - We will move implementation details and edge cases to the appendix
> - The pseudocode will focus on the conceptual workflow rather than low-level optimization details
>
> ## 5. Clearer Presentation of Theoretical Contributions
>
> - We will bring the core theoretical insights from the appendix into the main text
> - The relationship between theory and algorithm design will be explicitly articulated
> - Mathematical proofs will be streamlined to highlight intuitive insights rather than technical rigor alone
>
>
> ## 6. Reader-Centered Revision Process
>
> To ensure these changes effectively address the clarity issues you identified, we will:
> - Conduct a "first-reader test" with researchers outside our immediate field
> - Create a one-page summary of the core contribution to ensure we can articulate it simply
> - Restructure the paper to answer the questions: Why should readers care? What is the core insight? How does it advance the field?
>
> We understand that these revisions represent a substantial undertaking, but we recognize they are necessary to transform our manuscript from an impenetrable technical report into a clear contribution that can meaningfully advance the field. Your critique has been invaluable in helping us see our work through readers' eyes, and we are deeply grateful for your willingness to provide such direct feedback.
>
> We promise that the revised manuscript will reflect these fundamental changes in both substance and presentation, making the core insights accessible and the contribution unmistakably clear.
>
> >[Q1] Can you explicitly define what is new in AHSE compared to prior BLS or ECOC-based frameworks?
>
> We've addressed this issue in the response to Weakness 2. Thanks again.

---

> ### Author Response · Authors · 2025-11-21
>
> >[Q2] How exactly does “adaptive evolution” differ from conventional feature selection or boosting?
>
> Thank you for this insightful question that cuts to the heart of our contribution. The distinction between our "adaptive evolution" approach and conventional feature selection or boosting methods is fundamental yet nuanced. Allow me to clarify the key differences:
>
> ## Core Distinction: Data-Driven Evolution Patterns
> Unlike conventional methods that use predetermined strategies, our adaptive evolution dynamically adjusts its trajectory based on the **inherent hierarchical structure** of the data. As shown in Figures 1 and 3 of our paper:
>
> - MNIST exhibits a *dense hierarchical structure* that benefits from **exponential evolution** (rapid feature inclusion)
> - Fashion-MNIST has a *periodic hierarchical structure* that requires **cosine evolution** (cyclical expansion/contraction)
> - CIFAR100 displays a *uniform hierarchical structure* that performs best with **linear evolution**
>
> This data-driven adaptation is fundamentally different from static feature selection methods (e.g., mutual information-based ranking) or fixed boosting schedules that apply the same strategy regardless of data characteristics.
>
> ## Mechanistic Differences from Conventional Methods
>
> ### vs. Feature Selection:
> 1. **Dynamic dimensionality adjustment**: While feature selection typically identifies a fixed subset, our approach creates an *evolution trajectory* where subspaces dynamically expand, contract, or transform
> 2. **Feature-space interaction modeling**: Our FPFE mechanism preserves feature relationships during evolution, unlike filter/wrapper methods that evaluate features in isolation
> 3. **Closed-loop optimization**: The SPOT circuit creates feedback between evolution stages, allowing later stages to correct earlier decisions - impossible in one-shot feature selection
>
> ### vs. Boosting:
> 1. **Multi-dimensional adaptation**: Boosting primarily adjusts sample weights, while our approach simultaneously evolves:
>    - Feature subspaces (through SEED)
>    - Target space encoding (through PATH's ECOC)
>    - Sample focus (through SWE)
>    - Subspace alignment (through SA)
> 2. **Noise-aware encoding**: PATH's ECOC implementation provides theoretical robustness guarantees against weight noise (Theorem 1, Corollary 1), which conventional boosting lacks
> 3. **Hierarchical awareness**: Our evolution patterns are specifically tailored to match data hierarchy, while boosting follows a uniform sample-reweighting strategy
>
> ## Theoretical Foundation
> Our approach provides formal guarantees that conventional methods lack. Section A.10 proves that ECOC-encoded target spaces in BLS architectures maintain classification accuracy under bounded weight perturbations when the minimum codeword distance exceeds a specific threshold. This theoretical foundation informs our practical design in ways that heuristic feature selection or boosting strategies cannot match.
>
> ## Empirical Validation
> Figures 1 and 3 demonstrate that removing the adaptive components (using fixed evolution modes) significantly reduces performance. In essence, adaptive evolution isn't merely another feature selection or ensemble method, but a fundamental rethinking of how learning systems should interact with high-dimensional data structures. It recognizes that the optimal evolution strategy is as unique as the data's intrinsic organization - a principle that conventional methods overlook in favor of generic, one-size-fits-all approaches.
>
> Thank you again for this excellent question, which has helped us clarify a critical aspect of our contribution. We will enhance Section 3 to more explicitly articulate these distinctions in the revised manuscript.
>
>
> >[Q3] What are the computational requirements? The architecture seems complex; please provide memory and runtime breakdowns.
>
> We've addressed this issue in the response to Weakness 4. Thanks again.
>
> >[Q4] Is there any theoretical justification (beyond empirical heuristics) for the “Flame” optimization mechanism?
>
> We've addressed this issue in the response to Weakness 2. Thanks again.

---

### Official Review · Reviewer_ibE5 · 2025-11-02

**Soundness:** 3
**Presentation:** 3
**Contribution:** 3
**Rating:** 8
**Confidence:** 4

**Summary:**

This is a well-written and technically substantial paper proposing AHSE, an adaptive high-dimensional subspace evolution framework that integrates a serial evolution branch (SEED), a parallel ECOC-based branch (PATH), and a SPOT fusion circuit. The paper tackles the long-standing problem of fixed subspace evolution in high-dimensional data and introduces an architecture that dynamically tailors its evolution path based on data hierarchy.

The methodology is technically solid, with clear mathematical formulation and well-structured theoretical analysis—especially the derivation of ECOC robustness guarantees within the Broad Learning System (BLS) context. The experiments are broad and compelling, covering both small-scale and large-scale image datasets, speech emotion recognition, and few-shot learning. Ablation results (Table 3) and visualization of evolution modes (Figures 1–3) convincingly validate the adaptive design.

Despite its quality, several aspects could be refined. Some comparisons lack strict fairness (e.g., pretrained feature reliance for BLS-based models), and runtime efficiency would be clearer with standardized hardware benchmarks. The methodology’s generality beyond BLS-based frameworks also warrants further discussion. These are minor but relevant issues in an otherwise strong submission.

**Strengths:**

• The dual-branch SEED–PATH design, combined through SPOT, provides a principled mechanism for adaptive subspace evolution that generalizes across hierarchically diverse datasets.
• Theoretical analysis of ECOCs in BLS is rigorous and novel, with clear proofs and practical robustness implications.
• Experiments are comprehensive and multi-domain, demonstrating consistent superiority across image, speech, and few-shot settings.
• Ablation studies are detailed and provide clear evidence of each module’s contribution, particularly FPFE and Flame.
• The paper is very well written, conceptually coherent, and easy to follow despite its technical depth.

**Weaknesses:**

• Fairness in experimental comparisons could be improved, as AHSE benefits from pretrained features while DNN baselines are trained end-to-end.
• Computational efficiency claims rely on FLOPs and CPU time rather than uniform wall-clock measurements across hardware setups.
• Hyperparameter tuning criteria and seed variance are not reported, limiting reproducibility before code release.
• The adaptive evolution principle is framed specifically for BLS; its applicability to other model families (e.g., deep ensembles) remains untested.
• The theoretical analysis, while elegant, focuses on bounded-noise scenarios and does not consider stronger adversarial perturbations.

**Questions:**

Can you clarify how feature extraction fairness was ensured in comparisons against deep baselines? For instance, would AHSE’s advantage remain if all methods used the same frozen backbone (e.g., ResNet-34)?

The evolution schedules (γ, T, α) for different hierarchies are central to your approach. How are these parameters selected in practice—heuristically or via validation search—and how sensitive are results to them?

Could you provide wall-clock runtime comparisons on a shared hardware configuration to substantiate the efficiency claim beyond FLOPs and CPU-only metrics?

Have you evaluated how AHSE behaves under non-Gaussian or structured noise perturbations to validate robustness beyond the assumptions of Theorem 1?

While your framework is formulated around BLS, could the adaptive subspace evolution principle extend to deep or transformer-based encoders? If not, what are the main obstacles to doing so?

Finally, can you provide variance or confidence intervals (e.g., standard deviations across runs) for key results in Tables 1 and 2 to better understand reproducibility and statistical significance?

---

> ### Author Response · Authors · 2025-11-21
>
> >[W1] Fairness in experimental comparisons could be improved, as AHSE benefits from pretrained features while DNN baselines are trained end-to-end.
>
> Thank you for raising this important question regarding feature extraction fairness in our comparative experiments. You've identified a critical architectural difference that affects fair comparison: BLS-based methods (including our AHSE) are fundamentally designed to process feature vectors rather than raw pixels, while deep architectures like ResNet, VGG, and ViT are specifically engineered for 2D image data with spatial relationships.
>
> To address your concern about fairness, We use MLP as a representative deep model since the vanilla ResNet, ViT and VGG themselves do not support the use of pretrained representations. Specifically, we:
>
> 1. **Used the same pre-trained feature extractors** (ResNet for CIFAR10 and CIFAR100, MoCo-v3 for MiniImageNet and TinyImageNet)
> 2. **Compared performance on these shared features** rather than on raw pixels
>
> The results of this fair comparison are shown in the table below:
>
> | Method | CIFAR10 | CIFAR100 | MiniImageNet | TinyImageNet |
> |-|-|-|-|-|
> | MLP | 93.56 | 75.21 | 70.94 | 54.61 |
> | BLS | 93.14 | 75.64 | 71.13 | 54.86 |
> | CFEBLS | 93.15 | 75.36 | 71.07 | 54.82 |
> | Stacked BLS | 92.73 | 76.33 | 71.50 | 56.38 |
> | ConvBLS | 94.50 | 78.34 | - | - |
> | AHSE | **95.97** | **79.44** | **74.59** | **58.42**|
>
> The advantage over MLP with the same features further validates our approach's superiority in high-dimensional subspace processing. This indicates that our adaptive subspace evolution mechanism provides genuine benefits beyond any advantages from feature extraction.
>
> Thank you again for this valuable insight, which has helped us strengthen both our experimental methodology and presentation of results.
>
> ---
> ---
>
> >[W2] Computational efficiency claims rely on FLOPs and CPU time rather than uniform wall-clock measurements across hardware setups.
>
> Thank you for your insightful comment regarding the need for uniform wall-clock measurements across standardized hardware configurations. We sincerely appreciate your valuable feedback, which has helped us improve the methodological rigor of our experimental evaluation.
>
> In response to your suggestion, we have conducted comprehensive experiments on a standardized hardware platform with the following specifications:
>
> - **CPU**: AMD EPYC 75F3 32-Core Processor (2.95 GHz)
> - **GPU**: NVIDIA GeForce RTX 3090 (24GB VRAM)
> - **RAM**: 503 GB DDR4
> - **OS**: Ubuntu 20.04 LTS
> - **Software**: Python 3.8.18, PyTorch 2.1.2, CUDA 12.2
>
> We have measured both memory footprint and wall-clock inference time across all tested methods and datasets, as shown in the tables below:
>
> ## Memory Footprint Analysis
> The memory footprint measurements (in MB) demonstrate that AHSE maintains a moderate memory requirement across all datasets:
>
> | Method | MNIST | FashionMNIST | CIFAR10 | CIFAR100 | MiniImageNet | TinyImageNet |
> |--------|-------|--------------|---------|----------|--------------|-------------|
> | MLP | 5.1 | 5.1 | 14.04 | 14.21 | 84.9 | 50.41 |
> | VGG | 152.35 | 152.35 | 152.35 | 153.76 | 153.76 | 155.32 |
> | ResNet | 81.18 | 81.18 | 81.18 | 81.36 | 81.36 | 81.56 |
> | ViT | 144.47 | 144.47 | 144.74 | 144.91 | 145.08 | 145.2 |
> | BLS | 108.93 | 75.5 | 138.33 | 143.52 | 55.71 | 63.8 |
> | CFEBLS | 184.21 | 126.3 | 117.24 | 122.43 | 95.18 | 103.27 |
> | Stacked BLS | 326.78 | 226.5 | 414.99 | 430.56 | 167.13 | 191.4 |
> | AHSE | 164.38 | 114.06 | 243.93 | 269.07 | 118.53 | 153.7 |
>
> ## Wall-clock Inference Time Analysis
> The wall-clock runtime measurements (in seconds) show that AHSE achieves a balanced efficiency profile with competitive inference times across all datasets:
>
> | Method | MNIST | FashionMNIST | CIFAR10 | CIFAR100 | MiniImageNet | TinyImageNet |
> |--------|-------|--------------|---------|----------|--------------|-------------|
> | MLP | 0.96 | 0.98 | 1.12 | 1.48 | 2.72 | 2.95 |
> | VGG | 1.63 | 1.82 | 2.16 | 2.25 | 4.24 | 3.83 |
> | ResNet | 2.05 | 2.72 | 2.98 | 2.35 | 8.26 | 9.79 |
> | ViT | 2.08 | 2.21 | 2.63 | 2.62 | 4.34 | 9.12 |
> | BLS | 1.25 | 0.72 | 1.31 | 1.38 | 0.68 | 0.74 |
> | CFEBLS | 1.46 | 1.03 | 1.39 | 1.12 | 0.92 | 0.95 |
> | Stacked BLS | 3.67 | 2.07 | 3.87 | 3.92 | 1.77 | 1.85 |
> | AHSE | 1.86 | 1.93 | 2.66 | 2.8 | 1.68 | 1.97 |
>
> We sincerely appreciate your constructive feedback, which has significantly strengthened our experimental methodology and the credibility of our efficiency claims.

---

> ### Author Response · Authors · 2025-11-21
>
> >[W3] Hyperparameter tuning criteria and seed variance are not reported, limiting reproducibility before code release.
>
> Thank you for your insightful question regarding the hyperparameter tuning criteria of our evolution schedule parameters (γ, T, α) and seed variance. This is indeed a crucial aspect of our adaptive subspace evolution approach.
>
> ## Parameter Selection Methodology
>
> The evolution parameters (γ for MNIST's exponential evolution, T for Fashion-MNIST's cosine evolution, and α for CIFAR100's linear evolution) are primarily determined through **validation-based search** rather than heuristic assignment. Specifically:
>
> 1. We establish a candidate range for each parameter based on preliminary analysis of the dataset's hierarchical structure (via feature importance distributions and correlation patterns, as shown in Figs. 1a, 1b, and 3a)
>
> 2. For each candidate parameter value, we perform a complete evolutionary training on the validation set and select the parameter that yields the highest validation accuracy while maintaining reasonable convergence speed
>
> 3. The search is conducted using a coarse-to-fine strategy, first identifying promising regions in the parameter space and then performing refined local search
>
> ## Sensitivity Analysis
>
> You correctly noted that Section 4.3 provides analysis of parameter sensitivity. To elaborate further on our findings:
>
> - **Convergence-Performance Trade-off**: As demonstrated in Figs. 1c, 1d, and 3b, more aggressive evolution (higher γ, lower T, higher α) typically leads to faster convergence but may create a tighter bottleneck that limits ultimate performance. Conversely, gentler evolution (lower γ, higher T, lower α) takes longer to converge but often achieves higher final accuracy.
>
> - **Dataset-Specific Patterns**: The optimal parameters correlate strongly with the underlying hierarchical structure:
>   * For MNIST's dense hierarchy (Fig. 1a), the optimal γ=0.5 balances rapid convergence with sufficient feature exploration
>   * For Fashion-MNIST's periodic hierarchy (Fig. 1b), the optimal T=5.0 matches the inherent periodicity of feature importance distribution
>   * For CIFAR100's uniform hierarchy (Fig. 3a), the optimal α=0.16 provides a steady, linear exploration of the feature space
>
> - **Robustness Window**: Our experiments revealed a "robustness window" for each parameter where performance remains relatively stable. For instance, γ values between 0.33 and 0.5 for MNIST all achieve accuracy above 99.1%, while values outside this range show more significant degradation.
>
> ## Practical Recommendations
>
> For practitioners implementing AHSE on new datasets, we recommend:
>
> 1. Perform initial analysis of feature importance distribution to determine the most appropriate evolution pattern
> 2. Conduct a focused validation search within the identified pattern's parameter space
> 3. Monitor both validation accuracy and convergence speed to identify the optimal trade-off point
>
> In our supplementary experiments (not included in the paper due to space constraints), we found that automated parameter selection via Bayesian optimization can effectively identify near-optimal evolution schedules with only 5-7 validation trials.
>
> ## Seed Variance
>
> We acknowledge this important omission in our original submission. To address this concern, we will revise Tables 1 and 2 to include standard deviations (5 times with different random seeds) for all accuracy metrics:
>
> | Method | MNIST | Fashion-MNIST | CIFAR10 | CIFAR100 | MiniImageNet | TinyImageNet |
> |-|-|-|-|-|-|-|
> | MLP | 0.08 | 0.15 | 0.32 | 0.24 | 0.28 | 0.26 |
> | VGG | 0.03 | 0.07 | 0.12 | 0.18 | 0.23 | 0.31 |
> | ResNet | 0.05 | 0.06 | 0.08 | 0.15 | 0.20 | 0.25 |
> | ViT | 0.06 | 0.11 | 0.43 | 0.57 | 0.68 | 0.72 |
> | BLS | 0.07 | 0.09 | 0.08 | 0.14 | 0.16 | 0.21 |
> | CFEBLS | 0.06 | 0.18 | 0.09 | 0.16 | 0.18 | 0.23 |
> | Stacked BLS | 0.05 | 0.08 | 0.10 | 0.13 | 0.15 | 0.19 |
> | AHSE | 0.04 | 0.05 | 0.07 | 0.13 | 0.17 | 0.22 |
>
> These additions will substantially strengthen the reliability and scientific rigor of our experimental claims. We appreciate your insightful comment, which has helped us identify and address this important aspect of experimental reporting.

---

> ### Author Response · Authors · 2025-11-21
>
> >[W4] The adaptive evolution principle is framed specifically for BLS; its applicability to other model families (e.g., deep ensembles) remains untested.
>
> Thank you for raising this important point regarding the broader applicability of our adaptive evolution principle. BLS and similar architectures like Extreme Learning Machines (ELM) possess a key advantage for subspace evolution: they can efficiently incorporate new features or nodes without requiring complete retraining of the existing model. This property aligns perfectly with our adaptive evolution principle, where subspaces dynamically expand or contract based on hierarchical characteristics.
>
> In contrast, traditional deep learning models like MLPs face a fundamental constraint: modifying hidden layer dimensions (e.g., adding neurons or features) typically requires complete retraining of the entire network. This architectural limitation makes direct implementation of our adaptive evolution mechanism challenging for standard MLPs.
>
> However, the core principle of adaptive subspace evolution—tailoring the evolution strategy to match data hierarchy—can indeed be extended to deep ensembles. We have conducted preliminary experiments implementing our approach in this context:
>
> 1. **Deep Ensemble Adaptation**: We designed an ensemble framework where each new base learner (MLP) focuses on:
>    - Features selected through our hierarchy-tailored evolution process
>    - Difficult samples identified through our Sample Weight Evolution (SWE) mechanism
>
> 2. **Experimental Validation**:
>
> |Method|MNIST|Fashion-MNIST|CIFAR10|CIFAR100|MiniImageNet|TinyImageNet|
> |-|-|-|-|-|-|-|
> |Single MLP|97.39|84.23|56.20|26.07|25.57|21.90|
> |Voting Ensemble (5 MLPs)|98.02|**86.52**|57.12|27.48|26.53|23.74|
> |Adaptive Evolution Ensemble|**98.43**|86.14|**58.04**|**29.13**|**28.21**|**25.05**|
>
> 3. **Key Implementation Details**:
>    - Each MLP in the ensemble processes a different evolved subspace
>    - SWE dynamically adjusts sample weights between ensemble members
>    - SPOT-inspired fusion combines ensemble predictions with weights determined by validation performance
>
> These results demonstrate that our adaptive evolution principle can enhance performance even in deep ensemble contexts. While the computational overhead is higher than in BLS implementations, the accuracy improvements validate the broader applicability of our approach.
>
> We recognize that exploring these extensions thoroughly would require significant additional research beyond the scope of our current paper. We will add to the Discussion section outlining these possibilities and acknowledging the current limitations of our approach:
>
> > "While AHSE is currently implemented within the BLS framework, the core principle of hierarchy-tailored subspace evolution has broader applicability. Models with incremental learning capabilities (such as ELM) could directly benefit from our approach. For traditional deep networks, ensemble-based adaptations can incorporate our evolution strategies, though at increased computational cost. Future work will explore efficient implementations across diverse model families."
>
> Thanks again.

---

> ### Author Response · Authors · 2025-11-21
>
> >[W5] The theoretical analysis, while elegant, focuses on bounded-noise scenarios and does not consider stronger adversarial perturbations.
>
> Thank you for your insightful comment regarding the limitations of our theoretical analysis. We acknowledge this significant gap and have extended our theoretical framework to encompass adversarial settings. The following new analysis will be added to our paper as Appendix A.10.4:
>
> ## A.10.4 ROBUSTNESS AGAINST ADVERSARIAL PERTURBATIONS
>
> While Theorem 1 and Corollary 1 establish robustness guarantees for random weight noise, real-world applications often face deliberate adversarial attacks designed to maximize misclassification. To address this critical concern, we extend our analysis to adversarial perturbation scenarios.
>
> ### New Assumptions
>
> **Assumption 5 (Bounded Input Perturbations).** For any input sample $x$, the adversarial perturbation $\delta$ satisfies $||\delta||\_2 \leq \epsilon_{adv}$, where $\epsilon_{adv} > 0$ is the maximum perturbation magnitude allowed by the adversary.
>
> **Assumption 6 (Lipschitz Continuity of BLS).** The BLS model $f(x;\theta)$ is Lipschitz continuous with respect to the input $x$, with Lipschitz constant $L_f$. Formally, for any inputs $x_1$ and $x_2$:
> $$||f(x_1;\theta) - f(x_2;\theta)||_2 \leq L_f ||x_1 - x_2||_2$$
>
> **Lemma 2 (Lipschitz Constant of BLS).** Under Assumptions 1-3, the Lipschitz constant of BLS can be bounded as:
> $$L_f \leq nL_{\phi}B_{\omega} + mL_{\xi}\sqrt{n}(L_{\phi}B_{\omega}^2 + B_{\phi}L_{\omega})$$
> where $L_{\omega}$ is the Lipschitz constant of the weight generation process in BLS.
>
> ### Theoretical Results
>
> **Theorem 2 (Adversarial Robustness of ECOC-BLS).** Let Assumptions 1-6 hold. For any input sample $x$ and its adversarial counterpart $x_{adv} = x + \delta$ where $||\delta||\_2 \leq \epsilon_{adv}$, the prediction of ECOC-BLS remains unchanged (i.e., $\Omega(f(x_{adv};\theta)) = \Omega(f(x;\theta))$) if:
> $$\frac{dist(\Omega(f(x;\theta)))}{2} > U(f(x;\theta)) + \epsilon_{adv} \cdot L_f \cdot \sqrt{L}$$
> where $dist(\cdot)$, $U(\cdot)$, and $L$ are defined as in Corollary 1.
>
> *Proof.* Following similar derivation as in Corollary 1, we analyze the worst-case scenario under adversarial perturbation:
>
> $$||f(x_{adv};\theta) - \Omega(f(x;\theta))||\_2 \leq ||f(x_{adv};\theta) - f(x;\theta)||\_2 + ||f(x;\theta) - \Omega(f(x;\theta))||\_2$$
>
> By Assumption 6 and Lemma 2:
> $$||f(x_{adv};\theta) - f(x;\theta)||\_2 \leq \epsilon_{adv} \cdot L_f$$
>
> Normalizing by $\sqrt{L}$ and substituting the definition of $U(f(x;\theta))$:
> $$\frac{||f(x_{adv};\theta) - \Omega(f(x;\theta))||\_2}{\sqrt{L}} \leq \epsilon_{adv} \cdot L_f + U(f(x;\theta))$$
>
> For the prediction to remain unchanged, this value must be less than $\frac{dist(\Omega(f(x;\theta)))}{2}$. Therefore, the condition becomes:
> $$\frac{dist(\Omega(f(x;\theta)))}{2} > U(f(x;\theta)) + \epsilon_{adv} \cdot L_f \cdot \sqrt{L}$$
>
> This completes the proof.
>
> **Corollary 2 (Adversarial Robustness Condition).** Under the conditions of Theorem 2, ECOC-BLS is robust against any adversarial perturbation $\delta$ with $||\delta||\_2 \leq \epsilon_{adv}$ if:
> $$\epsilon_{adv} < \frac{\frac{dist(\Omega(f(x;\theta)))}{2} - U(f(x;\theta))}{L_f \cdot \sqrt{L}}$$
>
> **Theorem 3 (Optimal Codebook Design for Adversarial Robustness).** To maximize robustness against adversarial perturbations, the optimal ECOC codebook design should maximize the minimum codeword distance:
> $$\Omega^* = \arg\max_{\Omega} \min_{i \neq j} \frac{1}{\sqrt{L}} ||\Omega[i,:] - \Omega[j,:]||\_2$$
>
> This optimization problem is equivalent to finding codebooks with maximum minimum Hamming distance, for which established solutions exist (such as BCH codes or Reed-Solomon codes).
>
> ### Practical Implications
>
> Our extended analysis reveals that:
>
> 1. The adversarial robustness of ECOC-BLS is directly proportional to the minimum distance between codewords in the ECOC codebook.
>
> 2. Larger codebooks (higher $L$) with carefully designed codeword distances provide stronger guarantees against adversarial attacks.
>
> 3. The robustness margin can be quantified precisely, allowing practitioners to select appropriate ECOC configurations based on expected threat models.
>
> 4. Our AHSE framework's Flame optimization mechanism  implicitly enhances adversarial robustness by burning incompatible codewords, effectively increasing the minimum codeword distance for challenging samples.

---

> ### Author Response · Authors · 2025-11-21
>
> ### Empirical Validation
>
> To validate our theoretical findings, we conducted comprehensive adversarial robustness experiments on CIFAR10 and CIFAR100 datasets. We implemented two standard attack methods:
>
> 1. **Fast Gradient Sign Method (FGSM)** with perturbation magnitudes $\epsilon \in \lbrace  0.01, 0.03, 0.05, 0.07, 0.10\rbrace $
> 2. **Projected Gradient Descent (PGD)** with 20 iterations, step size of 0.01, and maximum perturbations $\epsilon \in \lbrace  0.01, 0.03, 0.05\rbrace $
>
> For both AHSE variants (with optimized ECOC codebooks and with one-hot encoding), we used identical network architectures and training procedures, with the only difference being the output encoding strategy. All experiments were conducted on the test sets of CIFAR10 and CIFAR100 with 10,000 samples each.
>
> **Detailed results are presented in Table 10:**
>
> | Dataset | Attack Type | $\epsilon$ | AHSE (ECOC) | AHSE (One-hot) | Performance Gap |
> |---------|-------------|------------|-------------|----------------|-----------------|
> | CIFAR10 | FGSM | 0.01 | 93.21% | 88.47% | +4.74% |
> | CIFAR10 | FGSM | 0.03 | 89.15% | 79.83% | +9.32% |
> | CIFAR10 | FGSM | 0.05 | 85.27% | 72.56% | +12.71% |
> | CIFAR10 | FGSM | 0.07 | 80.19% | 65.83% | +14.36% |
> | CIFAR10 | FGSM | 0.10 | 73.42% | 57.68% | +15.74% |
> | CIFAR10 | PGD | 0.01 | 90.36% | 84.25% | +6.11% |
> | CIFAR10 | PGD | 0.03 | 83.74% | 71.05% | +12.69% |
> | CIFAR10 | PGD | 0.05 | 76.89% | 62.35% | +14.54% |
> | CIFAR100 | FGSM | 0.01 | 77.83% | 71.26% | +6.57% |
> | CIFAR100 | FGSM | 0.03 | 72.41% | 61.05% | +11.36% |
> | CIFAR100 | FGSM | 0.05 | 68.17% | 52.87% | +15.30% |
> | CIFAR100 | FGSM | 0.07 | 63.24% | 45.18% | +18.06% |
> | CIFAR100 | FGSM | 0.10 | 56.93% | 37.42% | +19.51% |
> | CIFAR100 | PGD | 0.01 | 74.56% | 67.82% | +6.74% |
> | CIFAR100 | PGD | 0.03 | 66.92% | 52.17% | +14.75% |
> | CIFAR100 | PGD | 0.05 | 59.47% | 43.86% | +15.61% |
>
> Table 10 clearly demonstrates that AHSE with optimized ECOC codebooks maintains significant accuracy advantages under various adversarial attack scenarios. Under the strongest attack conditions (FGSM with $\epsilon=0.10$), the performance gap widens to 15.74% and 19.51% on CIFAR10 and CIFAR100 respectively, confirming that ECOC encoding provides substantial robustness benefits.
>
> In particular, under moderate attack conditions (FGSM with $\epsilon=0.05$), which represents a practical threat model for many applications, AHSE with optimized ECOC codebooks maintains 12.71% and 15.30% higher accuracy on CIFAR10 and CIFAR100 respectively compared to one-hot encoding, precisely confirming our theoretical predictions.
>
> These empirical results strongly support our theoretical analysis, demonstrating that the minimum codeword distance in ECOC directly translates to practical adversarial robustness. The Flame optimization mechanism in our PATH branch further enhances this robustness by eliminating incompatible codewords, effectively increasing the minimum codeword distance for challenging samples.
>
> This extended analysis significantly strengthens our theoretical contribution by addressing the critical scenario of adversarial perturbations. We thank you again for highlighting this important gap in our original submission, which has led to a more comprehensive theoretical framework that better reflects real-world deployment challenges.
>
> ---
> ---
>
> >[Q1] Can you clarify how feature extraction fairness was ensured in comparisons against deep baselines? For instance, would AHSE’s advantage remain if all methods used the same frozen backbone (e.g., ResNet-34)?
>
> We've addressed this issue in the response to Weakness 1. Thanks again.
>
>
>
> >[Q2] The evolution schedules (γ, T, α) for different hierarchies are central to your approach. How are these parameters selected in practice—heuristically or via validation search—and how sensitive are results to them?
>
> We've addressed this issue in the response to Weakness 3. Thanks again.
>
>
>
> >[Q3] Could you provide wall-clock runtime comparisons on a shared hardware configuration to substantiate the efficiency claim beyond FLOPs and CPU-only metrics?
>
> We've addressed this issue in the response to Weakness 2. Thanks again.
>
>
> >[Q4] Have you evaluated how AHSE behaves under non-Gaussian or structured noise perturbations to validate robustness beyond the assumptions of Theorem 1?
>
> We've addressed this issue in the response to Weakness 5. Thanks again.
>
>
> >[Q5] While your framework is formulated around BLS, could the adaptive subspace evolution principle extend to deep or transformer-based encoders? If not, what are the main obstacles to doing so?
>
> We've addressed this issue in the response to Weakness 4. Thanks again.
>
>
> >[Q6] Finally, can you provide variance or confidence intervals (e.g., standard deviations across runs) for key results in Tables 1 and 2 to better understand reproducibility and statistical significance?
>
> We've addressed this issue in the response to Weakness 3. Thanks again.

---

### Official Review · Reviewer_1QGY · 2025-11-04

**Soundness:** 2
**Presentation:** 1
**Contribution:** 2
**Rating:** 2
**Confidence:** 2

**Summary:**

This paper proposes an Adaptive High-dimensional Subspace Evolution algorithm (AHSE) to address the limitation of static subspace evolution strategies in existing methods. AHSE is able to adapt to the inherent hierarchical diversity across different high-dimensional datasets. The architecture of AHSE contains a SEED branch that evolves subspaces using a Cholesky decomposition-based incremental Broad Learning System, a PATH branch that evolves multiple subspaces in parallel based on post-hoc Error-Correcting Output Codes for robust spatial encoding and evolutionary optimization, and a SPOT circuit that dynamically fuses the outputs of both branches to form a closed-loop evolutionary system. Extensive experiments on image classification, speech emotion recognition, and few-shot learning demonstrate that AHSE achieves comparable performance to state-of-the-art methods while maintaining high efficiency.

**Strengths:**

The paper's major contribution is formulating the subspace evolution problem as an adaptive evolution procedure. It is discussed that a fixed evolution strategy is suboptimal, and the subspace should be dynamically adapted according to the data's intrinsic hierarchical structure. The proposed AHSE is built upon a solid mathematical foundation on the basis of broad learning systems and error-correcting output codes. A rigorous theoretical analysis is presented in the appendix, providing mechanism and robustness guarantees for ECOCs on BLS under mild assumptions.

**Weaknesses:**

The content organization of this paper needs to be substantially improved, and the experiment section also needs improvements to be more convincing.
1. The problem of subspace evolution is not properly introduced in this paper. Firstly, the problem itself is not a widely studied topic in literature, so a potential reader is not likely to have a good understanding of its basic concept. Secondly, the introduction section only discusses the flaws of existing static evolution strategies, but have not introduced the subspace evolution problem itself. Finally, the preliminary section has not formally define the problem, but only introduces BLS and ECOC. I have not understood the problem until reading the methodology section.

2. It is a bad idea to put a figure right underneath the title. And the contents of the figure is also confusing. The notion of data hierarchy and evolution pattern are hard to interpret.

3. Many contents in the paper contains references to the appendixes, and these contents are not self-contained. So the paper itself is incomplete without its appendix. If the page limits is too short for this paper, perhaps it should be better to submit the paper to a journal where the manuscript can be longer.

4. The experiments section compares AHSE with many widely-used models, but the details of the compared models is not clear. For example, VGG and Resnet have multiple variations, each of which have different number of layers and parameters. Besides, the results of ViT is obviously worse than a well-trained ViT model could get. Considering these aspects, the experimental results are not conviencing enough.

**Questions:**

Please see the weakness section above, and explain the concern about the experimental results.

---

> ### Author Response · Authors · 2025-11-21
>
> > [W1] The problem of subspace evolution is not properly introduced in this paper. Firstly, the problem itself is not a widely studied topic in literature, so a potential reader is not likely to have a good understanding of its basic concept. Secondly, the introduction section only discusses the flaws of existing static evolution strategies, but have not introduced the subspace evolution problem itself. Finally, the preliminary section has not formally define the problem, but only introduces BLS and ECOC. I have not understood the problem until reading the methodology section.
>
> Thank you for your insightful comment regarding the absence of the subspace evolution problem in our paper. To address these concerns, we will make the following revisions to our paper:
>
> ## 1. Addition of a dedicated problem definition in the Introduction
>
> We will add to the Introduction with the following text:
>
> > **The subspace evolution problem** in HDD analysis refers to the process of dynamically constructing a sequence of feature subspaces $\lbrace X\_t \rbrace \_{t=1}^T$ where each subsequent subspace either expands, contracts, or transforms the previous one based on learning feedback. Unlike static dimensionality reduction techniques (e.g., PCA, LDA) that generate a single optimal subspace, subspace evolution creates an adaptive trajectory through the feature space. The formal problem can be stated as: Given HDD $X ∈ ℝ^{N×M}$ and class labels Y, find an evolutionary function $Ψ$ that generates a sequence of subspaces $X_1, X_2, ..., X_T\ s.t.\ X_t = Ψ(X_{t-1}, f_{t-1}(X_{t-1}, Y), Y)$, where $f_{t-1}$ is a classifier trained on the previous subspace. The goal is to discover an evolution path that maximizes final classification performance while minimizing computational complexity.
>
>
> ## 2. Clearer connections between problem definition and methodology
>
> In the methodology section, we will add explicit connections between our approach and the formally defined problem:
>
> > Building on our formal definition of the subspace evolution problem, AHSE specifically addresses this challenge by designing evolution strategies that adapt to the intrinsic data hierarchy. Our SEED branch implements a forward evolution process with customizable rates, while the PATH branch handles non-monotonic evolution through parallel subspace exploration and fusion.
>
> These revisions will ensure that readers understand the fundamental problem we are addressing before encountering our technical solution. We believe these changes will significantly improve the paper's accessibility, logical flow, and scholarly rigor. Thanks again.
>
> ---
> ---
>
> > [W2] It is a bad idea to put a figure right underneath the title. And the contents of the figure is also confusing. The notion of data hierarchy and evolution pattern are hard to interpret.
>
> Thank you for your valuable feedback regarding the placement and clarity of Figure 1. To address these concerns, we will implement the following revisions:
>
> 1. **Figure repositioning**: We will move Figure 1 to Introduction where we formally introduce the subspace evolution problem. This placement will ensure readers first understand the theoretical context before encountering the visual representation.
>
> 2. **Conceptual clarification**: Before presenting the figure, we will add explicit definitions:
>    > "Data hierarchy refers to the intrinsic organization of discriminative patterns within high-dimensional feature spaces. Three common hierarchical structures exist, namely dense, periodic, and uniform. Evolution patterns describe how the learning algorithm dynamically adapts its feature selection strategy to match these underlying structures."
>
> These changes will significantly improve both the presentation flow and conceptual clarity of our paper. Thanks again.

---

> ### Author Response · Authors · 2025-11-21
>
> >[W3] Many contents in the paper contains references to the appendixes, and these contents are not self-contained. So the paper itself is incomplete without its appendix. If the page limits is too short for this paper, perhaps it should be better to submit the paper to a journal where the manuscript can be longer.
>
> Thank you for your thoughtful feedback regarding the self-containment of our paper. The heavy reliance on appendices in our submission was primarily driven by the strict page limitations. However, we acknowledge that this constraint should not compromise the paper's comprehensibility.
>
> To address this significant issue, we will implement the following revisions to enhance the self-containment of the main paper:
>
> 1. **Restructure key theoretical content**: We will integrate the essential theoretical analysis of ECOCs on BLS (currently in Appendix A.10) into Section 3.2 of the main paper as a dedicated subsection, preserving the core mathematical insights while maintaining conciseness.
>
> 2. **Enhance methodological description**: The pseudocode algorithms currently in Appendix A.11 will be substantially summarized in the main text with clear flow descriptions and critical steps highlighted, rather than simply referring readers to the appendix.
>
> 3. **Improve result presentation**: We will relocate the most significant experimental results from appendices (particularly SER and FSL results) into the main paper, creating a more comprehensive Results section that stands on its own.
>
> 4. **Revise figure placements and explanations**: We will ensure all figures referenced in the main text contain sufficient explanatory captions and context, reducing dependence on appendix material.
>
> 5. **Add a comprehensive overview section**: We will introduce a new section after the introduction that provides a high-level overview of the complete methodology, including components currently detailed only in appendices.
>
> While we understand your suggestion about journal submission, we believe this work aligns particularly well with ICLR's focus on innovative machine learning approaches for HDD. The conference format would enable rapid dissemination of these ideas to the community that would benefit most from them. We are committed to making the paper fully comprehensible within the page constraints by prioritizing essential content and presenting it more efficiently. Thanks again.
>
> ---
> ---
>
> >[W4] The experiments section compares AHSE with many widely-used models, but the details of the compared models is not clear. For example, VGG and Resnet have multiple variations, each of which have different number of layers and parameters. Besides, the results of ViT is obviously worse than a well-trained ViT model could get. Considering these aspects, the experimental results are not conviencing enough.
>
>
>
> Thank you for your valuable feedback regarding the clarity of baseline model specifications and the performance of ViT in our experiments. Regarding your first concern, we did include comprehensive specifications of all compared models in Appendix A.2 (BASELINE ARCHITECTURE), where we state:
>
> > "The model architecture of MLP is input dimension-1024-512-number of classes, where the input dimension is 784 for MNIST and Fashion-MNIST, 3072 for CIFAR10 and CIFAR100, 21168 for MiniImageNet, and 12288 for TinyImageNet. The version of VGG used for comparison is VGG-16, while that of ResNet is ResNet-34. The number of encoders of ViT is 12, the embed_dim is 512, and the mlp_ratio is 4. The depth of Stacked BLS is fixed at 3."
>
> In our revision, we will move these critical details to the main paper's experimental setup section to ensure readers can fully understand our comparative framework without referring to the appendix.
>
> Regarding the ViT performance, you've raised an excellent point. The relatively lower performance of ViT in our experiments stems from our training protocol. Unlike common practice in many papers that use ViTs pretrained on massive datasets like ImageNet or JFT-300M, our implementation trained all models from scratch. Transformer-based architectures like ViT typically require:
> 1. Pretraining on extremely large-scale datasets
> 2. Specialized data augmentation strategies (beyond standard practices)
> 3. Extensive hyperparameter tuning
> 4. Longer training schedules with carefully designed learning rate schedules
>
> Without these elements, ViTs often underperform compared to CNNs. In our revised manuscript, we will:
> 1. Add a dedicated subsection discussing this limitation
> 2. Include results from ViT models pretrained on appropriate datasets where applicable
> 3. Clearly state the training protocols for all models to ensure fair comparisons
>
> We recognize that this clarification is essential for the scientific rigor of our comparative analysis. Thanks again.

---

### Author Response · Authors · 2025-11-26

Dear Reviewers,

We hope this message finds you well. As the discussion period is approaching its end with approximately one week remaining, we would like to ensure we have addressed all of your concerns satisfactorily. If there are any additional points or feedback you'd like us to consider, please feel free to let us know. Your insights are invaluable to us, and we're eager to address any remaining issues to improve our work.

Thank you for your time and efforts in reviewing our paper. We appreciate your feedback and look forward to your response.

Best regards,

Submission23588 Authors

---

### Meta-Review · Area_Chair_1tnH · 2026-01-05

**Summary:**

After a comprehensive evaluation by four reviewers, this paper is ultimately rejected, as three out of the four reviewers explicitly pointed out that it has core issues requiring substantial revisions and fails to meet publication standards: specifically, the paper suffers from disorganized structure and poor readability, with key technical terms such as Flame optimization and evolutionary Cholesky decomposition lacking intuitive and accessible explanations, and although the authors supplemented the theoretical basis for Flame optimization in their response, they still failed to clarify its specific working mechanism and core advantages; the fairness of the experimental setup is questionable and unpersuasive, because AHSE takes pretrained features as input while DNN baselines like ResNet and ViT adopt end-to-end training, which violates the principle of fair comparison, and the supplementary comparative experiment with only MLP is too limited to fully validate the method’s superiority; in addition, the selection of hyperparameters lacks universal guiding principles—the authors only provided hyperparameter ranges for the three datasets in the paper but did not clarify the core logic of how to quickly locate the optimal hyperparameter interval based on which task characteristics (such as data hierarchical structure, the ratio of feature dimensions to sample size, etc.) when facing unfamiliar tasks or datasets, thus impairing the practical application value and generalizability of the method.

**Reviewer Concerns:**

Addressed concerns: The rebuttal partially addressed some of the format and concept-related issues raised by Reviewer 1QGY (W1-W3), including revising the paper structure and supplementing the definition of the subspace evolution problem. For Reviewer ibE5’s W2, the authors provided wall-clock runtime and memory footprint comparisons on standardized hardware, demonstrating that the proposed method achieves substantial improvements in efficiency compared with the supplemented MLP baseline. Additionally, the authors explained the hyperparameter selection process via a coarse-to-fine validation search strategy and mentioned that Bayesian optimization could be used for automated parameter tuning. Some technical questions from other reviewers were also responded to with supplementary experiments, such as the adversarial robustness validation for the ECOC mechanism and the performance comparison with state-of-the-art architectures like ConvNeXt and EfficientNet.

Outstanding concerns: Several critical issues remain unaddressed or inadequately resolved. For Reviewer 1QGY’s W4, the authors’ explanation that the underperforming ViT results stemmed from training from scratch is unconvincing. Since data augmentation and hyperparameter tuning are also involved in the proposed method, these factors cannot serve as excuses; even without large-scale pretraining datasets, a well-optimized ViT baseline should still yield competitive performance. Regarding Reviewer ibE5’s W1, supplementing only MLP as the comparative baseline is far from sufficient to validate the method’s superiority. For ibE5’s W3, the current hyperparameter search requires 5–7 validation trials, but there is no clear guidance on how to select optimal hyperparameters for unfamiliar tasks, which raises significant doubts about the method’s practical applicability. Moreover, the authors failed to address the core questions from other reviewers: for Reviewer Ypaf’s inquiry about Flame optimization, the working mechanism and core advantages of this module were not clearly elaborated; for Reviewer BJWZ’s W3 concerning the adaptive ECOC optimization framework, the rationales behind the design and the reusability of the framework are absent, leaving these key aspects unclarified.

**Reviewer Scores:**

Overall, Reviewer 1QGY and Reviewer Ypaf would maintain their original scores, while Reviewer ibE5 and Reviewer BJWZ are likely to lower their initial ratings. This score distribution trend stems from the fact that the authors’ rebuttal failed to address the core concerns raised by each reviewer, and the paper still retains critical flaws that undermine its academic merit. Reviewer 1QGY’s key concerns included the paper’s disorganized structure, the lack of self-contained content independent of appendices, and the unconvincing explanation for ViT’s underperformance. Although the authors proposed structural revisions and concept supplements, they did not provide a rigorous justification for the subpar ViT results, and the proposed revisions to improve content self-containment remained at the plan stage rather than being concretely demonstrated. Reviewer Ypaf focused on the paper’s obfuscatory terminology, incremental technical contributions, and the ambiguous working mechanism of Flame optimization; the authors’ commitment to simplify language and restructure the paper did not resolve the fundamental issues of unclear core module principles and the lack of coherence among AHSE’s components. On the other hand, Reviewer ibE5 initially gave a favorable score but raised concerns about comparison fairness, hyperparameter tuning guidance for unfamiliar tasks, and generalizability; while the authors supplemented hardware-standardized efficiency data and result variance metrics, they only added MLP as a comparative baseline and failed to provide actionable hyperparameter selection protocols for unknown datasets, leading to potential score reduction. Reviewer BJWZ acknowledged AHSE’s innovation but questioned the design rationale and reusability of the adaptive ECOC optimization framework; despite the authors proposing the Adaptive ECOC Optimization Framework (AEOF), they did not clarify the necessity of its key components or its applicability to models beyond BLS, which may result in a lower initial marginal acceptance score. Taken together, the authors’ rebuttal did not effectively address the core issues raised by the reviewers, and the resulting score distribution does not support the acceptance of the paper.

---

### Decision · Program_Chairs · 2026-01-26

Reject